# RYBP regulates selective genomic binding of TrxG and PcG components in embryonic stem cell fate control

Chao Wei [1,2,3,4,13], Jun Sun [4,13], Zhuoyan Liu [1,2,3,4,13], Mulan Wang [1,2,3,4,5,13], Jin Tan [1,2,3,4,13], Xiaona Huang [6], Ranran Dai [1,2,3], Kang Su [1,2,3], Shiwen Yang [2], Tara S R Chen [1], Qi Tian [1,2,3], Xiuxiao Tang [1,2,3], Xiaolin Tian [7], Dong-Feng Huang [1], Jin Bai [8], Xue Xiao [4], Xiaoting Shen [9✉], Juan Xia [2✉], Junjun Ding [1,2,3,4,10,11✉] & Lili Fan [12✉]

## Abstract

**Selective gene expression is pivotal in orchestrating human development. Specifically, trithorax group (TrxG) and polycomb group (PcG) components play crucial roles in transcriptional activation and repression of state-specific stem cell expression programs, yet the mechanisms underlying their selective genomic binding remain poorly understood. In this study, we report that the polycomb repressive complex 1 (PRC1) subunit RYBP co-localizes with TrxG component WDR5 and selectively enriches PcG component RING1B in condensates in murine embryonic stem cells (ESCs). RYBP deficiency impairs the genomic binding of WDR5 and RING1B. Further, STAT3 excludes RING1B binding at RYBP-associated transcriptionally active loci. Additionally, RYBP depletion attenuates WDR5-dependent activation of DNA repair gene expression and facilitates the transition of ESCs to 2-cell-like cells. Finally, RYBP depletion disrupts RING1B deposition at lineage-specific genes, promoting ESC differentiation towards mesendoderm fate. These findings uncover RYBP as a regulator of selective genomic binding of TrxG and PcG components, providing insights into their roles in cell fate determination during development.**

**Subject Categories** Chromatin, Transcription & Genomics; Development; Stem Cells & Regenerative Medicine

## Introduction

The diversity of cell types during development arises from the selective expression of genes, dysfunction of which is implicated in various human diseases (Porcu et al, 2021; Prados et al, 2018). Epigenetic mechanisms, including DNA methylation and histone modifications, play pivotal roles in this selection process by providing heritable alterations without altering the DNA sequence (Okae et al, 2014; Wang et al, 2018a). Meanwhile, histone modifications exert influence over chromatin structure, thereby modulating transcriptional activity (Fukuda et al, 2023; Sungalee et al, 2021). Specifically, certain modifications such as H3K4me3 and H3K27ac, are linked with transcriptional activation, whereas others like H3K27me3 and H2AK119ub1 are associated with transcriptional repression (Ding et al, 2015; Endoh et al, 2012; Rose et al, 2016). The presence of distinct combinations of histone modifications at specific DNA regions in different cell types contributes to the establishment of their unique transcriptional networks (Mikkelsen et al, 2007).

Histone modifications are mediated by specialized epigenetic modification complexes (Ali and Tyagi, 2017; Gao et al, 2012). Initially, PcG and TrxG proteins were characterized as the global repressors and activators, respectively (Ingham, 1983, 1998; Lewis, 1978). Subsequently, their histone modification function for transcription control were further revealed (Ali and Tyagi, 2017; Ang et al, 2011; Blackledge et al, 2014; Rose et al, 2016). Polycomb complexes consist of Polycomb Repressive Complex 1 (PRC1) and Polycomb Repressive Complex 2 (PRC2), these complexes typically co-localize at genome to form bivalent domains for gene repression (Ku et al, 2008). Generally, PRC1 catalyzes H2AK119ub1, leading

[1]Department of Rehabilitation Medicine, The Seventh Affiliated Hospital, Zhongshan School of Medicine, Sun Yat-Sen University, Shenzhen, China. [2]Hospital of Stomatology, Guanghua School of Stomatology, Key Laboratory for Stem Cells and Tissue Engineering, Ministry of Education, Zhongshan School of Medicine, Sun Yat-Sen University, Guangzhou, China. [3]Advanced Medical Technology Center, The First Affiliated Hospital, Zhongshan School of Medicine, Sun Yat-sen University, Guangzhou, China. [4]Frontiers Medical Center, Tianfu Jincheng Laboratory, Department of Gynecology and Obstetrics, West China Second Hospital, West China Biomedical Big Data Center, West China Hospital/West China School of Medicine, Sichuan University, Chengdu, China. [5]Nanning Institute of Transfusion Medicine, Nanning Blood Center, Nanning, China. [6]Guangzhou Medical University-Guangzhou Institute of Biomedicine and Health (GMU-GIBH) Joint School of Life Sciences, Guangzhou Medical University, Guangzhou, China. [7]MOE Key Laboratory of Bioinformatics, School of Life Sciences, Tsinghua University, Beijing, China. [8]Cancer Institute, Xuzhou Medical University, Jiangsu, China. [9]Department of Reproductive Medicine Center, Guangdong Provincial Reproductive Science Institute (Guangdong Provincial Fertility Hospital), Guangzhou, China. [10]Shenzhen Eye Hospital, Shenzhen Eye Medical Center, Southern Medical University, Shenzhen, China. [11]School of Basic Medical Sciences, Guangdong Medical University, Guangdong, China. [12]Guangzhou Key Laboratory of Formula-Pattern of Traditional Chinese Medicine, School of Traditional Chinese Medicine, Jinan University, Guangzhou, China. [13]These authors contributed equally: Chao Wei, Jun Sun, Zhuoyan Liu, Mulan Wang, Jin Tan. ✉E-mail: shenxt@gdszjk.org.cn; xiajuan@mail.sysu.edu.cn; dingjunj@mail.sysu.edu.cn; fanlili@jnu.edu.cn

to chromatin compaction (Rose et al, 2016), whereas PRC2 catalyzes H3K27me3 onto DNA, further contributing to gene repression (Blackledge et al, 2014). As a result, H3K27me3 and H2AK119ub1 usually coexist at these repressed chromatin regions (Endoh et al, 2012). TrxG components are responsible for H3K4me3 and H3K27ac (Ang et al, 2011; Tie et al, 2009). Initially, it was believed that TrxG and PcG proteins were recruited to their respective binding sites by specific DNA elements (Bloyer et al, 2003). Recent studies have identified large regions of the genome where both active (H3K4me3) and repressive (H3K27me3) histone modifications coexist, along with concurrent enrichment of TrxG and PcG proteins (Onodera et al, 2016). However, a substantial portion of the genome are thought to be exclusively occupied by either TrxG or PcG proteins (Onodera et al, 2016). This raises a key question regarding how TrxG and PcG proteins selectively bind to the genome for transcription control.

PRC1 can be classified into two distinct categories: canonical PRC1, which is defined by the presence of CBX7, and non-canonical PRC1, which is characterized by the presence of RYBP (Morey et al, 2013; Tavares et al, 2012). RYBP plays a crucial role in modulating H2AK119ub1 levels on PcG target genes, facilitating communication between PRC1 and PRC2, and thereby inhibiting the expression of developmental genes (Morey et al, 2013; Rose et al, 2016). However, genes associated with RYBP exhibit reduced levels of RING1B and H2AK119ub modification, resulting in a higher transcriptional activity compared to those associated with CBX7. Functionally, RYBP-bound genes are predominantly implicated in the regulation of metabolic processes and cell cycle progression (Morey et al, 2013). Further studies have revealed that RYBP can activate the expression of pluripotency genes through a PRC1-independent pathway (Li et al, 2017). Our previous research also demonstrated that RYBP organizes long-range interactions between pluripotency-associated genes to facilitate co-activation via phase separation (Wei et al, 2022), and is required for super-enhancer activity in embryonic stem cells (Hong et al, 2024). In addition to RYBP, several PcG proteins, including PCGF6 and YY1 (Huang et al, 2019; Wang et al, 2018c), have also been reported to exhibit dual functions in transcriptional activation and repression. However, whether proteins with this characteristic are involved in the selective genomic binding of TrxG and PcG components remains to be determined.

Our findings demonstrate that RYBP co-localizes with both TrxG and PcG components at transcriptionally repressive loci. At transcriptionally active loci, it associates exclusively with TrxG protein WDR5 (lacking PcG protein RING1B). STAT3 prevents the genomic binding of RING1B at RYBP-targeted active loci. RYBP coordinates with WDR5 to activate the expression of DNA repair genes, which regulates pluripotency-to-totipotency transition. Additionally, RYBP coordinates with RING1B to inhibit the differentiation of ESCs towards mesendoderm cells. These findings provide insights into the selective binding of PcG and TrxG proteins, shedding light on their relationship with cell fate determination.

# Results

## RYBP co-localizes with TrxG component and selectively enriches RING1B in condensates

PcG and TrxG oppositely regulate transcription (Ang et al, 2011; Morey et al, 2013), while they exhibit overlap in certain components (Erokhin et al, 2023), raising the question of whether these shared proteins modulate their selective chromatin binding. The proteomes of the core factors for PcG (RING1B) and TrxG (WDR5) (Ali and Tyagi, 2017; Gao et al, 2012) were detected by immunoprecipitation mass spectrometry (IP-MS). In detail, the coding sequences (CDS) for FLAG and biotin-tagged RING1B, FLAG and biotin-tagged WDR5 were transfected into BirA mESCs (Fig. EV1A). Following drug selection, results from the western blot showed the successful establishment of cell lines with exogenously expressed RING1B or WDR5 (Fig. EV1B,C). Subsequently, nuclear extracts from control cells (BirA mESCs) and from those expressing FLAG and biotin-tagged RING1B were subjected to pull-down using streptavidin (SA) beads. The pulled-down proteins were then analyzed by mass spectrometry (Fig. EV1A). To identify the WDR5 interactome, the Stable Isotope Labeling using Amino Acids in Cell Culture (SILAC) method was employed. FLAG and biotin-tagged GFP mESCs, FLAG and biotin-tagged WDR5 mESCs were cultured in media containing different isotopes. Their nuclear extracts were then subjected to anti-FLAG bead pull-down followed by mass spectrometry. The signal intensities were compared with those from the control cells, and a cutoff of 1.5 times the control signal was established to identify significant interactions. A total of 52 proteins were identified as RING1B partners, and 269 proteins were identified as WDR5 partners (Fig. 1A). Meanwhile, 11 proteins, including RYBP, WDR18 and LUC7L2, were found to be shared between RING1B and WDR5 partners (Fig. 1A). To determine which of the above shared proteins has the potential to regulate the genomic binding of TrxG and PcG proteins, we firstly analyzed the characteristics of their partners. Cellular component analysis of these partners indicated their association with various nuclear bodies such as nucleolus and nuclear speck (Appendix Fig. S1A,B), along with the presence of multiple RNA binding proteins (Appendix Fig. S1C,D). Given numerous nuclear bodies and RNA binding proteins exhibit phase separation behavior (Dion et al, 2022; Hur et al, 2020; Shao et al, 2022), these findings indicate that the genomic binding of TrxG and PcG proteins might be regulated in phase separation, the notion is further supported by the high percentage of intrinsically disordered region (IDR) amino acid residues among RING1B and WDR5 partners (Appendix Fig. S1E). Among the 11 shared proteins, RYBP exhibited the highest proportion of IDR amino acid residues (Fig. 1B), which has been validated its function of triggering phase separation in our previous work (Wei et al, 2022). In details, the immunofluorescence staining results in mESCs showed that RYBP signals were enriched in the nucleus and exhibited punctate characteristics (Fig. 1C). By stably expressing exogenous EGFP-tagged RYBP in mESCs, the results also showed that RYBP displayed punctate characteristics (Fig. EV1D). Additionally, fluorescence recovery after photobleaching (FRAP) experiment was performed to detect the dynamics of RYBP puncta, the results indicated that the fluorescence of RYBP puncta quickly diminished after photobleaching, but could rapidly recover, suggesting the dynamics of RYBP puncta (Fig. EV1D; Appendix Fig. S1F). RING1B is known to be concentrated within Polycomb bodies (Hernández-Muñoz et al, 2005). Immunofluorescence imaging in ESCs revealed significant co-localization of RYBP with RING1B, along with the RING1B-catalyzed histone modification, H2AK119ub1 (Fig. 1D; Appendix Fig. S1G). Similarly, the core module components of histone-modifying complexes among TrxG, including WDR5, DPY30, ASH2L and RBBP5 (Ali

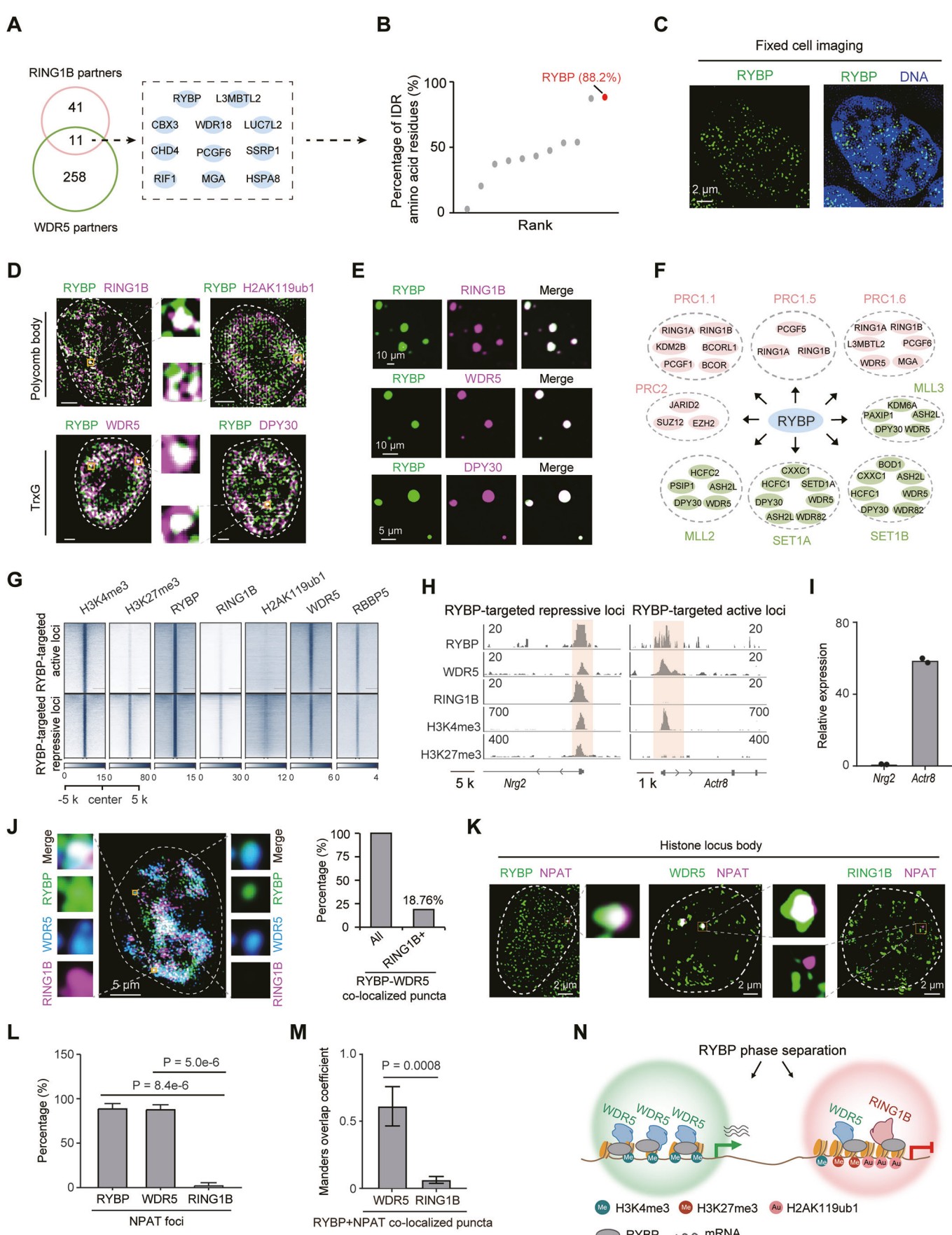

Figure 1.   **RYBP co-localizes with TrxG component and selectively enriches RING1B in condensates.**

(**A**) IP-MS showing the overlap of WDR5 and RING1B partners. (**B**) IDR analysis for the overlapped partners of WDR5 and RING1B. (**C**) SIM microscopic images of RYBP immunofluorescence in ESCs, three biological replicates. (**D**) Representative immunofluorescence images showing the co-localization between RYBP and RING1B, RYBP and H2AK119ub1, RYBP and WDR5, RYBP and DPY30. Scale bar denotes 2 μm. (**E**) RYBP droplets incorporate RING1B, WDR5 and DPY30 protein in vitro. (**F**) The complex analysis for the WDR5 and RING1B partners from both IP-MS and APEX. Proteins in red circles denote PcG components, proteins in green circles denote TrxG components. (**G**) Heatmaps showing the ChIP signal of H3K4me3, H3K27me3, RING1B, H2AK119ub1, WDR5 and RBBP5 at RYBP binding sites. Data ref: GEO GSE136584, 2020; GSE96107, 2017. (**H**) ChIP-seq profiling at the *Nrg2* and *Actr8* locus. (**I**) Histogram showing the relative expression of *Nrg2* and *Actr8* genes in ESCs, $n = 2$. (**J**) Representative image showing the co-localization among RYBP, WDR5 and RING1B (left). Histogram showing the percentage of RING1B-enriched RYBP_WDR5 co-localized puncta (right). (**K**, **L**) Representative immunofluorescence images (**K**) and quantification (**L**) showing the co-localization between RYBP and NPAT, WDR5 and NPAT, RING1B and NPAT. Two-tailed Welch's *t* test; number of foci (from left to right) are 32, 41 and 25, presented as the mean ± SEM. (**M**) Manders overlap coefficient showing the co-localization between RYBP_NPAT co-localized puncta and RING1B, between RYBP_NPAT co-localized puncta and WDR5 puncta. Two-tailed Welch's *t* test; cell numbers (from left to right) are: $n = 171$, $n = 112$, presented as the mean ± SD. (**N**) The model showing that RING1B is selectively enriched in RYBP and WDR5 co-localized puncta. Source data are available online for this figure.

and Tyagi, 2017), also exhibited co-localization with RYBP (Figs. 1D and EV1E; Appendix Fig. S1G). Furthermore, purification of recombinant proteins from both PcG and TrxG demonstrated their incorporation into RYBP droplets (Fig. 1E; Appendix Fig. S1H). Thus, RYBP, a phase-separated protein, co-localizes with PcG and TrxG components in condensates, respectively.

To confirm the interaction of RYBP with multiple components from PcG and TrxG, both IP-MS and APEX-based proximity labeling were performed to identify the partners of RYBP (Appendix Fig. S1I). Biotin-tagged or APEX-tagged RYBP ESCs were established, respectively (Appendix Fig. S1J). The results revealed the presence of PcG subfamilies (PRC1.1, PRC1.5, PRC1.6 and PRC2), as well as TrxG subcomplexes (MLL2, MLL3, SET1A, and SET1B complex) (van Nuland et al, 2013) among RYBP partners (Fig. 1F). Co-immunoprecipitation (Co-IP) experiment was performed to further demonstrate their interactions. RYBP and RING1B are well-defined components of PcG (Gao et al, 2012), and we confirmed their interaction in ESCs (Fig. EV2A). Our recent study demonstrated the interaction between RYBP and exogenously expressed TrxG component WDR5 (Hong et al, 2024), we further confirmed their physical interaction between endogenous proteins (Fig. EV2B). Additional endogenous TrxG proteins including ASH2L and DPY30, were also validated for their interactions with RYBP (Fig. EV2B). Thus, both mass spectrometry and co-IP confirmed the interaction of RYBP with PcG or TrxG components.

Then we evaluated whether RYBP is a potential factor to regulate genomic binding of PcG and TrxG proteins based on its transcriptional regulatory function. Considering the opposing functions of PcG and TrxG in transcriptional control (Ali and Tyagi, 2017; Gao et al, 2012), RYBP is expected to have the capacity associated with both gene repression and activation. The deposition pattern of H3K4me3 and H3K27me3 is usually used to define transcriptional status across the genome (Harikumar and Meshorer, 2015). As a known transcription repressor, RYBP localizes at repressive loci enriched with H3K27me3 (Morey et al, 2013) (referred to as RYBP-targeted repressive loci) (Fig. 1G). However, RYBP was also observed at transcriptionally active loci characterized by enrichment in H3K4me3 but absence of H3K27me3 (referred to as RYBP-targeted active loci) (Fig. 1G). In total, RYBP deposited at both poised and active promoters, accounting for 38.9% and 25.8% of total loci, respectively (Appendix Fig. S1K). The higher expression levels of genes at RYBP-targeted active loci compared to those at RYBP-targeted

repressive loci corroborated their transcriptionally active status (Appendix Fig. S1L). Upon RYBP deficiency, RYBP-targeted repressive loci tended to be upregulated, while genes at RYBP-targeted active loci tended to be downregulated (Appendix Fig. S1M; Fig. EV2C–E). Conversely, overexpression of RYBP led to a significant increase in the expression of RYBP-targeted active genes, such as *Ezh2* and *Pcgf6* (Appendix Fig. S1N,O), while showed minimal effect on the global expression of RYBP-targeted repressive genes (Appendix Fig. S1P,Q). These findings reveal that RYBP is associated with both gene activation and repression.

Next, the co-localization pattern across RYBP, PcG and TrxG proteins was investigated. Results from ChIP-seq data revealed that TrxG components, specifically WDR5 and RBBP5, were present at both RYBP-targeted active and repressive loci (Fig. 1G). Similar results were also observed in our recent study, which showed that RYBP and TrxG proteins are co-enriched at super-enhancers (SEs) (Hong et al, 2024). However, their co-localization at active loci is not largely restricted to SEs (Appendix Fig. S1R). Compared to other TrxG components such as RBBP5 and SET1A, the binding signals of WDR5 on the genome exhibit a stronger correlation with the enrichment signals of H3K4me3 catalyzed by TrxG, regardless of whether it concerns global H3K4me3 or H3K4me3 at RYBP-targeted active and repressive loci (Appendix Fig. S1S). Conversely, RING1B was predominantly deposited at RYBP-targeted repressive loci, along with H2AK119ub1 (Fig. 1G). These findings were further supported by analyzing the deposition of RYBP, WDR5, RING1B, histone modifications, and mRNA levels at specific gene loci such as *Nrg2* and *Actr8* (Fig. 1H,I). Furthermore, the loss of RYBP also leads to a significant reduction in the WDR5 enrichment signals at a large number of RYBP-targeted active loci, as well as a decrease in the RING1B enrichment signals at RYBP-targeted repressive loci (Fig. EV2F,G). Therefore, at transcriptionally repressive loci, RYBP co-localizes with both TrxG and PcG components. At transcriptionally active loci, RYBP co-localizes with TrxG components with minimal PcG deposition.

Subsequently, the selective aggregation of RING1B into RYBP and WDR5 co-localized condensates was explored. Among RYBP and WDR5 co-localized puncta, both RING1B-enriched and RING1B-lacked puncta were observed (Fig. 1J). A total of 18.76% RYBP and WDR5 co-localized puncta obviously enrich RING1B (Fig. 1J; Appendix Fig. S1T). To further illustrate the result above, we examined the histone locus body (HLB), a nuclear body associated with the active transcription of *Hist* cluster genes (Fritz et al, 2018). NPAT serves as a marker for HLB (Fritz et al, 2018),

both RYBP and WDR5 were found to significantly aggregate within HLB (Fig. 1K–M). Conversely, RING1B was rarely observed within HLB (Fig. 1K–M).

Altogether, these results indicate RYBP co-localizes with WDR5 in biomolecular condensates and genome, selectively enriching RING1B (Fig. 1N).

## STAT3 co-localizes with WDR5 and excludes RING1B

We hypothesized the presence of distinct motifs in RING1B selectively enriched DNA regions, which recruit specific protein to determine the deposition of RING1B. The DNA regions co-localized by RYBP and WDR5 were partitioned into regions lacking RING1B and RING1B-enriched regions (Fig. 2A,B). RING1B-lacked regions exhibited a significant enrichment of the STAT3 motif, which was less abundant at RING1B-enriched regions (Fig. 2A). RNA-seq result in ESCs revealed that the RPKM value of *Stat3* was comparable to that of *Rybp* (Appendix Fig. S2A), suggesting that STAT3 was expressed in ESCs. STAT3 was predominantly deposited at RING1B-lacked regions (Fig. 2B). These regions exhibited a tendency towards transcriptionally active, characterized by enrichment of RNA Polymerase II (Pol_II), but low enrichment of H3K27me3 and H2AK119ub1 (Appendix Fig. S2B,C). In contrast, RING1B-enriched regions were depleted of STAT3 (Fig. 2B), which were transcriptionally repressive and characterized by high levels of H3K27me3 and H2AK119ub1 (Appendix Fig. S2C). These findings were further supported by the deposition patterns of STAT3, RYBP, RING1B, WDR5, and histone modifications nearby the TSS of *Atad2b* and *Fam110c* (Fig. 2C). The higher RPKM value of *Atad2b* compared to that of *Fam110c* further supports the transcriptionally active state of *Atad2b* and the transcriptionally repressive state of *Fam110c* (Fig. EV3A). Among RYBP and RING1B co-localized puncta, the vast majority (96.1%) lacked STAT3 (Fig. 2D,E). Conversely, the histone locus body (HLB), marked by NPAT and enriched in RYBP and WDR5, was devoid of RING1B (Fig. 1K–M), with 67.4% of NPAT foci enriched in STAT3 (Fig. 2E,F). Due to the technical challenges of co-staining the same cell for RYBP, WDR5, NPAT, RING1B, STAT3, and DNA, we instead examined their deposition patterns at genes located within NPAT foci. Histone cluster genes are known to be transcribed within NPAT foci (Fritz et al, 2018). The absence of RING1B deposition and the presence of WDR5 and RYBP around *Hist1h4a* and *Hist1h3a* genes further support the lack of RING1B from NPAT foci (Fig. 2G). Similarly, RING1B exhibited minimal aggregation within RYBP and STAT3 co-localized puncta (Fig. 2D,H; Appendix Fig. S2D). In conclusion, STAT3 predominantly occupies RING1B-lacked regions and is less enriched at RING1B-enriched regions.

Whether STAT3 is responsible for the exclusion of RING1B from genome was further investigated. A cell line expressing both STAT3-fused RYBP and EGFP construct induced by DOX was established (Figs. 2I and EV3B,C). After inducing the expression of STAT3-fused RYBP for 2 days, the expression level of total RING1B did not reduce (Fig. EV3C). Following the extraction of chromatin proteins, minimal TUBULIN was observed (Fig. EV3D), and the global deposition of RING1B at chromatin was impaired (Fig. EV3E), leading to the loss of more than 4,000 RING1B peaks (Fig. EV3F; Appendix Fig. S2E). The impairment of H2AK119ub1 at chromatin was also observed (Fig. EV3G). Regarding the

RING1B-enriched loci, inducing expression of STAT3-fused RYBP significantly increased the deposition of STAT3 at these loci (Fig. 2J; Appendix Fig. S2F), while the enrichment of RING1B at these loci significantly reduced (Fig. 2K; Appendix Fig. S2G). To further support the exclusion effect of STAT3-fused RYBP on RING1B from a gene expression perspective, we first examined whether the genes co-targeted by RYBP and RING1B showed a tendency to be upregulated following the induction of STAT3-fused RYBP expression. The results indicated that 1333 genes were upregulated, while only 547 genes were downregulated (Appendix Fig. S2H), with the upregulated genes being significantly enriched in developmental pathways (Appendix Fig. S2I). Next, whether the gene expression trends following the induction of STAT3-fused RYBP expression resembled those observed after RING1B-deficiency were assessed. A total of 3,220 genes were found to be significantly upregulated following RING1B-deficiency (Appendix Fig. S2J). After inducing the expression of STAT3-fused RYBP, 26.4% of these genes were also significantly upregulated, while only 10% of the genes were significantly downregulated (Appendix Fig. S2K). As an example, near the TSS of *Trim67* gene, RYBP, WDR5 and RING1B are enriched at this locus, while minimal STAT3 signal is observed (Fig. 2L). Upon induced expression of STAT3-fused RYBP, the enrichment of STAT3 increased, and RING1B signal decreased (Fig. 2L), leading to a significant increase in the expression of *Trim67* gene (Fig. 2M). Therefore, STAT3-RYBP fusion protein impaired the deposition of RING1B at RING1B-enriched loci (Fig. 2N).

To further support the exclusion effect of STAT3 on RING1B, STAT3 proteins were knocked down using shRNA (Fig. EV3H), resulting in an increased global intensity of RING1B at RING1B-lacked loci (Fig. 2O,P). Therefore, STAT3-deficiency increased the deposition of RING1B at RING1B-lacked loci (Fig. 2Q).

The above results indicate that RYBP collaborates with STAT3 to activate gene expression. However, a subset of RING1B-lacked regions still exhibits a lack of STAT3 signals (Appendix Fig. S2L). These regions are enriched with other factors, such as OCT4 and NMYC (Appendix Fig. S2M). Additionally, various factors, including KLF4 and NMYC, are enriched in both RING1B-lacked and RING1B-enriched regions, while NANOG and cMYC tend to be more enriched in RING1B-lacked regions (Appendix Fig. S2N). Notably, compared to several factors, RYBP exhibits a higher peak overlap ratio with STAT3 on the genome (Appendix Fig. S2O). These genes are primarily enriched in terms such as transcription regulation and DNA repair (Appendix Fig. S2P). There are also instances where STAT3-targeted loci lack RYBP enrichment, with these genes mainly associated with terms such as cholesterol homeostasis and endosomal transport (Appendix Fig. S2Q). In summary, multiple factors are likely involved in the transcriptional regulatory role of RYBP.

Finally, whether RYBP is required for the deposition of STAT3 was further investigated. RYBP depletion (*Rybp*⁻/⁻) was performed in *Rybp*ᶠˡ/ᶠˡ *Rosa26::CreERT2* (*Rybp*⁺/⁺) ESCs for treatment with 4-OHT treatment. A two-day treatment of 4-OHT is sufficient to deplete RYBP in *Rybp*⁺/⁺ ESCs (Wei et al, 2022), and this depletion does not result in a reduction in STAT3 expression (Fig. EV3I), but it reduced the global deposition of STAT3 at both RYBP bound and RYBP non-bound loci (Appendix Fig. S2R–T). Additionally, the depletion of RYBP led to decreased STAT3 signals at RING1B-lacked loci (Fig. 2R–T). Therefore, these results indicate that RYBP is required for the STAT3 deposition on chromatin.

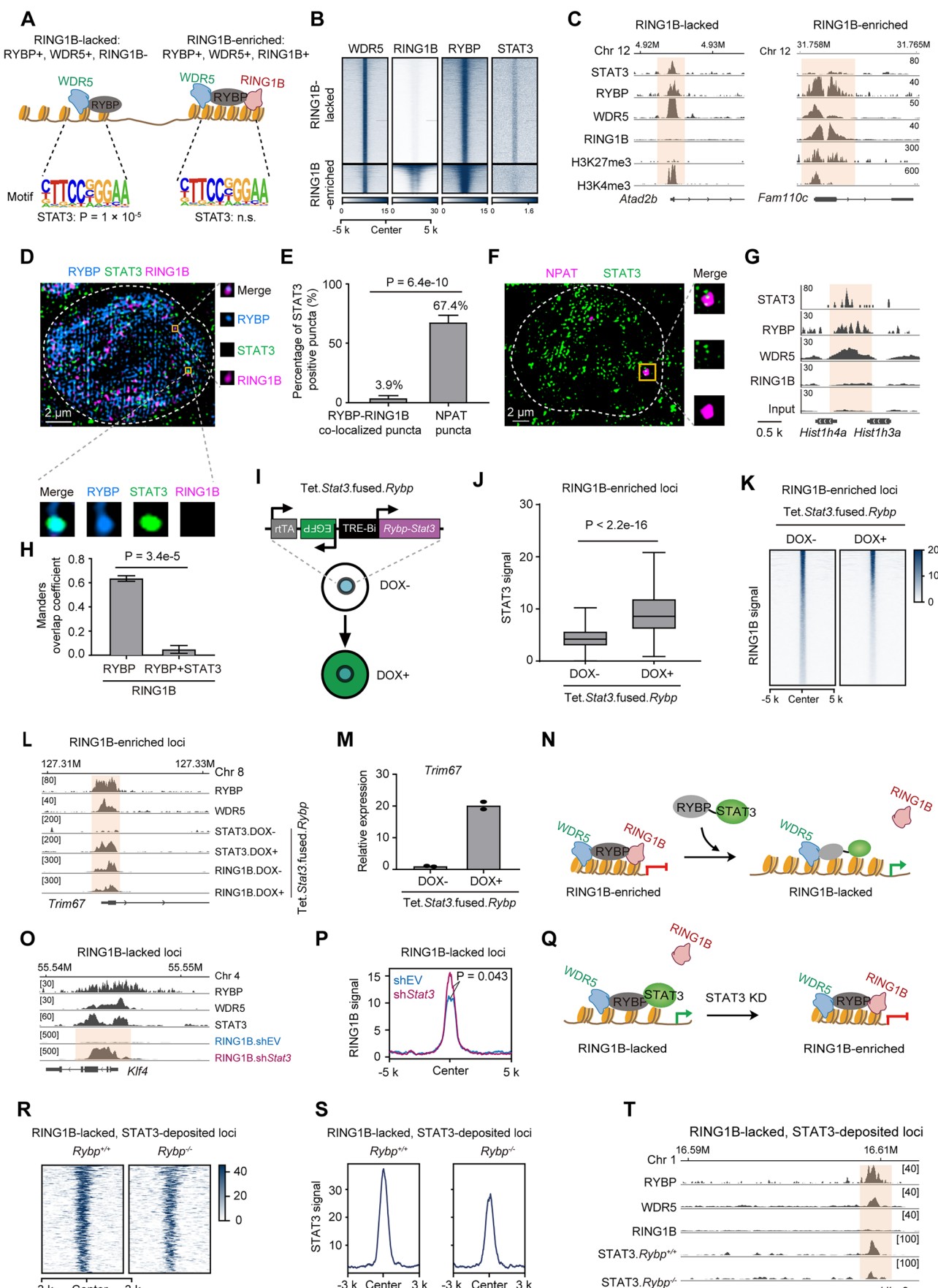

Figure 2.   STAT3 co-localizes with WDR5 and excludes RING1B.

(A) The model showing that STAT3 motifs are enriched at RING1B-lacked regions. "RING1B-lacked" denotes the DNA regions co-localized by RYBP and WDR5, but lacks RING1B deposition. "RING1B-enriched" denotes the DNA regions co-localized by RYBP, WDR5 and RING1B. (B) Heatmaps of RYBP binding loci are sorted by RING1B and WDR5 in ESCs. (C) ChIP-seq profiling at the *Atad2b* and *Fam110c* locus. (D) Representative image showing the co-localization among RYBP, STAT3 and RING1B. (E) Histogram showing the percentage of STAT3 positive puncta among RYBP_RING1B co-localized puncta and NPAT puncta; two-tailed Welch's *t* test, cell numbers are (from left to right): $n = 93$ cells, $n = 80$ cells, presented as the mean ± SD. (F) Representative immunofluorescence images and quantification showing the co-localization between NPAT and STAT3; two-tailed Welch's *t* test, cell numbers are (from left to right): $n = 93$ cells, $n = 80$ cells. (G) ChIP profiling at the *Hist1h4a* and *Hist1h3a* locus. (H) Histogram showing the percentage of RING1B positive puncta in RYBP_STAT3 co-localized puncta and RYBP puncta; two-tailed Welch's *t* test, cell numbers are (from left to right): $n = 91$ cells, $n = 59$ cells, presented as the mean ± SD. (I) Experimental pipeline for the establishment of STAT3-fused RYBP exogenously expressed cell line. TRE-BI denotes bidirectional promoter. (J) ChIP signal of STAT3 at RING1B-enriched loci upon inducing expression of STAT3-fused RYBP protein, one-tailed Wilcoxon. n values are 2331 for all groups, data are presented as box plots showing the median (centre line), the 25th and 75th percentiles (box limits), and the minimum and maximum values (defined as the whisker ends, which extend to data points within 1.5 times the interquartile range from the box). (K) Heatmap showing the ChIP signal of RING1B at RING1B-enriched loci upon inducing expression of STAT3-fused RYBP protein. (L) Representative loci showing the STAT3 and RING1B signal at RING1B-enriched loci upon inducing expression of STAT3-fused RYBP protein. (M) The expression of *Trim67* upon inducing expression of STAT3-fused RYBP protein, $n = 2$. (N) A model showing the inducible expression of STAT3-fused RYBP excludes RING1B. (O) Representative loci showing the RING1B signal at RING1B-lacked loci upon STAT3-deficiency. (P) Global signal of RING1B at RING1B-lacked loci upon STAT3-deficiency, one-tailed Wilcoxon. (Q) A model showing that STAT3 knockdown (KD) increased the RING1B signal at RING1B-lacked loci. (R, S) The heatmap and curve graph showing the ChIP signal of STAT3 at STAT3-deposited and RING1B-lacked loci following RYBP depletion. The RING1B-lacked loci refer to regions where RYBP and WDR5 are deposited, but RING1B is absent, as defined in (A). (T) Representative loci showing the ChIP signal of STAT3 at STAT3-deposited and RING1B-lacked loci after RYBP depletion. Source data are available online for this figure.

In conclusion, our findings suggest that STAT3 co-localizes with WDR5 at RYBP-targeted active loci, and excludes RING1B at these loci. RYBP is required for the genomic binding of STAT3.

## APEX-DNA-seq identifies RING1B-enriched and RING1B-lacked genes in RYBP condensates

Previous results indicate that RYBP selectively interacts with RING1B at specific genomic regions, and it also selectively forms condensates with RING1B. Next, we aim to determine whether RYBP regulates gene expression within these condensates. Thus, it is essential to identify the genes located within RYBP condensates. Protein proximal DNA could be identified by TSA-seq, and their distance to nuclear body was evaluated by TSA-seq score (Chen et al, 2018). Based on the theory, an APEX-based DNA sequencing (APEX-DNA-seq) methodology to evaluate the distance of DNA to RYBP puncta was established (Fig. 3A). Firstly, a cell line expressing RYBP-APEX fusion protein was established (Appendix Fig. S1I). Upon treatment with biotin-phenol and $H_2O_2$, RYBP-proximal DNA was specifically labeled (Fig. 3A; Appendix Fig. S3A–C). By comparing with the input DNA, the intensity of RYBP at DNA was quantified using an enrich score (ES) (Fig. 3A). Subsequently, FISH probes targeting DNA with different enrich scores were designed. Single-cell DNA FISH coupled with RYBP immunofluorescence was performed using these probes, and the distance to the nearest RYBP puncta was calculated (Fig. 3A). Consistent with the findings from TSA-seq, DNA with high enrich scores displayed a closer distance to RYBP puncta (Fig. 3B,C). Conversely, DNA with low enrich scores exhibited a farther distance from RYBP puncta (Fig. 3B,C). Utilizing the distance data from multiple FISH probes (Appendix Table S1), we derived a formula describing the distance of DNA to RYBP puncta based on the enrich score (Fig. 3D). According to the formula, the inflection point value of the curve is 0.2924. DNA fragments above this value are minimally affected by the enrichment score and are spatially very close to the puncta. Thus, DNA with an enrich score above 0.2924 was identified as residing within RYBP condensates (Fig. 3D). Altogether, APEX-DNA-seq successfully identified DNA located within RYBP condensates.

Whether gene promoters located within RYBP condensates exhibited RING1B-enriched or RING1B-lacked characteristics was further investigated. Among the regions with an enrich score above 0.2924, both WDR5-enriched and RING1B-lacked regions were observed (referred to as RYBP-WDR5 loci), as well as regions displaying both WDR5 and RING1B enrichment (referred to as RYBP-WDR5-RING1B loci). RYBP-WDR5 loci displayed transcriptionally active features, including the deposition of H3K4me3, high levels of H3K27ac, while minimal enrichment of H3K27me3 and H2AK119ub1 (Fig. 3E). Genes within RYBP-WDR5 loci genes were primarily associated with biological processes related to cell maintenance, such as cell cycle, DNA repair and cell division (Fig. 3F). In contrast, RYBP-WDR5-RING1B loci exhibited transcriptionally repressive features characterized by the deposition of H3K27me3 and H2AK119ub1 (Fig. 3E). Genes within RYBP-WDR5-RING1B loci were predominantly associated with developmental processes, including brain, kidney and liver development (Fig. 3G). The higher level of RNA expression observed in RYBP-WDR5 loci compared to RYBP-WDR5-RING1B loci further confirmed their transcriptional activity in these two types of loci (Fig. 3E; Appendix Fig. S3D). In addition, STAT3 motif is notably enriched at RYBP-WDR5 loci, but not prominently at RYBP-WDR5-RING1B loci (Appendix Fig. S3E). ChIP-seq analysis showed a high intensity of STAT3 enrichment at RYBP-WDR5 loci, in contrast to minimal enrichment at RYBP-WDR5-RING1B loci (Fig. 3E). The expression of RYBP-STAT3 fusion protein significantly reduced the enrichment of RING1B at RYBP-WDR5-RING1B loci (Appendix Fig. S3F,G).

Then whether the two clusters of genes above localize at RING1B-enriched or RING1B-lacked puncta was further investigated by 3D-FISH experiment. As examples, *Hist1h2ai*, a member of *Hist* cluster genes with an enrich score of 0.51, was deposited by WDR5 and lacks RING1B (Fig. 3H). Using DNA FISH probes targeting *Hist1h2ai*-included *Hist* cluster genes, we observed that they mainly localized within the histone locus body (Appendix Fig. S3H,I), where enriched RYBP, WDR5, STAT3, and lacked RING1B (Figs. 1K and 2F). 3D-FISH analysis further revealed the localization of *Hist1h2ai* nearby RYBP and WDR5 co-localized puncta (Fig. 3H; Appendix Fig. S3J). Similarly, *Pcdhb19*, a member

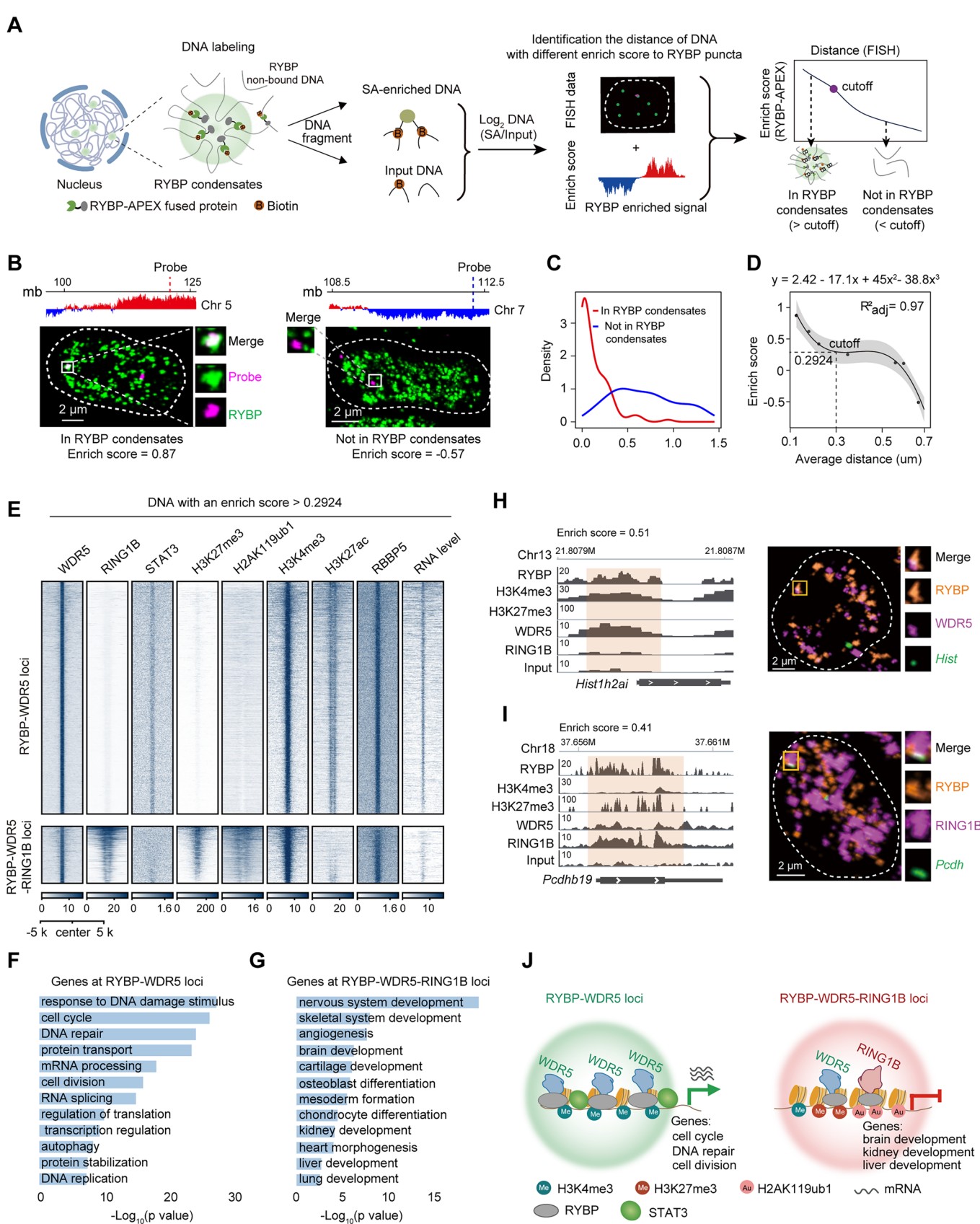

**A** DNA labeling — RYBP non-bound DNA — RYBP-APEX fused protein — Biotin — Nucleus — RYBP condensates — DNA fragment — SA-enriched DNA — Input DNA — Log₂ DNA (SA/Input) — Identification the distance of DNA with different enrich score to RYBP puncta — FISH data — Enrich score — RYBP enriched signal — Distance (FISH) — cutoff — In RYBP condensates (> cutoff) — Not in RYBP condensates (< cutoff)

**B** In RYBP condensates Enrich score = 0.87 — Not in RYBP condensates Enrich score = -0.57

**C** In RYBP condensates / Not in RYBP condensates

**D** $y = 2.42 - 17.1x + 45x^2 - 38.8x^3$, $R^2_{adj} = 0.97$, cutoff 0.2924

**E** DNA with an enrich score > 0.2924

**F** Genes at RYBP-WDR5 loci

**G** Genes at RYBP-WDR5-RING1B loci

**H** Enrich score = 0.51 — Hist1h2ai

**I** Enrich score = 0.41 — Pcdhb19

**J** RYBP-WDR5 loci — Genes: cell cycle DNA repair cell division — RYBP-WDR5-RING1B loci — Genes: brain development kidney development liver development

of the *Pcdh* cluster genes, with an enrich score of 0.41, was deposited by both WDR5 and RING1B (Fig. 3I). 3D-FISH analysis demonstrated the localization of *Pcdhb19* nearby RYBP and RING1B co-localized puncta (Fig. 3I; Appendix Fig. S3K). Therefore, both RING1B-enriched and RING1B-lacked genes were confirmed in RYBP condensates by 3D-FISH.

Figure 3. APEX-DNA-seq identifies RING1B-enriched and RING1B-lacked genes in RYBP condensates.

(A) Schematic diagram showing the flow of the APEX-DNA-seq methodology to identify DNA within RYBP condensates. (B) DNA FISH coupled with immunofluorescence images show the distance between the RYBP puncta and the FISH probe with high and low enrich score. (C) Quantification showing the distribution of distance between the RYBP puncta and the FISH probe with high (red line, corresponding to left panel of (B)) and low (blue line, corresponding to right panel of (B)) enrich score. (D) A mathematical formula reveals the relationship between the enrich score and the distance from DNA to RYBP puncta. (E) The heatmap showing the different ChIP signals at the WDR5-binding loci with or without RING1B deposition. The enrich scores of these regions are above 0.2924. (F, G) Gene ontology analysis revealing the biological processes of genes at RYBP-WDR5 and RYBP-WDR5-RING1B loci. (H) Left: Different ChIP signals at the *Hist1h2ai* locus. Right: 3D immuno-FISH image showing the co-localization between RYBP puncta, WDR5 puncta and probe of *Hist* cluster genes. (I) Left: Different ChIP signals at the *Pcdhb19* locus. Right: 3D immuno-FISH image showing the co-localization between RYBP puncta, RING1B puncta and probe of *Pcdh* cluster genes. (J) A model showing the histone modifications and gene expression within two types of condensates. Source data are available online for this figure.

Altogether, the established APEX-DNA-seq method facilitated the identification of DNA within RYBP condensates. Both RING1B-lacked and RING1B-enriched genes were identified in RYBP condensates, and they are associated with stem cell maintenance and development, respectively (Fig. 3J).

## RYBP depletion reduces the genomic binding of RING1B and WDR5 at genes in RYBP condensates

Given that RYBP and WDR5 co-localized loci selectively enrich RING1B (Figs. 1–3), we aimed to investigate how RYBP influences the aggregation and deposition of WDR5 and RING1B (Fig. 4A). After depletion of RYBP for 2 days, the total expression of TrxG components (WDR5 and ASH2L) and PcG component (RING1B) was not affected (Appendix Fig. S4A,B), the number of WDR5 and ASH2L puncta remained unchanged (Fig. 4B,C; Appendix Fig. S4C), while there was a notable decrease in the number of RING1B puncta (Fig. 4D,E). Furthermore, in vitro experiments with purified RYBP revealed its significant role in enhancing RING1B aggregation (Appendix Fig. S4D). These findings suggest that RYBP plays a role in promoting the aggregation of RING1B, without significantly impacting the puncta formation of WDR5.

TrxG proteins are known to facilitate the deposition of H3K4me3 and H3K27ac (Ang et al, 2011; Tie et al, 2009). RYBP depletion significantly reduced the deposition of WDR5, H3K4me3 and H3K27ac across RYBP-WDR5 loci (Fig. 4F,G; Appendix Fig. S4E,F). This conclusion was further validated through specific examples at *Twf2*, *Orai3*, and *Pitpnc1* gene promoters, where the decreased deposition of WDR5, H3K4me3 and H3K27ac, along with a reduction in mRNA levels after RYBP depletion were observed (Fig. 4F,G; Appendix Fig. S4F). To confirm these findings, ChIP-qPCR was performed to assess the deposition of TrxG components and H3K4me3 across the *Hist1h2bh* and *Hist1h4j* locus upon RYBP depletion. Both regions were found to be occupied by RYBP, WDR5, and H3K4me3 (Appendix Fig. S4G). WDR5-deficiency significantly reduced H3K4me3 deposition at these regions (Appendix Fig. S4H). Similarly, RYBP-deficiency significantly impaired the deposition of WDR5, RBBP5, and H3K4me3 at these regions (Appendix Fig. S4I–K). Then the deposition of RING1B, H2AK119ub1 and H3K27me3 were detected after RYBP depletion, considering that both PRC1 and

PRC2 components were identified among RYBP partners (Fig. 1F). A significant decrease of RING1B, H2AK119ub1 and H3K27me3 across RYBP-WDR5-RING1B loci were observed (Fig. 4H,I; Appendix Fig. S4L,M). This conclusion was validated by specific examples at *Loxl4*, *Mov10l1* and *Cdx1* promoters, where we observed decreased deposition of RING1B, H2AK119ub1 and H3K27me3, along with an upregulation of mRNA levels after RYBP depletion (Fig. 4H,I; Appendix Fig. S4M).

To further investigate the effects of RYBP phase separation on the genomic binding of WDR5 and RING1B, ChIP-seq was performed upon phase disruption. Our previously study reported that the deletion of 21 amino acids in RYBP (RYBP-ΔIDR21) disrupts its aggregation (Wei et al, 2022), we confirmed that the localization of RYBP-ΔIDR21 in the nucleus (Appendix Fig. S4N). As a control, TF-AA mutation within the zinc finger domain of RYBP impaired its physical interaction with ubiquitinated proteins (Arrigoni et al, 2006), we verified that the TF-AA mutation does not significantly reduce the number of RYBP puncta (Appendix Fig. S4O). Disruption of RYBP phase separation (RYBP-ΔIDR21) impaired the deposition of STAT3, WDR5 and H3K4me3 in numerous RYBP-WDR5 loci (Fig. EV4A; Appendix Fig. S4P,Q), as well as impairing the deposition of RING1B and H2AK119ub1 in numerous RYBP-WDR5-RING1B loci (Fig. EV4B; Appendix Fig. S4R). Regarding these signal-impaired loci, the TF-AA mutation reduced STAT3 binding at 49.8% of the loci, while 50.2% of the loci did not show a reduction (Appendix Fig. S4S), WDR5 did not decrease at 69.1% of the loci (Fig. EV4C). This indicates that RYBP's phase separation contributes to the signal enrichment at these sites by approximately 50% or even higher. Regarding the enrichment of RING1B, 77% of the loci showed a decrease in RING1B signal after the TF-AA mutation (Fig. EV4D).

In summary, our findings indicate that phase disruption of RYBP leads to a reduction in the deposition of TrxG protein WDR5 and TrxG-mediated histone modifications at RYBP-WDR5 loci. Furthermore, phase disruption of RYBP impairs the deposition of PcG protein RING1B at chromatin, as well as PcG-mediated histone modifications at RYBP-WDR5-RING1B loci.

## RYBP depletion downregulates DNA repair genes at WDR5-targeted loci, and upregulates lineage genes at RING1B-targeted loci

Given that DNA repair genes and lineage genes were enriched at RYBP-WDR5 loci and RYBP-WDR5-RING1B loci, respectively (Fig. 3), the impact of RYBP depletion on expression of DNA repair and lineage genes, as well as its role in DNA damage and ESC differentiation was investigated (Fig. 5A). Upon RYBP depletion,

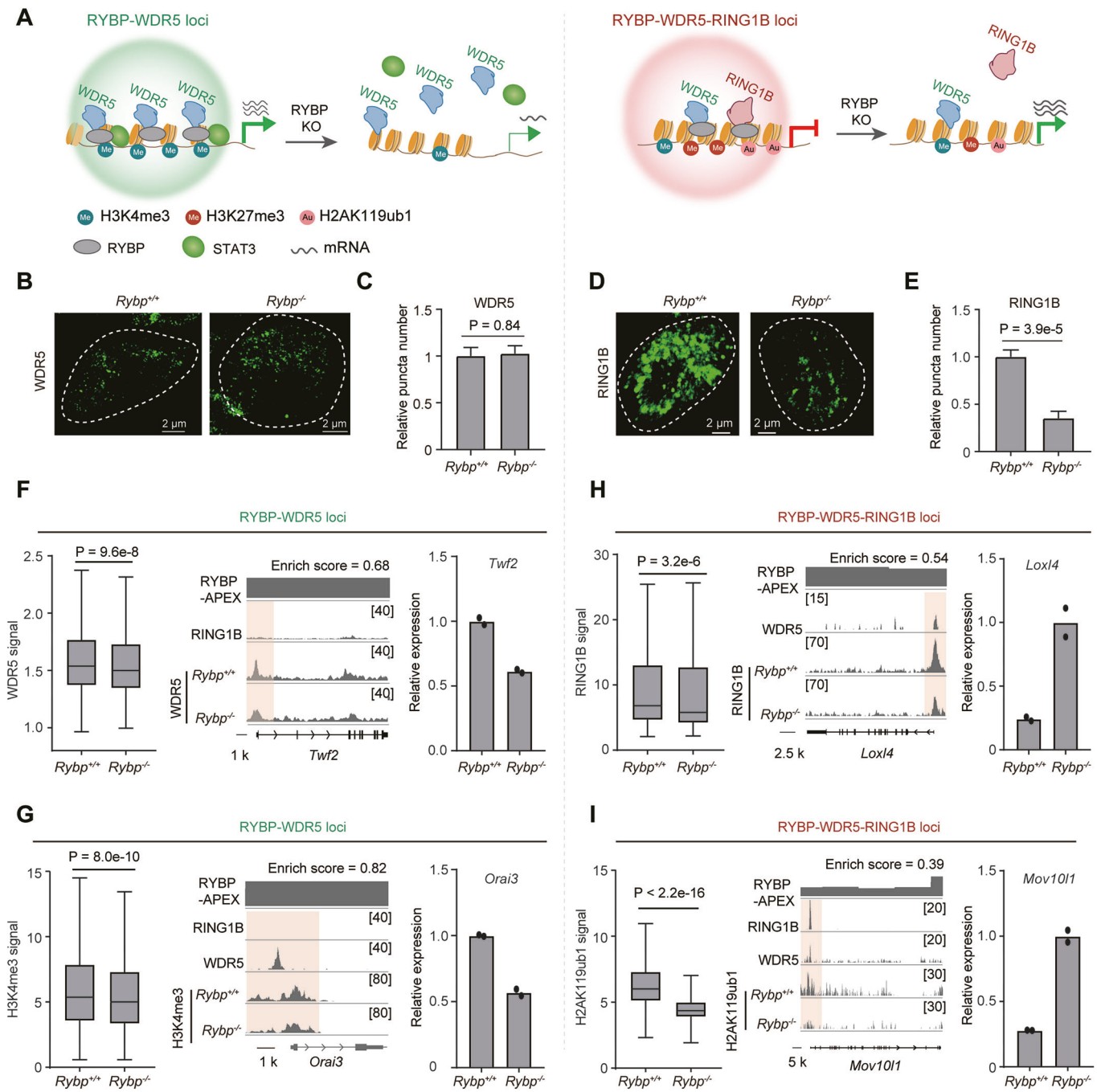

there is a decrease in the expression of several DNA repair-associated genes at RYBP-WDR5 loci (Fig. 5B, C). As an example, POLD1, which is known to facilitate DNA damage repair (Song et al, 2015), exhibits an enrich score of 0.66. WDR5 is found to occupy the *Pold1* locus with minimal RING1B deposition around its transcription start site (TSS) (Fig. 5D). RYBP depletion impaired the deposition of WDR5 and H3K4me3 at *Pold1* locus, and reduced the mRNA level of *Pold1* (Fig. 5D). This depletion also results in a notable increase in the number of γH2AX foci (Fig. 5E), indicating exacerbated DNA damage. To further determine the impact of phase separation on gene expression in ESCs, RNA-seq was

performed following phase disruption. The downregulated genes were significantly enriched in terms related to DNA repair (Appendix Fig. S5A). Among the downregulated genes upon RYBP phase disruption, 67.9% of them did not downregulated upon TF-AA mutation (Appendix Fig. S5B), these genes are also enriched in DNA repair term (Appendix Fig. S5C). Consequently, RYBP depletion downregulates the expression of DNA repair genes at RYBP-WDR5 loci, and induces DNA damage of ESCs.

The role of RYBP in ESC differentiation was further evaluated, because RYBP depletion upregulated the expression of majority lineage genes at RYBP-WDR5-RING1B loci (Fig. 5C,F). The impact

◄ **Figure 4. RYBP depletion reduces the genomic binding of RING1B and WDR5 at genes in RYBP condensates.**

(A) A model showing the impact of RYBP depletion on the enrichment of epigenetic regulators at two types of loci. (B, C) Representative immunofluorescence images and quantification showing WDR5 puncta number before and after RYBP depletion. Two-tailed Welch's *t* test, cell numbers are (from left to right): $n = 135$ cells, $n = 117$ cells, presented as the mean ± SEM. (D, E) Representative immunofluorescence images and quantification showing RING1B puncta number before and after RYBP depletion. Two-tailed Welch's *t* test, cell numbers are (from left to right): $n = 33$ cells, $n = 68$ cells, presented as the mean ± SEM. (F) Left: ChIP signals of WDR5 at the RYBP-WDR5 loci after RYBP depletion, one-tailed Wilcoxon, *n* values are 4463 for the two groups, data are presented as box plots showing the median (centre line), the 25th and 75th percentiles (box limits), and the minimum and maximum values (defined as the whisker ends, which extend to data points within 1.5 times the interquartile range from the box). ChIP-seq binding profiles (middle) of WDR5 at *Twf2* locus and its expression (right) before and after RYBP depletion, $n = 2$. (G) Left: ChIP signals of H3K4me3 at the RYBP-WDR5 loci after RYBP depletion, one-tailed Wilcoxon, *n* values are 5393 for the two groups. Data are presented as box plots showing the median (centre line), the 25th and 75th percentiles (box limits), and the minimum and maximum values (defined as the whisker ends, which extend to data points within 1.5 times the interquartile range from the box); ChIP-seq binding profiles (middle) of H3K4me3 at *Orai3* locus and its expression (right) before and after RYBP depletion, $n = 2$. (H) Left: ChIP signals of RING1B at the RYBP-WDR5-RING1B loci after RYBP depletion, one-tailed Wilcoxon, *n* values are 1934 for the two groups. Data are presented as box plots showing the median (centre line), the 25th and 75th percentiles (box limits), and the minimum and maximum values (defined as the whisker ends, which extend to data points within 1.5 times the interquartile range from the box); ChIP-seq binding profiles (middle) of RING1B at *Loxl4* locus and its expression (right) before and after RYBP depletion, $n = 2$. (I) Left: ChIP signals of H2AK119ub1 at the RYBP-WDR5-RING1B loci after RYBP depletion, one-tailed Wilcoxon, n values are 2501 for the two groups, data are presented as box plots showing the median (centre line), the 25th and 75th percentiles (box limits), and the minimum and maximum values (defined as the whisker ends, which extend to data points within 1.5 times the interquartile range from the box); ChIP-seq binding profiles (middle) of H2AK119ub1 at *Mov10l1* locus and its expression (right) before and after RYBP depletion, $n = 2$. Source data are available online for this figure.

of RYBP-deficiency on mesendoderm cells (MEC) was first investigated, since RYBP depletion increased the expression of majority genes at RYBP-WDR5-RING1B loci associated with mesoderm and endoderm cells (Appendix Fig. S5D). For instance, *Bmp1*, with an enrich score of 0.45, was occupied by WDR5, and RING1B around its TSS. RYBP depletion impaired the deposition of RING1B and H2AK119ub1 at *Bmp1* locus, and increased its mRNA level (Fig. 5G). After 4 days of differentiation towards mesendoderm cells, cells derived from RYBP-depleted ESCs exhibited scattered epithelioid-like morphology compared to the flat colony morphology of cells derived from WT ESCs (Fig. 5H). Furthermore, after 5 days of differentiation, the expression of mesendoderm genes, including *Mixl1*, *Gsc* and *Sox17*, was significantly promoted in cells derived from RYBP-depleted ESCs compared to WT ESCs (Fig. 5I). To further validate the influence of RYBP on ESC differentiation, AP staining of ESCs and embryoid body (EB) differentiation was assessed upon RYBP-deficiency. IAA-induced RYBP degradation cells (RYBP-AID) were used for alkaline phosphatase (AP) staining, the results indicated that the absence of RYBP significantly increases the proportion of partially and fully differentiated ESC colonies (Appendix Fig. S5E). Following RYBP knockout and EB differentiation, the expression of several germ layer-related genes was significantly increased compared to the control group (Appendix Fig. S5F–H), including ectoderm-related genes such as *Gfap* and *Blbp* (Appendix Fig. S5F); mesoderm-related genes such as *Mesp1* and *Twist1* (Appendix Fig. S5G), as well as endoderm-related genes *Gata4*, *Sox17* and *Eomes* (Appendix Fig. S5H). In contrast, RYBP overexpression did not significantly increases the proportion of partially and fully differentiated ESC colonies (Appendix Fig. S5I). This result is consistent with the finding that there is no significant change in repressed genes following the overexpression of RYBP in ESC (Appendix Fig. S1Q). Following RYBP overexpression during EB differentiation, compared to the control group, the expression of three germ layer-related genes significantly decreased (Appendix Fig. S5J–L). In addition, the upregulated genes upon RYBP phase disruption were also significantly enriched in terms related to development (Appendix Fig. S5M). Among these upregulated genes, 70.4% of them did not upregulated upon TF-AA mutation (Appendix Fig. S5N), these genes are also enriched in

developmental terms (Appendix Fig. S5O). Therefore, RYBP depletion upregulates the expression of lineage genes at RYBP-WDR5-RING1B loci, and promotes ESC differentiation.

In summary, RYBP depletion induces ESC DNA damage and promotes their differentiation into various cell lineages, underscoring its crucial role in maintaining pluripotency.

## RYBP depletion-induced DNA damage facilitates ESC-to-2CLC transition

The dramatic upregulation of totipotency-associated genes after RYBP depletion was observed, including *Dux*, *Zscan4a*, *Zscan4b*, *Zscan4c*, *Zscan4d*, *Zscan4e* and *Zscan4f* (Fig. 6A; Appendix Fig. S6A). Among the significantly changed totipotency-associated genes, 72% of them exhibited upregulation, while only 28% of them showed downregulation (Fig. 6B). The significant upregulation of MERVL was also observed (Fig. 6C). These results indicated the positive role of RYBP-deficiency in ESC-to-2CLC transition. To validate this, RYBP-inducible knockout ESCs with a MERVL-GFP reporter (MERVL-GFP_*Rybp*$^{+/+}$) were established (Fig. 6D). Before RYBP knockout, only a minimal fraction of ESCs (< 1%) exhibited MERVL positivity (Fig. 6E,F). RYBP depletion (MERVL-GFP_*Rybp*$^{-/-}$ cells) in MERVL-GFP_*Rybp*$^{+/+}$ was achieved through 4-OHT induction (Appendix Fig. S6B). After inducing RYBP depletion for 4 days, the proportion of MERVL-GFP positive cells increased with nearly 35-fold (Fig. 6F), indicating a substantial enhancement in the generation of 2CLCs. The remarkable generation of 2CLCs was not attributed to the side effects of 4-OHT, as evidenced by its minimal impact on 2CLC generation in an ESC cell line lacking RYBP-inducible knockout (Appendix Fig. S6C). To confirm the role of RYBP in totipotency regulation, MERVL-GFP_*Rybp*$^{+/+}$ ESCs were induced toward 2CLCs, and found that the mRNA levels of totipotency-associated genes were significantly higher in the RYBP KO group compared to the control group (Appendix Fig. S6D). These results collectively suggest that RYBP depletion promotes the ESC-to-2CLC transition.

Previous studies have indicated that DNA damage facilitates the ESC-to-2CLC transition (Bosnakovski et al, 2017), and RYBP depletion induces DNA damage in ESCs (Fig. 5E). Whether RYBP regulates ESC-to-2CLC transition via inducing DNA damage was

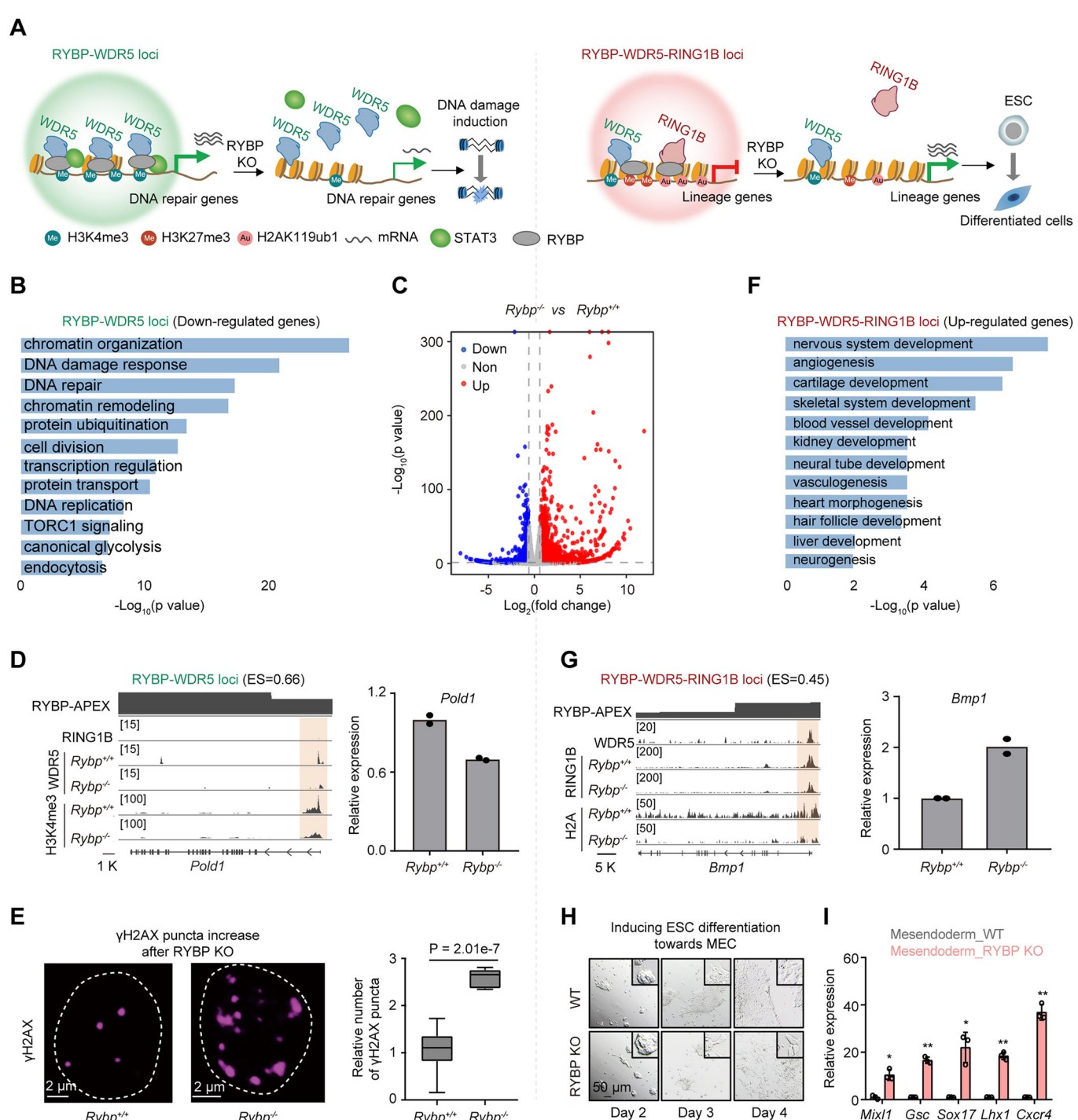

**Figure 5. RYBP depletion downregulates DNA repair genes at WDR5-targeted loci, and upregulates lineage genes at RING1B-targeted loci.**

(A) The model showing the roles of RYBP on the expression of DNA repair genes and lineage genes. (B) Gene ontology analysis showing the biological processes of downregulated genes at RYBP-WDR5 loci. (C) Volcano plot showing the gene expression changes in ESCs after *Rybp* knockout. (D) Left: Signals for the RING1B, WDR5 and H3K4me3 at the *Pold1* locus. Right: Histogram showing the relative expression of *Pold1* after RYBP depletion, $n = 2$. ES denotes enrich score. (E) Representative immunofluorescence images (left) and statistics (right) showing relative number change of γH2AX puncta after RYBP depletion. Two-tailed Welch's *t* test, cell numbers are (from left to right): $n = 109$ cells, $n = 56$ cells, data are presented as box plots showing the median (centre line), the 25th and 75th percentiles (box limits), whisker ends are the minimum and maximum values. (F) Gene ontology analysis showing the biological processes of upregulated genes at RYBP-WDR5-RING1B loci. (G) left: ChIP signals at the *Bmp1* locus, "H2A" denotes H2AK119ub1. Right: Histogram showing the relative expression of *Bmp1* after RYBP depletion, $n = 2$. (H) Representative fluorescence microscopy images showing ESC differentiation into MEC after RYBP depletion. (I) Relative expression of MEC-associated genes after inducing ESC differentiation to MEC for 5 days. Two-tailed Welch's *t* test, all n values are 3 technical replicates. *P* values (from left to right) are 0.012, 0.0024, 0.029, 0.0027 and 0.0029, presented as the mean ± SD. *$P < 0.05$, **$P < 0.01$. Source data are available online for this figure.

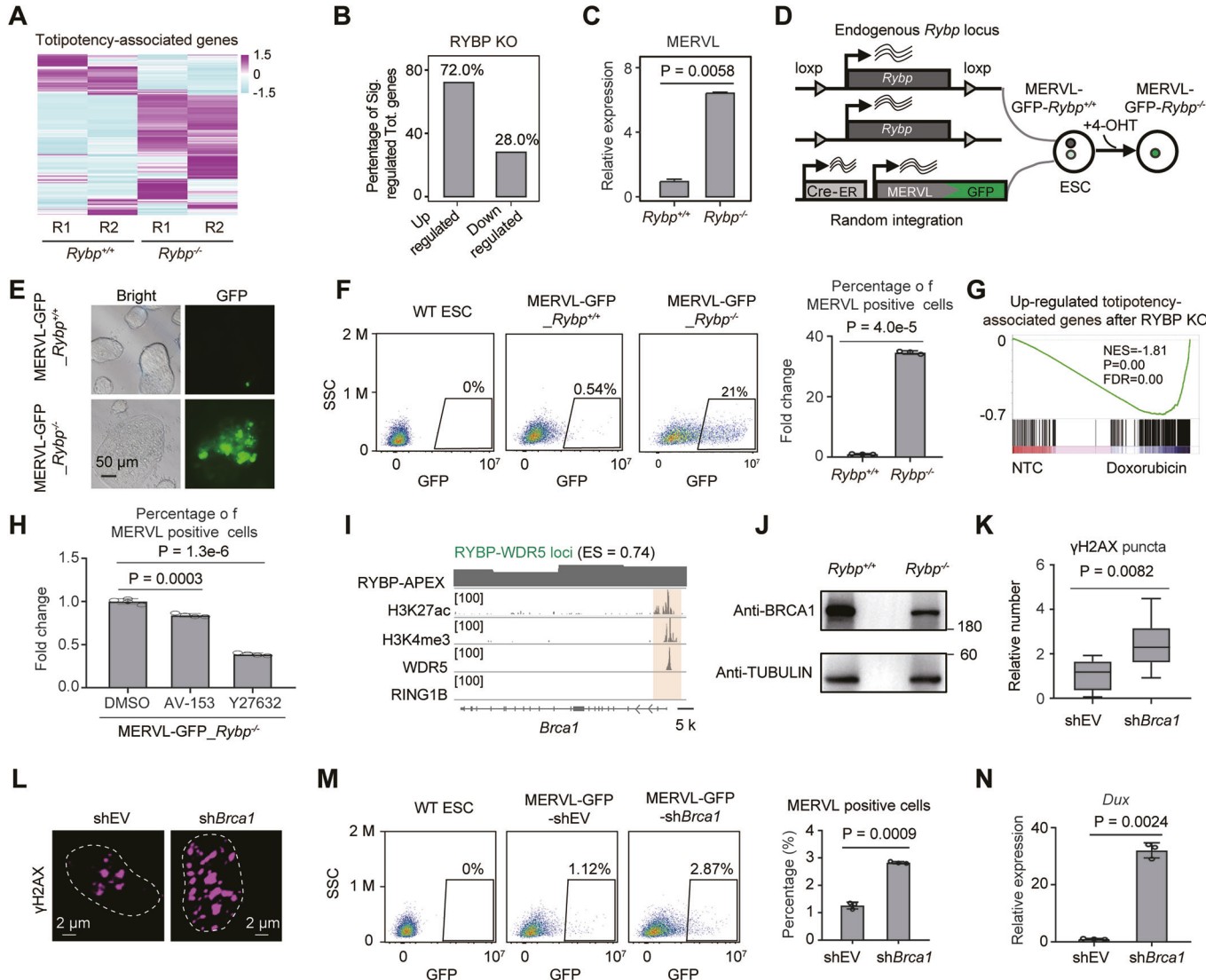

**Figure 6. RYBP depletion-induced DNA damage facilitates ESC-to-2CLC transition.**

(A) Relative expression of totipotency-associated genes before and after RYBP depletion. (B) The percentage of up- and downregulated genes among the significantly changed totipotency-associated genes after RYBP KO. (C) Relative expression of MERVL after RYBP depletion. Two-tailed Welch's *t* test, all *n* values are 2, presented as the mean ± SD. (D) Experimental pipeline for the construction of MERVL-GFP_*Rybp*$^{+/+}$ cell line. (E) Representative fluorescence images showing the EGFP signals before and after RYBP depletion. (F) Flow cytometry and histogram showing the percentage of MERVL-GFP positive cells after RYBP depletion. Two-tailed Welch's *t* test, *n* = 3 for the two groups. The "Fold change" indicates the fold change in the proportion of MERVL-positive cells in the *Rybp*$^{+/+}$ group compared to the *Rybp*$^{-/-}$ group, presented as the mean ± SD. (G) Gene set enrichment analysis of the RNA-seq data after doxorubicin treatment. (H) Histogram showed that the percentage of relative MERVL-GFP positive cells after AV-153 and Y27632 treatment in MERVL-GFP_*Rybp*$^{-/-}$ cells. Two-tailed Welch's *t* test, *n* = 4 technical replicates for each group. The "Fold change" indicates the fold change in the proportion of MERVL-positive cells in the AV-153 or Y27632 group compared to the DMSO group, presented as the mean ± SD. (I) ChIP-seq binding profiles at *Brca1* locus. (J) Western blot showing the expression change of BRCA1 after RYBP depletion. (K, L) Histogram (K) and images (L) showing the number of γH2AX puncta after *Brca1* knockdown. Two-tailed Welch's *t* test, cell number are (from left to right): *n* = 80 cells, *n* = 135 cells, data are presented as box plots showing the median (centre line), the 25th and 75th percentiles (box limits), whisker ends are the minimum and maximum values. (M) Flow cytometry and histogram showing the percentage of MERVL-GFP positive cells after *Brca1* knockdown. Two-tailed Welch's *t* test, *n* = 3 for the two groups, presented as the mean ± SD. (N) Relative expression of *Dux* after BRCA1 knockdown. Two-tailed Welch's *t* test, *n* = 3 for the two groups, presented as the mean ± SD. Source data are available online for this figure.

investigated. The expression of numerous totipotency-associated genes increased upon both RYBP depletion or inducing DNA damage via doxorubicin treatment (Appendix Fig. S6E,F), and a subset of totipotency-associated genes exhibited upregulation following both RYBP KO and doxorubicin treatment (Fig. 6G). AV-153 and Y27632 have been reported to alleviate DNA damage

(Ryabokon et al, 2005; Yang et al, 2022). The addition of AV-153 or Y27632 significantly reduced the number of RYBP KO-induced γH2AX foci (Appendix Fig. S6G,H), as well as reducing the MERVL-positive cells among MERVL-GFP_*Rybp*$^{-/-}$ cells (Fig. 6H; Appendix Fig. S6I). These results revealed that anti-DNA damage reagent impaired RYBP KO-induced 2CLC generation.

Given that RYBP collaborates with WDR5 to activate DNA repair gene expression (Fig. 5B,C), whether RYBP depletion decreased the transcription of DNA repair genes to induce DNA damage, and further causing 2CLC generation, was further investigated. BRCA1, which plays a vital role in DNA break repair (Salunkhe et al, 2024). *Brca1* gene localized at RYBP-WDR5 loci in ESCs (Fig. 6I). RYBP depletion reduced the level of BRCA1 protein (Fig. 6J). To further investigate whether the reduction in *Brca1* expression leads to an accumulation of DNA damage and an increase in the proportion of 2CLC cells, we knocked down *Brca1* expression in ESCs (Appendix Fig. S6J). This knockdown resulted in a significant increased the number of γH2AX puncta (Fig. 6K,L), as well as the percentage of apoptotic cells (Appendix Fig. S6K). These results revealed that BRCA1-deficiency induced DNA damage in ESCs. Additionally, BRCA1-deficiency resulted in a significant increase in the proportion of MERVL-GFP positive cells (Fig. 6M), and also dramatically increased the expression of *Dux* (Fig. 6N). Therefore, RYBP depletion reduced the expression of BRCA1, BRCA1-deficiency promotes DNA damage and ESC-to-2CLC transition.

In summary, RYBP coordinates with WDR5 to activate the expression of DNA repair genes, which are downregulated upon RYBP depletion, and further causing DNA damage. The accumulation of DNA damage in ESCs induces the expression of totipotent genes, and further promotes ESC-to-2CLC transition.

# Discussion

In this study, RYBP co-localizes with TrxG and PcG components in phase separation, and RYBP is required for the genomic binding of WDR5 and RING1B. RYBP and WDR5 co-localized loci selectively enrich RING1B, and STAT3 is found to exclude RING1B at transcription active loci. Moreover, depletion of RYBP results in reduced enrichment of both WDR5 and RING1B, and decreases the expression of DNA repair genes and increases the expression of lineage genes. These alterations ultimately drive ESC-to-2CLC transition and ESC-to-MEC differentiation.

The components within phase separation typically consist of scaffold and client molecules (Woodruff et al, 2017). Scaffold molecules play a crucial role in triggering phase separation, while client molecules are recruited into condensates by scaffold molecules (Woodruff et al, 2017). IDR-based protein-protein interactions also contribute to the assembly of phase separation (Yang et al, 2020), and the sequence-encoded molecular grammar exists to underly their driving forces (Wang et al, 2018b). RYBP, which is abundant in IDRs, and our previous work has demonstrated that RYBP triggers the formation of phase separation, and highlighted its role in mediating the aggregation of CTCF (Wei et al, 2022). In this study, the necessity of RYBP for the aggregation of RING1B were further demonstrated, suggesting its function as a scaffold molecule. However, RYBP does not influence the aggregation of TrxG components. Previous studies have indicated that phase separation of TrxG enhances enzyme activity by concentrating its subunits (Namitz et al, 2023). RYBP has been shown to stimulate the enzyme activity of RING1B (Rose et al, 2016). These findings suggest a potential role of RYBP in regulating the catalytic activity of TrxG within nuclear condensates, which is supported by the observed decrease in H3K4me3 and H3K27ac deposition at chromatin upon RYBP depletion.

Disruption of RYBP phase separation via 21aa deletion revealed a significant reduction in the enrichment signals of STAT3 and WDR5 at a large proportion of loci. This underscores the importance of RYBP's phase separation for their genomic binding. However, among the RING1B signal-decreased loci upon RYBP phase disruption, the Zn-finger mutation also detected a significant reduction (77%). This result might be reasonable, as previously published data indicate that the zinc finger domain of RYBP is necessary for the integration of RYBP and RING1B into the PcG body (Arrigoni et al, 2006). Although there was no significant reduction in the number of RYBP puncta upon Zn-finger mutation, the content of RING1B within RYBP phase separation might reduced. This could be an important reason for the substantial decrease in RING1B enrichment across the genome. The alteration of components within RYBP phase separation, leading to changes in their genomic binding, also supports the importance of RYBP phase separation for RING1B genomic binding.

Our results indicate that RYBP participates in both transcriptional activation and repression. However, at a global level, the expression of repressed genes does not show significant changes following RYBP overexpression. This finding may be reasonable, as RYBP represses the expression of these genes, which are already expressed at low levels. Even if there is further repression of their expression upon RYBP overexpression, the alteration may be minor. STAT3 assists RYBP-regulated gene activation, while it does not exclude the involvement of other factors in the transcriptional regulation process of RYBP. Some factors, such as ESRRB, KLF4, and NMYC, may participate in both the activation and suppression of RYBP, while others, similar to STAT3, primarily contribute to the activation of RYBP, such as NANOG and SOX2. These factors may also play a role in the exclusion of RING1B at activation sites. Nonetheless, the activation regulation of RYBP may not be entirely dependent on STAT3; other factors like OCT4 and NMYC may assist in the transcriptional regulatory role of RYBP at sites where STAT3 is not enriched. However, OCT4 and NMYC were also detected in RING1B-enriched regions, suggesting that they may not be the primary factors responsible for excluding RING1B from the STAT3-deficient within RING1B-lacked regions. It is possible that other factors are involved in this exclusion of RING1B in those regions.

The effect of RYBP on the genomic binding of STAT3 may involve both direct and indirect actions, as the loss of RYBP results in decreased enrichment of STAT3 at both RYBP-bound and non-bound sites. Previous studies have shown that STAT3 is necessary for maintenance of embryonic stem cells (Wang et al, 2017), and also regulates the fate determination of trophectoderm cells (Tai et al, 2014). The deficiency of RYBP impairs the genomic binding of STAT3, which may be one of the key reason why the absence of RYBP leads to the failure of blastocyst survival or the inability to yield extraembryonic cells during early embryonic development (Pirity et al, 2005).

RYBP deletion led to a significant reduction in the genomic signals of RING1B and STAT3, these signals did not completely disappear. The incomplete loss of RING1B may be attributed to the distinction between canonical PRC1 and non-canonical PRC1, with the former lacking RYBP (Morey et al, 2013; Tavares et al, 2012). Although canonical and non-canonical PRC1 often co-bind at

genomic loci (Morey et al, 2013), RYBP may not directly interacts with RING1B in canonical PRC1. Therefore, the impact of RYBP deletion on RING1B enrichment in canonical PRC1 may be relatively minor. Regarding STAT3, it possesses specific amino acid sequences that enable it to directly bind to specific DNA sequences (Timofeeva et al, 2012). The acetylation of STAT3 can facilitate its sequence-specific DNA binding ability (Wang et al, 2005). Additionally, various factors might regulate the genomic binding of STAT3. In embryonic stem cells, STAT3 can co-localize with multiple transcription factors, including NANOG, OCT4 and SOX2. The loss of OCT4 also impaired the genomic binding of STAT3 (Chen et al, 2008). Therefore, STAT3 binding may depend partly on direct interactions with DNA sequences or the recruitment of other factors, while RYBP might further enhance the genomic enrichment of STAT3, thereby collaboratively regulating gene expression.

Several mechanisms may contribute to the ESC-to-2CLC transition induced by RYBP deficiency. Reagent-induced DNA damage can stimulate the expression of totipotency-related genes, such as *Dux*, thereby augmenting the totipotency ratio of ESCs (Bosnakovski et al, 2017). Consequently, RYBP deficiency-induced DNA damage represents one of the underlying factors driving 2CLC generation. However, it is important to note that RYBP also acts to repress the expression of totipotency-related genes via PcG-dependent mechanisms (Cossec et al, 2018). Thus, the derepression of totipotency-related genes due to RYBP-deficiency may also contribute to the generation of 2CLCs.

# Methods

### Reagents and tools table

| Reagent/resource | Reference or source | Identifier or catalog number |
|---|---|---|
| **Experimental models** | | |
| R1 mESCs | This paper | N/A |
| RYBP-EGFP (R1) mESCs | This paper | N/A |
| RYBP-RFP (R1) mESCs | This paper | N/A |
| RYBP-EGFP-APEX (R1) mESCs | This paper | N/A |
| Tet on-RYBP-STAT3 (R1) mESCs | This paper | N/A |
| PTripz-RYBP (J1) mESCs | This paper | N/A |
| Piggybac-WT RYBP (R7.15) mESCs | This paper | N/A |
| Piggybac-WT RYBP_del21 (R7.15) mESCs | This paper | N/A |
| **Recombinant DNA** | | |
| pET28a-RYBP | This paper | N/A |
| pET28a-RING1A | This paper | N/A |
| pET28a-RING1B | This paper | N/A |
| pET28a-RBBP5 | This paper | N/A |
| pET28a-DPY30 | This paper | N/A |
| pET28a-WDR5 | This paper | N/A |
| pTripz-RYBP | This paper | N/A |
| Tet on-RYBP-STAT3 | This paper | N/A |

| Reagent/resource | Reference or source | Identifier or catalog number |
|---|---|---|
| **Antibodies** | | |
| RYBP | Abcam | ab5976 |
| TUBULIN | Abcam | ab6046 |
| RING1B | CST | 5694 |
| H2AK119ub1 | CST | 8240S |
| WDR5 | Santa | sc-393080 |
| γH2AX | Millipore | 05-636-I |
| H3K4me3 | Abcam | ab8580 |
| H3K27ac | Abcam | ab4729 |
| Goat anti-Rabbit IgG Alexa Fluor 488 | Thermo Fisher | A11034 |
| Goat anti-Rabbit IgG Alexa Fluor 594 | Thermo Fisher | A11032 |
| **Oligonucleotides and other sequence-based reagents** | | |
| Primers used for RT-qPCR analysis, see Appendix Table S2 | | |
| sh*Rybp*, targeting sequence CCAGGAAACCTCGCATCAATT | This paper | N/A |
| sh*Stat3*, targeting sequence CCTGAGTTGAATTATCAGCTT | This paper | N/A |
| **Chemicals, enzymes and other reagents** | | |
| Lipofectamine 2000 | Invitrogen | 11668019 |
| 4-hydroxytamoxifen | Sigma | H7904 |
| BSA | Sigma | A7906 |
| Dynabeads™ Protein G | Invitrogen | 10004D |
| formaldehyde solution | Sigma | F8775 |
| Protease Inhibitor Cocktail | Sigma | P8340 |
| Biotin Phenol (BP) | Sigma | SML2135 |
| Sodium Ascorbate | Sigma | A7631 |
| Trolox | Sigma | 238813 |
| DSG (Disuccinimidyl Glutarate) | Thermo Fisher | 20593 |
| Biotin | Sigma | B4501 |
| Dithiothreitol (DTT) | Sigma | D9760 |
| Puromycin Dihydrochloride | Thermo Fisher | A1113803 |
| RNaseA | Thermo Fisher | EN0531 |
| Proteinase K | Thermo Fisher | 25530049 |
| **Software** | | |
| RStudio | RStudio | https://posit.co/downloads |
| Python 2.7.16 | Python | https://www.python.org/ |
| FastQC v-0.11.8 | Babraham Bioinformatics | https://www.bioinformatics.babraham.ac.uk/projects/fastqc/ |
| Cutadapt v-2.4 | Marcel Martin | https://cutadapt.readthedocs.io/en/v2.4/installation.html |
| Bowtie2 v-2.3.5.1 | N/A | https://github.com/BenLangmead/bowtie2/releases/tag/v2.3.5.1 |

| Reagent/resource | Reference or source | Identifier or catalog number |
|---|---|---|
| Samtools v-1.3.1 | N/A | https://github.com/samtools/samtools/releases/tag/1.3.1 |
| Bedtools v-2.29.2 | N/A | https://github.com/arq5x/bedtools2/releases/tag/v2.29.2 |
| Star v-2.6.1b | N/A | https://github.com/alexdobin/STAR/releases/tag/2.6.1b |
| EdgeR v-3.16.5 | N/A | https://bioconductor.org/packages/3.4/bioc/src/contrib/edgeR_3.16.5.tar.gz |
| MACS2 v-2.2.9.1 | N/A | https://pypi.org/project/MACS2/2.2.9.1/ |
| David | N/A | https://davidbioinformatics.nih.gov/ |
| **Other** | | |
| Colloidal Blue Staining Kit | Invitrogen | LC6025 |
| EndoFree Plasmid Midi Kit | Cwbiotech | CW2105S |
| Pierce 660-nm Protein Assay Kit | Thermo Fisher | 22662 |
| His-tag Protein Purification Kit | Beyotime | P2226 |
| Qubit 1X dsDNA HS Assay Kit | Invitrogen | Q33230 |

## Cell lines and culture conditions

Wild-type (WT) mouse ESCs were cultured in ESC medium containing DMEM (Hyclone), 15% (v/v) fetal bovine serum (FBS, Hyclone), 0.1 mM β-mercaptoethanol (Sigma), 2 mM L-glutamine (Thermo Fisher), 0.1 mM nonessential amino acids (Thermo Fisher), 1% (v/v) nucleoside mix (Sigma), 1000 U/mL recombinant LIF (Millipore).

Mouse *Rybp^fl/fl Rosa26::CreERT2* (Wei et al, 2022) (*Rybp^+/+*) ESCs were cultured in ESC medium. RYBP-depleted ESCs (*Rybp^−/−*) were generated by supplementing the medium with 5 μM 4-hydroxytamoxifen (4-OHT, Sigma). To establish the fusion expression of RYBP and STAT3 cell line, the reverse tetracycline-controlled transactivator (rtTA) was first expressed randomly in ESCs. Then a plasmid containing a bidirectional promoter (TRE-BI) responsive to DOX and rtTA, which can independently drive the expression of EGFP and the STAT3-fused RYBP, was transfected into rtTA ESCs using Fugene (Promega). To establish an RYBP-inducible overexpression cell line, pTRIPZ-RYBP was initially transfected into ESCs. Following selection with 1 μg/ml puromycin, cells were induced with 1 μg/ml doxycycline (DOX) for overexpression.

## ESC differentiation

*Rybp^+/+* ESCs with or without 4-OHT were used for inducing into mesendoderm cells. These cells were cultured in a medium composed of half DMEM/F12 and half neurobasal medium supplemented with 1% $N_2$ supplement, 2% B27 supplement, 2 mM L-glutamine, 100 μM β-mercaptoethanol, and 25 ng/mL Activin A for 5 days.

*Rybp^+/+* ESCs and Tet-RYBP ESCs were seeded in low-adhesion plates and cultured in DMEM supplemented with 10% FBS for EB formation. In the differentiation medium for *Rybp^+/+* ESCs, 5 μM 4-OHT was added to induce the knockout of RYBP, while 2 μg/mL DOX was added to the differentiation medium for Tet-RYBP ESCs to induce RYBP overexpression.

## Induced reprogramming of ESCs toward 2CLCs

The MERVL-GFP vector was transfected into *Rybp^+/+* ESC cells, and they were induced towards 2CLCs by culturing in KnockOut DMEM with 20% KOSR, nonessential amino acids, Glutamax, 50 μg/mL BSA, 100 μg/mL L-ascorbic acid, 100 μM 2-mercaptoethanol, 10 ng/mL IL6, 10 ng/mL sIL-6R, 10 ng/mL IL6, 2 μM SGC0946 and 3 μM AS8351. RYBP depletion was achieved by adding 4-OHT.

## Recombinant protein expression and purification

The cDNA corresponding to the target proteins, including RYBP (NM_019743.3), RING1A (NM_009066.3), RING1B (NM_001360844.2), WDR5 (NM_080848.2), RBBP5 (NM_172517.2) and DPY30 (NM_001146222.1), was incorporated into pET28a vectors that contain His-tags and EGFP/mCherry, respectively. Protein expression and purification were performed following reported work (Wei et al, 2022). The above plasmids were transformed into BL21 *E.coli* cells, respectively. Cultures were subsequently grown overnight in LB medium with kanamycin at 37 °C. The cultures were then diluted and cultivated until an $OD_{600}$ of 0.6–0.8 was reached, after which IPTG induction took place at 16 °C overnight. For protein purification, the procedure outlined in the Protein Purification Kit was employed. Briefly, cell harvesting was performed, followed by lysis via sonication and clarification through centrifugation. The resultant His-tagged proteins underwent purification using resin, were washed, and then eluted with 250 mM imidazole. Verification of the proteins was conducted using SDS-PAGE, and they were subsequently dialyzed and concentrated with centrifugal filters. Finally, recombinant GFP/mCherry fusion proteins were visualized using fluorescence microscopy.

## Immunoprecipitation and mass spectrometry (IP-MS)

BirA transgene was firstly introduced into mESCs followed by G418 (300 μg/mL) selection to obtain BirA ESCs. Then the coding sequences (CDS) for FLAG and biotin-tagged GFP, FLAG and biotin-tagged RYBP, FLAG and biotin-tagged RING1B, FLAG and biotin-tagged WDR5 were transfected into BirA mESCs. Following drug selection with 1 μg/mL puromycin, stable cell lines of ^FB^GFP, ^FB^RYBP, ^FB^RING1B and ^FB^WDR5 were established, respectively. Based on the principles from published literature (Kim et al, 2009), BirA utilizes endogenous biotin within the cells to biotinylate the target protein.

Obtaining nuclear extracts: The cell pellet from the above cell lines was resuspended in three times the pellet volume of Nuclear Extract Buffer A, which contained 10 mM HEPES, 1.5 mM $MgCl_2$, 10 mM KCl, 0.5 mM DTT, 0.2 mM PMSF, and a protease inhibitor cocktail. The mixture was incubated on ice for 10 min, followed by centrifugation at $4300 \times g$ for 5 min. The supernatant was discarded, and the pellet was washed twice with Nuclear Extract Buffer A. After centrifugation and removal of the supernatant, Nuclear Extract Buffer C (containing 20 mM HEPES, 25% glycerol,

1.5 mM MgCl$_2$, 0.42 M NaCl, 0.2 mM EDTA, 0.5 mM DTT, 0.2 mM PMSF, and a protease inhibitor cocktail) was added at a ratio of 2.5 mL per $1 \times 10^9$ cells. The mixture was thoroughly mixed by pipetting and incubated at 4 °C for 1 h with rotation. Subsequently, it was centrifuged at $20,000 \times g$ for 30 min at 4 °C, and the supernatant was transferred to a new tube. The sample was then diluted with 100 volumes of Buffer D (containing 20 mM HEPES, 0.2 mM EDTA, 1.5 mM MgCl$_2$, 100 mM KCl, 20% glycerol, and 0.02% NP-40), and was dialyzed thoroughly. After transferring the sample to a new tube, it was centrifuged at $20,000 \times g$ for 20 min at 4 °C. Then the supernatant was transferred to a new tube as nuclear extracts.

RING1B and RYBP IP-MS: The protocol for immunoprecipitation followed by mass spectrometry (IP-MS) was adapted from our previous study (Costa et al, 2013). Nuclear extracts from BirA (control group), $^{FB}$RYBP and $^{FB}$RING1B ESCs were subjected to pre-clearing using 0.5 mL of Protein G agarose beads (Thermo Scientific) in IP DNP buffer containing Benzonase (Sigma) overnight at 4 °C, incubated with 0.5 mL SA agarose beads (Invitrogen), and rotated for 6 h at 4 °C. The beads underwent five washes with Buffer D. Elution of the bound proteins was performed by boiling the beads in Laemmli sample buffer for 5 min. For identification, samples were separated on a 10% NuPAGE 4%–12% Bis-Tris Gel (Thermo Scientific) and stained with GelCode Blue Safe Protein Stain buffer (Thermo Scientific). Protein bands were excised and sent for whole lane LC-MS/MS sequencing. The number of peptides corresponding to specific proteins in the $^{FB}$RYBP or $^{FB}$RING1B group exceeds two and is greater than 1.5 times that of BirA group, these proteins are identified as interactors.

WDR5 IP-MS: The SILAC labeling strategy was employed to identify the WDR5 protein interactomes, following a previously published protocol with slight modifications (Ong et al, 2002). In brief, $^{FB}$GFP ESCs were cultured in "Heavy" SILAC media, which contains $^{13}$C$_6$, $^{15}$N$_2$ L-Lysine and $^3$C$_6$, $^{15}$N$_2$ L-Arginine (CambridgeIsotope Laboratories). $^{FB}$WDR5 ESCs were cultured in "Light" SILAC media, which contains $^{12}$C, $^{14}$N L-Lysine and L-Arginine (Thermo). Then the SlLAC-labeled ESCs were trypsinized, washed with DPBS, and then processed for nuclear protein extraction. The affinity purification using anti-FLAG agarose beads (M2, Sigma) was conducted following the above protocol for SA agarose purification, with some modifications. Nuclear extracts were first pre-cleared with Protein G agarose (500 μl slurry per 10 mg protein) for overnight at 4 °C with continuous mixing in 15-ml tubes. Then the pre-cleared nuclear extracts were mixed with 500 μL of anti-FLAG M2 agarose beads. The immuno-complexes were subsequently eluted four times for 1 h each at 4 °C, utilizing 0.3 mg/mL FLAG peptide in Buffer D, which contained 0.02% NP-40. Protein bands were excised and sent for LC-MS/MS sequencing. Proteins in the $^{FB}$WDR5 group were identified as interactors if their signal intensities were greater than 1.5 times those of the $^{FB}$GFP group.

## APEX-based proximity labeling and pull down

FLAG-EGFP-APEX-tagged RYBP plasmid were transfected in mESCs. Stable cell lines were obtained after hygro selection (300 μg/mL). The backbone plasmid was transfected in mESCs as a negative control. The two types of cells were firstly cross-linked with 1% formaldehyde, after washing with PBS, the samples were cross-linked with 300 μM Disuccinimidyl glutarate. Then they were quenched with 2.5 M glycine. After three times of wash, samples were permeabilized using 0.2% NP-40 for 20 min. Subsequently, the cells were incubated with 500 μM Biotin phenol for 10 min, followed by incubation with 1 mM H$_2$O$_2$ for 1 min. After three washes with a washing buffer composed of 5 mM Trolox and 10 mM Na-ascorbate in DPBS, the cells were lysed in RIPA buffer (50 mM Tris, 150 mM NaCl, 0.1% SDS, 0.5% sodium deoxycholate, and 1% Triton X-100 in water). For protein pull-down, the labeled samples were sonicated (pulse on 5 s, pulse off 15 s, 50%) for three times. Then the 1 mg samples were incubated with streptavidin (SA) beads at room temperature for 1 h. The beads were then captured using a magnetic rack at room temperature for 2 h, and sequentially washed with RIPA buffer, 1 M KCl and 0.1 M Na$_2$CO$_3$, 2 M urea. Then the samples were eluted with elution buffer (3× protein loading buffer +2 mM biotin +20 mM DTT). The eluted proteins were subjected to SDS-PAGE, and after Coomassie brilliant blue staining, the target bands were excised and sent for mass spectrometry analysis. Two replicates were conducted for sequencing and analysis.

For APEX-DNA-seq, the labeled samples were subjected to cross-link reversal at 65 °C for 20 h. Subsequently, DNA extraction was performed using ethanol precipitation, followed by sonication to achieve a fragment length of 200–500 bp. The sonicated samples were then incubated with streptavidin (SA) beads at room temperature for 1 h, followed by further incubation at 4 °C for 8 h. After thorough washing with DWB buffer (1 M Tris-HCl, 0.5 M EDTA, 5 M NaCl, 10% Triton X-100), DNA elution was carried out, and the eluted DNA was purified for subsequent sequencing analysis.

## DNA dot blot

To assess the DNA labeling efficiency of the APEX enzyme, both the experimental group's DNA (biotin-labeled) and the control group's DNA (unlabeled with biotin) underwent heat treatment at 95 °C for 3 min. Subsequently, they were individually applied onto cellulose acetate membranes and exposed to ultraviolet light for 30 min to cross-link the nucleic acids onto the membrane. Following thorough washing with TBST, the membranes were blocked using a 5% BSA solution at room temperature for 1 h. After removing the blocking buffer, the membranes were incubated overnight at 4 °C with streptavidin-HRP (Invitrogen). Following three TBST washes, the membranes were incubated with a chemiluminescent substrate for 5 min. The treated membranes were then exposed using the Chemiluminescence Imaging System.

## RNA-seq

For the RYBP depletion, $Rybp^{+/+}$ ESCs were cultured in ESC medium containing 4-OHT for 4 days. Regarding the knock down of RYBP, lentivirus containing shRNA targeting $Rybp$ mRNA and empty vector (shEV) were prepared. Then mESCs were infected with sh$Rybp$ or shEV lentivirus for 96 h with puromycin (1 μg/ml) selection; For the RNA-seq of Tet.$Stat3$.fused.$Rybp$ cells, cells were treated with or without DOX for 4 days. Regarding RYBP overexpression, Tet.RYBP cells were treated with or without DOX for 4 days. For RYBP phase disruption, $Rybp^{+/+}$ ESCs with similar amount WT RYBP or RYBP-ΔIDR21 exogenously expression was

treated with 4-OHT for 4 days. Regarding Zn-finger mutation of RYBP, WT ESCs and TF-AA mutation ESCs were harvested. The total mRNA was purified to sequence with two replicates.

## ChIP-seq

ChIP-seq was performed according to a well-established protocol (Lee et al, 2006). Cells were cross-linked with 1% formaldehyde. After quenching formaldehyde with 2.5 M glycine, the samples were sonicated to fragment the DNA. Subsequently, Dynal beads were separately incubated overnight with antibodies targeting RING1B, STAT3, H3K4me3, H3K27ac and WDR5. Then the chromatin complexes were immunoprecipitated using antibody-coated beads overnight. After reversing the cross-linking, proteins and RNA were removed with proteinase K and RNase. Then the DNA fragments were purified for either sequencing. The DNA was purified to sequence with two replicates.

## RNA isolation and quantitative real-time PCR (RT-qPCR)

Total RNA was extracted from cells using Trizol reagent, followed by cDNA synthesis using a reverse transcriptase kit (Takara). Real-time quantitative PCR was conducted using SYBR qPCR Master Mix on a LightCycler 480 II system. The primer sequences utilized in the qPCR assays are provided in Appendix Table S2.

## Co-immunoprecipitation (Co-IP)

Cell pellets were resuspended in ice-cold nuclear extract buffer A (10 mM HEPES, 1.5 mM MgCl2, 10 mM KCl, 0.5 mM DTT, 0.2 mM PMSF, and protease inhibitor cocktail). After centrifuging and washing with nuclear extract buffer A, cell pellets were resuspended in nuclear extract buffer C (20 mM HEPES, 20% glycerol, 0.42 M NaCl, 1.5 mM MgCl2, 0.2 mM EDTA, 0.5 mM DTT, 0.2 mM PMSF, and protease inhibitor cocktail), and rotating the samples at 4 °C for 2 h. After centrifuging at $20,000 \times g$ at 4 °C for 30 min, removing the insoluble material. Then the supernatant was diluted and incubated with antibody-coated Protein G agarose. After three times of wash, the samples on beads were eluted by 1.5× SDS sample buffer for western blot. IgG and input samples were included as controls in the assay.

## In vitro droplet assay

Recombinant proteins were introduced into a solution containing 125 mM NaCl, 50 mM Tris-HCl, 10% glycerol, and 1 mM DTT to facilitate droplet formation. Subsequently, 10 µL of the resulting mixture was promptly loaded onto a glass slide and imaged using a microscope (Nikon) under consistent parameters.

## Immunofluorescence

After culturing mouse mESCs on gelatin-coated glass for 24 h, the cells were fixed with 4% paraformaldehyde (PFA) for 15 min at room temperature. Subsequently, the cells were washed twice with PBS and permeabilized with 0.25% Triton X-100 for 5 min. Following this, the cells were blocked with 10% BSA (Sigma) for 30 min at 37 °C and then incubated with primary antibodies overnight at 4 °C. After three washes with PBS, the cells were incubated with secondary antibodies for 1 h at room temperature. Following staining with DAPI for 15 min, the cells were imaged using either N-SIM or confocal microscopy.

## Immunofluorescence coupled with fluorescence in situ hybridization (FISH)

For fixation, cells were treated with 4% paraformaldehyde (PFA) for 12 min, followed by two washes with PBS to remove excess fixative. To reduce background, 1 mg/mL NaBH4 was applied, and cells were permeabilized using 0.25% Triton X-100 for 10 min at room temperature. Next, RNA degradation was achieved by incubating the cells with 100 µg/mL RNaseA at 37 °C for 45 min. Chromatin relaxation was facilitated by overnight incubation at 4 °C in 50% formamide in 2× SSC, followed by a heating step at 78 °C for 10 min and sequential dehydration with 70%, 85%, and 100% ethanol, each for 1 min. Probes in hybridization buffer (50% formamide, 10% dextran sulfate, and 1% Triton X-100 in 2× SSC) were subsequently added to the dish, and the cells were incubated at 37 °C for 20 h. Following six washes with 2× SSC, primary antibodies were applied and incubated overnight at 4 °C, after which secondary antibodies were introduced for 1 h at room temperature. After an additional six washes with PBS, imaging of the cells was performed using a Z-stack microscope. The distance between the two loci was calculated using NIS-Elements software (Nikon).

## Fluorescence recovery after photobleaching (FRAP)

ESCs stably expressing RYBP-EGFP were grown on gelatin-coated dishes for 24 h. FRAP experiments were carried out using a Nikon Eclipse Ti microscope equipped with a 100x oil-immersion objective lens and a 488 nm laser. Fluorescence intensity was quantified in the background region, and normalization was done using the fluorescence intensity of an adjacent unbleached cell.

## ChIP-seq data analysis

The initial assessment of FASTQ file quality was performed using FastQC (v0.11.8) to ensure high sequencing standards. Adapters were trimmed with Cutadapt (v2.4, https://cutadapt.readthedocs.io/en/v2.4/installation.html). The resulting trimmed reads were aligned to the mm9 reference genome using Bowtie2 (v2.3.5.1, https://github.com/BenLangmead/bowtie2/releases/tag/v2.3.5.1) with an end-to-end alignment strategy and the very sensitive preset option. Subsequently, the BAM files generated from the alignment were sorted and indexed using Samtools (v1.3.1, https://github.com/samtools/samtools/releases/tag/1.3.1).

Peak calling was conducted with MACS2 (v2.2.9.1), applying a false discovery rate (FDR) threshold of 0.01 to compare treatment and input samples, thus identifying significant binding sites. Peaks were deemed significant if they met the FDR criterion of less than 0.01.

To create coverage files in BigWig format, we utilized deepTools (v3.3.0, https://github.com/deeptools/deepTools/releases/tag/3.3.0) with the bamCoverage function, setting a bin size of 10 bases and normalizing using reads per kilobase per million mapped reads (RPKM). The analysis incorporated biological replicates by merging their corresponding BAM files with Samtools merge, followed by peak calling on the combined files to enhance the robustness and reproducibility of detected peaks. The resultant

ChIP-seq peaks were utilized for various bioinformatic analyses and data visualization.

For overlap analysis of peaks across different conditions or samples, bedtools intersect (v2.29.2, https://github.com/arq5x/bedtools2/releases/tag/v2.29.2) was employed. Heatmaps and profile plots were generated using the computeMatrix and plotHeatmap functions from deepTools to illustrate the distribution of ChIP-seq signals around identified peaks or gene bodies.

## RNA-seq data analysis

The Fastq data underwent quality checking using trim_galore, and subsequent alignment of paired-end reads to the MM9 reference genome was performed using STAR. Gene-specific read counts were generated using Htseq-count, followed by the calculation of differentially expressed genes.

## Definition of totipotency-associated genes

Genes that are significantly upregulated more than twofold in TLSC compared to ESC (GSE166216) are defined as totipotency-associated genes.

## Motif analysis

Motif analysis was performed using HOMER software suite (v4.11, http://homer.ucsd.edu/homer/), using the mm9 reference genome. The parameters for motif finding included the exact size of the regions specified in the BED file for the analysis and searching for motifs of length 8 base pairs. All other parameters were kept at their default settings.

## Criteria for setting the cutoff of APEX-DNA-seq

The cutoff value of 0.2924 was determined based on the inflection point of the fitted cubic polynomial curve ($y = 2.42 - 17.1x + 45 \times^{2} - 38.8 \times^{3}$). Specifically, the inflection point corresponds to the location where the second derivative equals zero, indicating a transition in curvature and therefore a change in the rate of increase of the response variable.

## Statistical analysis

All statistical analyses were conducted using R software (version 3.6.1). The figure legends provide detailed information on the replicates and testing methods used for each figure.

# Data availability

All sequencing files including ChIP-seq and RNA-seq data newly generated in this study were deposited in NCBI's Gene Expression Omnibus (GEO) repository (GSE261485).

The source data of this paper are collected in the following database record: biostudies:S-SCDT-10_1038-S44318-026-00788-y.

# Peer review information

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

## Acknowledgements

This research was funded by grants from the National Key Research and Development Program of China (2024YFA1106900, 2023YFA1800900), National Science Foundation for Distinguished Young Scholars of China (32425022), National Natural Science Foundation of China (32170798, U23A20445 and 32430031), the Tianfu Jincheng Laboratory (No. TFJCPI20250031) to JD, the National Natural Science Foundation of China (32470844) and the Natural Science Foundation of Guangdong Province, China (grant no. 2023A1515010197) to CW, the Natural Science Foundation of Guangdong Province, China (2023A1515010148), the Postdoctoral Fellowship Program of CPSF (GZC20241143), the China Postdoctoral Science Foundation (2024M752202), the National Natural Science Foundation of China (32400561) to JS the China Postdoctoral Science Foundation (2023M744086), the Natural Science Foundation of Guangdong Province, China (grant no. 2025A1515012775) to JT the Natural Science Foundation of Guangdong Province, China (grant no. 2023A1515110174) to XH; Basic and Applied Basic Research Foundation of Guangdong Province (2025B1515020011), Guangzhou Municipal Science and Technology Program key projects (2025A04J5154), the National Natural Science Foundation of China (82304746) to LLF; the National Natural Science Foundation of China (8247061458), Guangdong Province Science and Technology Innovation Strategic Special Fund Project (ZC202401) to XTS.

## Author contributions

**Chao Wei**: Data curation; Formal analysis; Funding acquisition; Investigation; Methodology; Writing—original draft; Writing—review and editing. **Jun Sun**: Data curation; Formal analysis; Funding acquisition; Investigation; Visualization; Methodology; Writing—review and editing. **Zhuoyan Liu**: Data curation; Formal analysis; Investigation; Visualization; Methodology. **Mulan Wang**: Data curation; Formal analysis; Investigation; Visualization; Methodology. **Jin Tan**: Data curation; Formal analysis; Funding acquisition; Investigation; Visualization; Methodology. **Xiaona Huang**: Data curation; Formal analysis. **Ranran Dai**: Data curation; Visualization. **Kang Su**: Data curation; Visualization. **Shiwen Yang**: Data curation; Formal analysis. **Tara S R Chen**: Writing—review and editing. **Qi Tian**: Data curation; Formal analysis. **Xiuxiao Tang**: Data curation; Formal analysis. **Xiaolin Tian**: Data curation; Formal analysis. **Dong-Feng Huang**: Writing—review and editing. **Jin Bai**: Writing—review and editing. **Xue Xiao**: Conceptualization; Supervision. **Xiaoting Shen**: Supervision. **Juan Xia**: Conceptualization; Supervision. **Junjun Ding**: Conceptualization; Supervision; Funding acquisition; Project administration. **Lili Fan**: Supervision; Funding acquisition.

Source data underlying figure panels in this paper may have individual authorship assigned. Where available, figure panel/source data authorship is listed in the following database record: biostudies:S-SCDT-10_1038-S44318-026-00788-y.

## Disclosure and competing interests statement

The authors declare no competing interests.

# Expanded View Figures

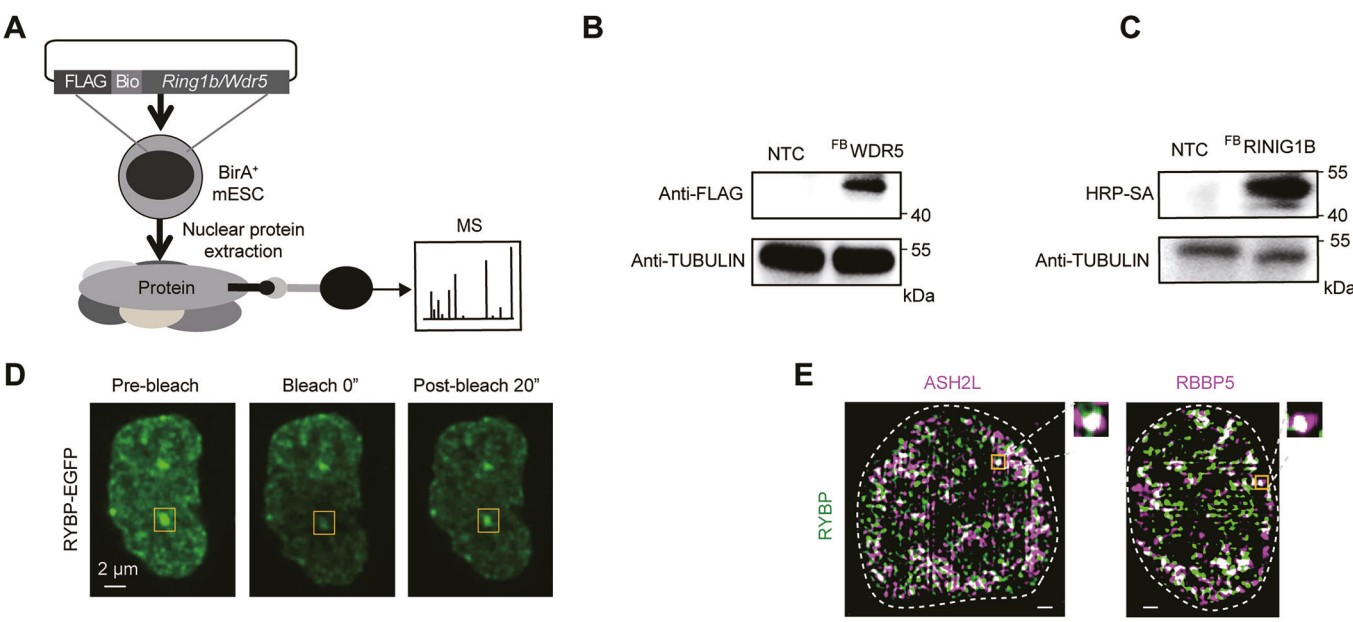

**Figure EV1. RYBP co-localizes with TrxG and PcG components in condensates.**

(A) IP-MS schematic diagram for identifying WDR5 and RING1B protein Interactome. (B, C) Western blot showing the exogenous expression of FLAG-biotin-tagged WDR5 ($^{FB}$WDR5) (B) and FLAG-biotin-tagged RING1B ($^{FB}$RING1B) (C). (D) Representative images of fluorescence recovery after photobleaching (FRAP) in mESCs expressing exogenous RYBP-EGFP. (E) Representative immunofluorescence images showing the co-localization between RYBP and TrxG components (ASH2L, RBBP5). Scale bar denotes 2 μm. Source data are available online for this figure.

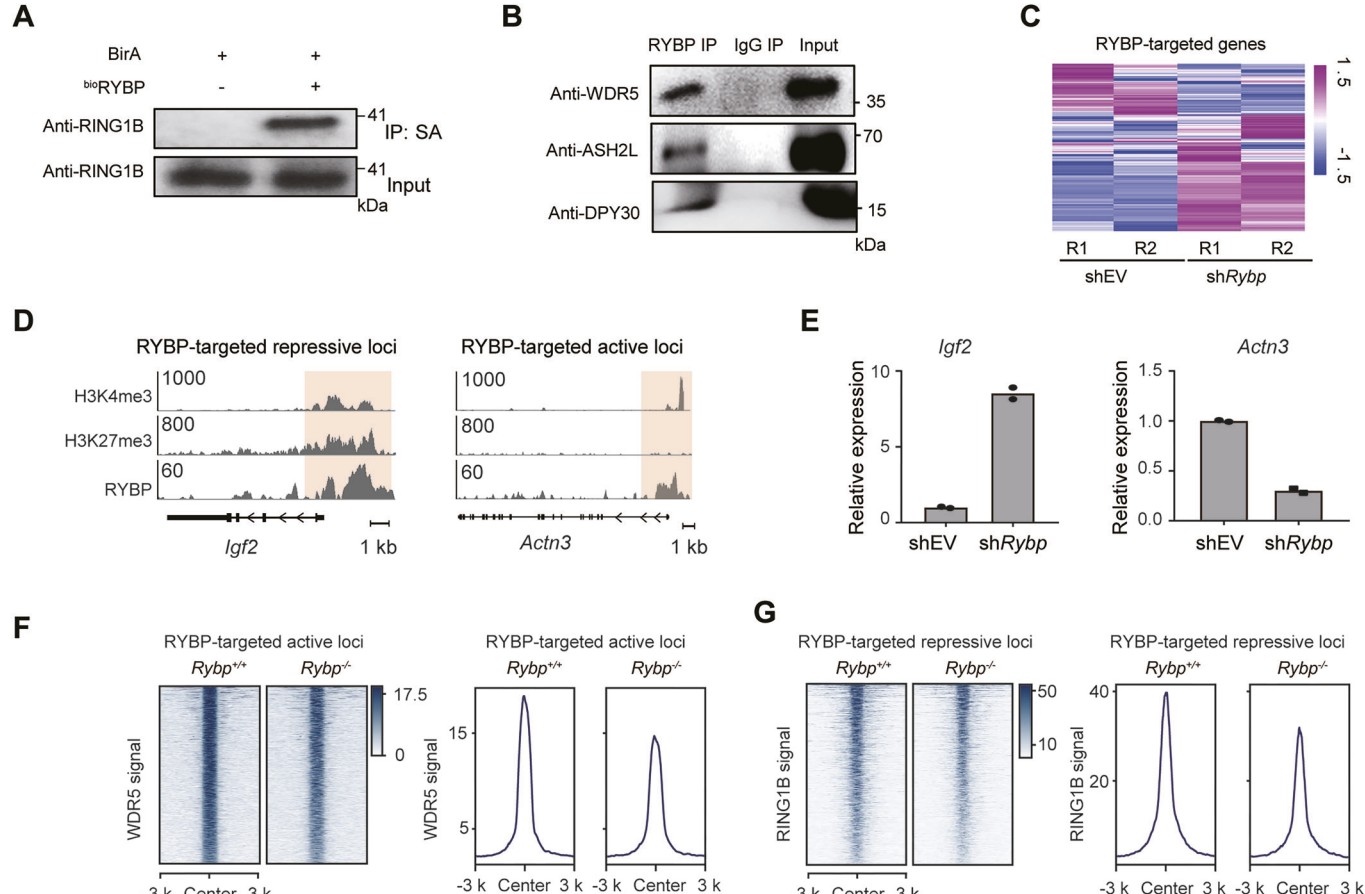

**Figure EV2. RYBP is involved in both transcriptionally active and repressive functions.**

(A, B) Co-IP validates the interaction between RYBP and RING1B (A), between RYBP and TrxG components (B) in ESCs. (C) Relative expression of RYBP-targeted genes in RYBP-deficiency ESCs (shRybp) compared with empty vector lentivirus-infected ESCs (shEV). (D) Deposition of RYBP, H3K4me3 and H3K27me3 at the Igf2 and Actn3 locus. (E) The expression changes of Igf2 and Actn3 genes after Rybp knockdown, $n = 2$. (F, G) Heatmaps showing the ChIP signal of WDR5 at RYBP-targeted active loci (F), and RING1B at RYBP-targeted repressive loci after RYBP knockout (G). Source data are available online for this figure.

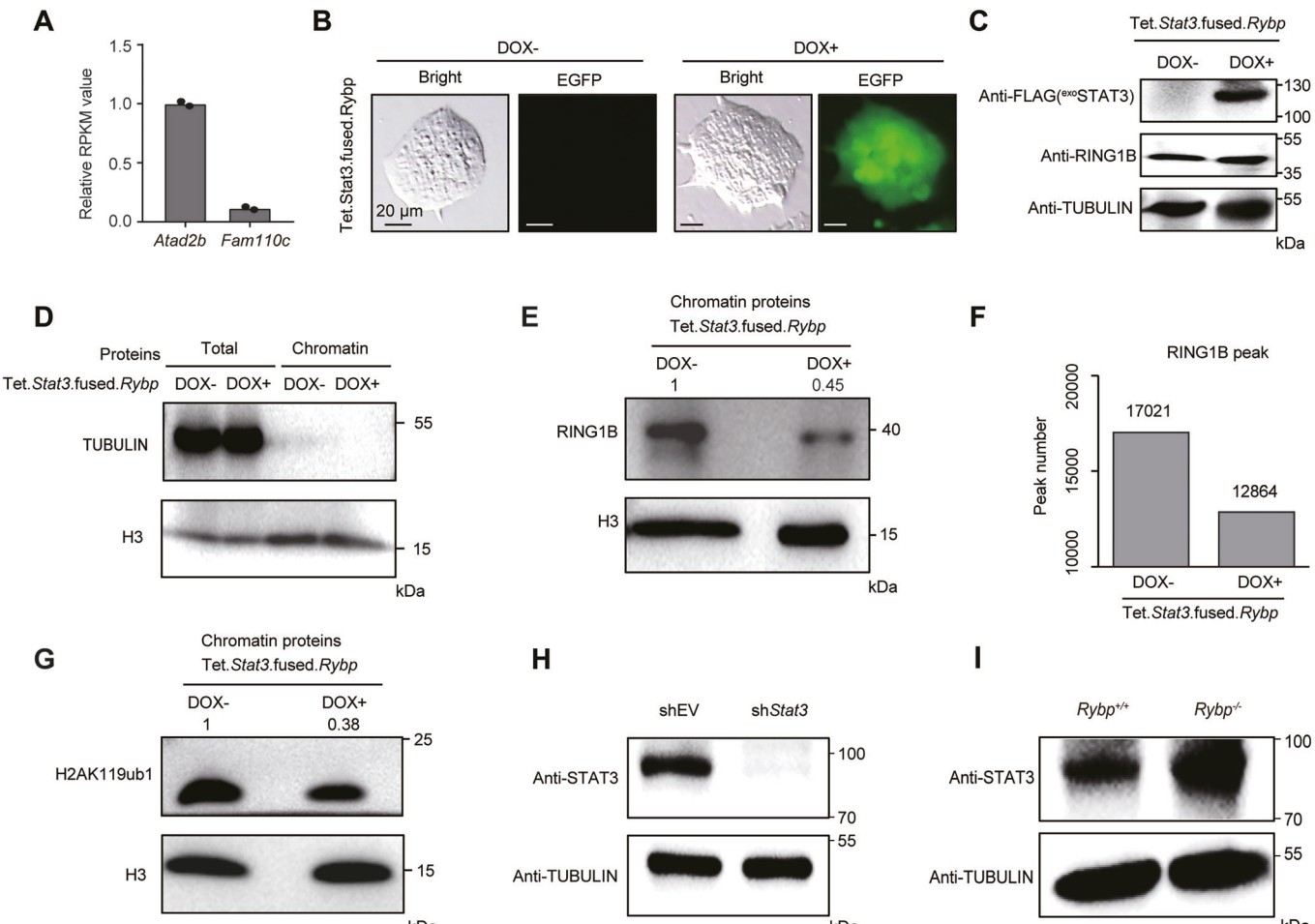

**Figure EV3. STAT3 excludes RING1B on chromatin.**

(A) The histogram showing the relative RPKM value of *Atad2b* and *Fam110c*, n = 2. (B, C) Representative fluorescence images and western blot verifying the successful establishment of Tet.*Stat3*.fused.*Rybp* cell line after DOX addition. (D) Western blot showing the level of TUBULIN and H3 from total and chromatin proteins. (E) Western blot showing the deposition of RING1B at chromatin after DOX treatment in Tet.*Stat3*.fused.*Rybp* cells. (F) Number of RING1B peaks before and after inducing expression of STAT3-fused RYBP protein. (G) Western blot showing the deposition of H2AK119ub1 at chromatin after DOX treatment in Tet.*Stat3*.fused.*Rybp* cells. (H, I) Western blot showing the expression of STAT3 after STAT3 knockdown (H) or RYBP depletion (I). Source data are available online for this figure.

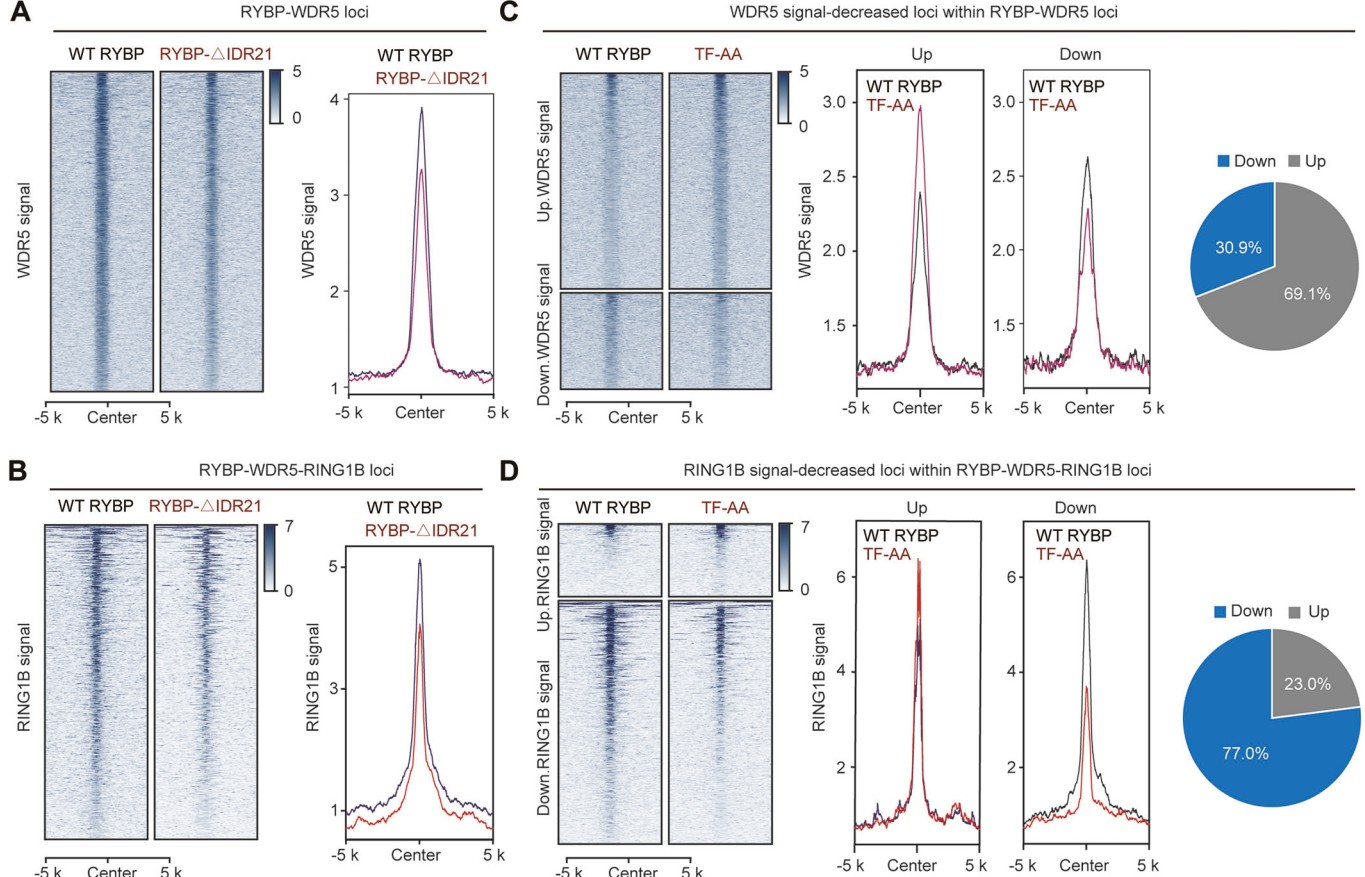

**Figure EV4. RYBP depletion reduces the genomic deposition of H3K27ac and H3K27me3.**

(A, B) The heatmap and curve graph showing the ChIP signal of WDR5 (A) and RING1B (B) at numerous loci following the phase disruption of RYBP. (C, D) Among the signal-decreased loci following RYBP phase disruption, the ratio of upregulation and downregulation of WDR5 (C) and RING1B (D) signals following the TF-AA mutation.

