## [Peer Review File · The EMBO Journal]

RYBP regulates selective genomic binding of TrxG and PcG components in embryonic stem cell fate control

Chao Wei, Jun Sun, Zhuoyan Liu, Mulan Wang, Jin Tan, Xiaona Huang, Ranran Dai, Kang Su, Shiwen Yang, Tara SR Chen, Qi Tian, Xiuxiao Tang, Xiaolin Tian, Dong-feng Huang, Jin Bai, Xue Xiao, Xiaoting Shen, Juan Xia, Junjun Ding, and Lili Fan

Corresponding authors: Junjun Ding (dingjunj@mail.sysu.edu.cn) , Juan Xia (xiajuan@mail.sysu.edu.cn), Xiaoting Shen (shenxt@gdszjk.org.cn), Lili Fan (fanlili@jnu.edu.cn)

Review Timeline:

Submission Date:	16th Sep 25
Editorial Decision:	21st Nov 25
Revision Received:	14th Feb 26
Editorial Decision:	17th Mar 26
Revision Received:	26th Mar 26
Accepted:	14th Apr 26

Editor: Daniel Klimmeck

Transaction Report:

Dear Dr Ding,

Thank you again for the submission of your manuscript (EMBOJ-2025-122466) to The EMBO Journal, as well as for your patience with our feedback at this time. As mentioned earlier, your study was assessed by two reviewers with expertise in stem cell fate decision control and transcription, whose comments are enclosed below.

As you will see from the experts' reports, the referees acknowledge the analysis and potential interest and value of your findings. However, they also express a number of important issues which need to be addressed thoroughly to make them supportive of publication in the EMBO Journal.

In more detail, Referee #2 states that the functional relevance of the results and specifically requirement of RyBP condensation remains unaddressed, which substantially dampens his/her enthusiasm for the work (ref#2, pt.3). This expert also expresses the need for a more rigorous mechanistic exploration of STAT3 involvement in the phenotypes (ref#2, pts.4-6). Reviewer #1 points to major issues with technical robustness and conclusive methods documentation of your study (ref#1, pts.2-5; 7-10). This expert also remains critical regarding the structure and clarity of the manuscript and data presentation. Further, the experts raise a number of issues related to the additional controls required, and statistics applied, that would need to be conclusively addressed to achieve the level of robustness and clarity needed for The EMBO Journal.

Given the overall interest stated and broader angle of your findings, we are able to invite you to revise your manuscript experimentally to address the referees' comments. However, please note that the extent of revisions requested appear threshold in our view for the amount of complementary work we typically invite for our venue; also, I need to stress that we do require strong support from the referees on a revised version of the study in order to move on to publication of the work.

I would appreciate if you could contact me during the next weeks for exchange e.g. a video call to discuss your perspective on the comments and potential plan for revisions.

When submitting your revised manuscript, please carefully review the instructions below.

Please feel free to approach me any time should you have additional questions related to this.

Thank you for the opportunity to consider your work for publication.

I look forward to your revision.

Kind regards,

Daniel Klimmeck

Daniel Klimmeck, PhD
Senior Editor
The EMBO Journal

Instruction for the preparation of your revised manuscript:

Read our guidance for manuscript revisions and related editorial policies: <https://link.springer.com/journal/44318/submission-guidelines#cms-Revised-submissions>

- 1) a .docx formatted version of the manuscript text (including legends for main figures, EV figures and tables). Please make sure that the changes are highlighted to be clearly visible.
- 2) individual production quality figure files as .eps, .tif, .jpg (one file per figure).
- 3) a .docx formatted letter INCLUDING the reviewers' reports and your detailed point-by-point response to their comments. As

part of the EMBO Press transparent editorial process, the point-by-point response is part of the Review Process File (RPF), which will be published alongside your paper.

4) a complete author checklist, which you can download from our author guidelines. Please insert information in the checklist that is also reflected in the manuscript. The completed author checklist will also be part of the RPF.

6) It is mandatory to include a 'Data Availability' section after the Materials and Methods. Before submitting your revision, primary datasets produced in this study need to be deposited in an appropriate public database, and the accession numbers and database listed under 'Data Availability'. Please remember to provide a reviewer password if the datasets are not yet public. In case you have no data that requires deposition in a public database, please state so in this section. Note that the Data Availability Section is restricted to new primary data that are part of this study.

7) Our journal encourages inclusion of *data citations in the reference list* to directly cite datasets that were re-used and obtained from public databases. Data citations in the article text are distinct from normal bibliographical citations and should directly link to the database records from which the data can be accessed. In the main text, data citations are formatted as follows: "Data ref: Smith et al, 2001" or "Data ref: NCBI Sequence Read Archive PRJNA342805, 2017". In the Reference list, data citations must be labeled with "[DATASET]". A data reference must provide the database name, accession number/identifiers and a resolvable link to the landing page from which the data can be accessed at the end of the reference.

8) At EMBO Press we ask authors to provide source data for the main and EV figures. Our source data coordinator will contact you to discuss which figure panels we would need source data for and will also provide you with helpful tips on how to upload and organize the files.

Numerical data can be provided as individual .xls or .csv files (including a tab describing the data). For 'blots' or microscopy, uncropped images should be submitted (using a zip archive or a single pdf per main figure if multiple images need to be supplied for one panel).

9) We replaced Supplementary Information with Expanded View (EV) Figures and Tables that are collapsible/expandable online. A maximum of 5 EV Figures can be typeset. EV Figures should be cited as 'Figure EV1, Figure EV2' etc. in the text and their respective legends should be included in the main text after the legends of regular figures.

<https://media.springernature.com/original/springer-cms/rest/v1/content/27825798/data/v1>

11) For data quantification: please specify the name of the statistical test used to generate error bars and P values, the number (n) of independent experiments (specify technical or biological replicates) underlying each data point and the test used to calculate p-values in each figure legend. The figure legends should contain a basic description of n, P and the test applied. Graphs must include a description of the bars and the error bars (s.d., s.e.m.).

Further information is available in our Guide For Authors: <https://link.springer.com/journal/44318/submission-guidelines#cms-Revised-submissions>

We realize that it is difficult to revise to a specific deadline. In the interest of protecting the conceptual advance provided by the work, we recommend a revision within 3 months (19th Feb 2026). Please discuss the revision progress ahead of this time with the editor if you require more time to complete the revisions.

Referee #1:

The main finding of the work by Wei et al. is that RYBP associates with both TrxG and PRC components. RYBP mediates the binding of the TrxG component WDR5 and the PRC1 component RING1B at genomic loci. Through STAT3, RYBP prevents the binding of RING1B at loci that must remain active. RYBP, WDR5, and RING1B all contain intrinsically disordered regions (IDRs) and can form droplets in vitro. In cells, they form bodies or condensates; however, the presented data do not clearly demonstrate a functional role for these condensates in RYBP activity. Depletion of RYBP promotes ESC differentiation, facilitates the transition of ESCs to the 2CLC state, and downregulates factors involved in DNA repair. Accordingly, RYBP knockout ESCs exhibit elevated DNA damage, as indicated by increased γ H2AX levels. Additional data suggest that RYBP promotes the ESC-to-2CLC transition through increased DNA damage. Overall, the results are potentially interesting. However, the description and presentation of the data are extremely disorganized, making the study difficult to evaluate. There is a lack of precise methodological description, which complicates the assessment of the experiments and their conclusions. The rationale for including certain experiments in the main figures, others in the extended data (EV), and some in the appendix is unclear. In my opinion, there are too many datasets and figure panels, and some of the data representations are not informative, or in some cases, even misleading.

Below are some my major points.

1. The authors should provide some description of RYBP in the Introduction.
2. Additional details for the proteome analyses in the results section should be provided. Many of the experiment are not technical described in the Result section, this should be done. (Figs. 1C, EV1D and Appendix Fig. S1F, EV1A-C)
3. Lanes 130-131: "Thus, RYBP, a phase-separated protein, is a potential factor that regulates the genomic binding of TrxG and PcG proteins." This statement appears too strong in the context of the data presented. Up to this point, the results only show that RYBP associates and colocalizes with TrxG and PcG proteins. Since the authors have not introduced that RYBP binds DNA, readers have no basis to understand how RYBP could directly regulate the genomic binding of these complexes. Therefore, this conclusion should be toned down or better supported by additional information.
4. Appendix Fig. S1L,M. Overexpression of RYBP led to a significant increase in the expression of RYBP-targeted active genes. What does it happen to repressed genes?
5. It is not clear to me how it was done the comparison between the mRNA levels of STAT3 and RYBP in mouse ESCs (Appendix Fig. S2A). Was it a qRT-PCR and normalized to what? For such comparisons it must be demonstrated that primers have the same efficiency. Moreover, I do not understand the reason of this information. I have a similar technical question for EV3A.
6. Lane 214. Consider to modify the sentence "low levels of STAT3 were enriched at RING1B-enriched regions" in "RING1B-enriched regions were depleted of STAT3"
Is the STAT3-fused RYBP construct also tagged with EGFP? The images in EV3B show EGFP expression in dox induced cells. This should be stated in the result section.
7. The ChIPseq data in ESCs expressing STAT3-fused RYBP construct should be represented also with heat maps (Figs. 2J,K and EV3F,) as done for the other ChIPseq data.
The WB for RING1B and H2A119ub on chromatin fractions are very unclear, as the bands are almost cut away (EV3D,E).

Moreover, they need to be quantified and additional controls should be included to show that these are chromatin fractions.

8. I found the description of the RNAseq sparse among EV and Appendix Figures very confused and not all precise. For example, lanes 240-241 "inducing expression of STAT3-fused RYBP leading to an increase in 240 the expression of majority genes (Appendix Fig. S2E)". How many genes? This Figure (heatmap) is not at all informative as it does not show downregulated genes. I think the information that is here relevant is the proportion of genes bound by RYBP and RING1B that become upregulated upon the expression of the STAT3-fused RYBP construct. The representation of the data should be simplified and made clearer, using for example Volcano plots and Venn-diagrams. Similarly, the appendix Figs. S2H,I showing RNAseq data with Gene set enrichment analysis is not the way to show this data. It is also unclear what set of genes were analysed: "upregulated genes upon RING1B deficiency that tended to remain upregulated after the induced expression of STAT3-fused RYBP" (Lanes 243-244) and "RING1B-targeted genes that exhibited upregulation after the induced expression of STAT3-fused RYBP" (lanes 245-246). If they want to show the results of this RNAseq, first they have to describe how the experiment was done (how many days after the washout of dox the analysis was performed?), show that the fusion proteins is not anymore expressed, and then provide the number and proportion of genes bound by RING1B that become upregulated upon STAT3-fused RYBP expression and that remain upregulated after the expression of the fusion protein was turned off. Lanes 247-248 "As an example, TRIM67 functions in brain development." Consider to revise it since this work does not study brain development

The quantification of Ring1B at RING1B-lacked foci is not very convincing (Fig. 2O). It looks like that these sites are bound by RinG1B and the increased signal upon STAT3-KD is minimal.

9. It is not clear how the depletion of RYBP was performed. Figs. 2R-T and EV3H,I were labelled with RYBP^{-/-}, which is usually used for KO and not sh/siRNA-KD. This is a relevant information as it was indicated that these measurements were done after "depletion of RYBP for 2 days" (lane 262).

Lanes 264-265. "Additionally, the depletion of RYBP led to decreased STAT3 signals at RING1B-lacked loci (Fig. 2S,T)." I think that here the authors must be more precise as RING1B-lacked loci is too generic. They should have measured RYBP & STAT3-bound regions (which they show to lack RING1B association).

Similar to the above point. Lanes 268: "RYBP is required for the genomic binding of STAT3." The statement implies that RYBP is required for the binding of STAT3 everywhere in the genome, which I do not think it is the case. The heatmap of Fig. 2R should be stratified according to RYBP-bound and not bound sites. This will determine whether the reduction of STAT3 binding occurs only or not at RYBP-bound sites.

10. Figure 3 is for me very problematic. Why did they introduced TSA-seq when they used the APEX technology to map genomic domains at the RYBP bodies? Why did they do this experiment having already performed a RYBP ChIPseq. Why RYBP ChIPseq tracks of Figs. 3H,I are not shown? In other words, the DNA-FISH could be performed without the APEX-based data since RYBP-associated domains were already provided by the ChIPseq. There is no data showing a role of condensates in RYBP function. As such, the data are just correlative. To study this, they should have generated a RYBP mutant unable to form condensates

Lanes 279-280. Neither Fig. 3A nor Appendix Fig. S3A show that RYBP-proximal DNA was specifically labelled. The schematic of Fig. 3A was not all explained, in particular for the DNA-FISH used for the cut off. Which probes and how many probes were used?

The heatmap of the ChIPseqs (Fig. 3G) was already more or less described in the previous sections (RYBP/WDR5 loci lacking RING2B are active, the ones with RING2B are repressed). Why the GO terms have been placed on the right of the heatmap? The DNA probe for Figure 3C showed a very large signal and it does not overlap with the RYBP bodies, it is adjacent to them. The signal of the histone locus and Pdch are not visible in the merges of Figs. 3H,I.

Why in the model of Fig. 3J there is not RYBP?

11. Figure 4 shows that RYBP depletion decreases the binding of WDR5 and RING1B. This result should have been placed much earlier in the manuscript, even before the STAT3 experiments of Figure 2. Also in this case, the role of the condensates remains marginal as no data were provided to explain whether the condensates play a role. has not be clarified. Note that for this part I have not raises specific points in the analyses as done with the previous results, but only general remarks.

12. Lanes 371-373. The sentence is not complete.

Figure 5. Out of the blue, the authors show data on DNA damage and ESC differentiation, suggesting that RYBP depletion induced DNA damage and promote differentiation. As for the other results, I found very annoying to follow the data by looking at the many data distribute between main Figures, EV data, and Appendix Figures. The experiments shown for ESC differentiation are very unclear. A good way should have been the analysis of pluripotency with AP staining in RYBP depleted cells and the gene expression measurements of mesoderm, ectoderm, and endoderm markers upon LIF withdrawal. Similar analyses should have been done of the overexpression. The representation of the data in EV5A is not at all informative.

13. The transition of ESCs to 2CLC an interesting results. However, also in this case the data are not well represented. Fig. 6E shows an RYBP^{+/+} ESC colony with one cell expressing GFP-MERVL. Why to not to show ESCs upon RYBP-KO? What does it mean "fold changes of percentage" in the y-axis of Figs. 6F,H?

As indicated by the authors, a previous study showed that DNA damage induces the ESC-to-2CLC transition upon DNA

damage. The data showing that RYBP deletion induced this transition via DNA damage is extremely correlative and does not add anything. If this was the case they should find a system where RYBP deletion does not induce DNA damage. The data of Fig. 6H upon treatment with AV-153 or Y27632 is very interesting, but it needs more data to support the results by showing images and quantifications of gH2AX and FACS plots of GFP-MERVL. The results obtained with BRAC1 deficiency are unnecessary and do not add anything related to RYBP as the decrease of the expression of DNA damage factors was already shown in Fig. 5.

Referee #2:

The manuscript entitled "RYBP regulates the selective genomic binding of TrxG and PcG components in phase separation for cell fate control" by Wei et al. investigates the mechanism of RyBp in regulating gene expression in mouse pluripotent stem cells. Following their earlier work the authors perform proteomics of RING1B and WDR5 interactors and identify 15 shared proteins including Rybp. Prompted by a large number of IDR containing proteins the authors investigate properties of phase separation. In vitro recombinant RYBP forms aggregates, which is further shown to recruit TrxG proteins. Interaction of RyBp and TrxG proteins including MLL2/3, SET1A/B methyltransferases, Ash2L and DPY30 is confirmed by APEX or biotin tagged RYBP proteomics in cells.

The authors show RyBp binding at active genes independent of the presence of PcG components but in association with TrxG proteins H3K4me3 and H3K27ac, but not H3K27me3, which is consistent with earlier work (eg Morey et al, 2013 and the authors own work). Active RyBp bound gene promoters are associated with binding sites for STAT3. Conversely, the authors find that Ring1b bound regions are generally low in STAT3 binding, further STAT3-RING1B fusion protein expression tethering STAT3 to RyBp bound regions decreases Ring1b occupancy. Albeit some difficulty in interpreting this experiment, it supports the idea for an antagonism of STAT3 on Ring1b binding. In addition, depleting STAT3 by RNAi, leads to increased Ring1 binding. Genetic deletion of RyBp is used to show that both gene activation and repression are affected, albeit, to various and potentially lineage specific extent. Similarly, STAT3 bindings and Ring1 binding are affected after loss of RyBp. Notably, in pluripotent stem cells loss of RyBp associates with either induction of DNA repair signals or DNA damage leading on to upregulation of ZSCAN4 and Dux genes that have been associated with a transition to a 2 cell like state. The authors infer that this is likely caused by reduced Braca1 binding after RyBp depletion. They find that increase of 2C like cell MERVL reporter upregulation in RyBp or Braca1 depleted ESCs. Lack of 2C reporter induction by RyBp depletion in the presence of inhibitors of DNA damage shows a contribution to induction of 2C like cell states. RyBp depletion also affected lineage gene repression by Ring1 and neural differentiation.

In conclusion, this is a comprehensive study that makes a substantial mechanistic advance on PcG/TrxG regulation. The conclusions are based on high quality evidence and of interest to a wide readership in gene regulation and stem cells. However, the present version falls short on the clear positioning over earlier work and details of the mechanistic model/generalization. A number of points should be considered by the authors before publication can be considered.

Specific points:

1. For Figure 1A it would be important to show some data that the antibodies are indeed of appropriate specificity, in particular TxG antisera have been associated with often cross react with other cellular proteins. The authors have mutant cell lines that could be useful for showing control IPs where the targets are absent to confirm that no interactors are detected. The text describes a large number of interactors are IDR domain proteins, which the reader might suspect to be prone to unspecific interactions in the proteome.
2. WDR5 appears to have a larger number of interactors and the number would suggest that these are in excess of what would be included in TrxG proteins. Is WDR5 a good choice for representing Trithorax complexes?
3. Figure 1 and the first section in the results convincingly demonstrates phase separation properties of RYBP. However, I am wondering if a requirement of phase separation for regulation of TrxG and PcG binding to chromatin can be inferred from the presented data in the study as its title suggests. The authors show clearly that loss of RyBp protein affects STAT3 binding and gene activation as well as repression via Ring1. Could a mutant form of RyBp that does no longer form aggregates but nuclear localizes to show that phase separation properties are functionally relevant. Conversely, it could potentially be interesting if the Zn-finger mutation of RyBp which Ub interaction would result in a phenotype despite the phase separation properties not affected. The authors analysis does not differentiate between absence of RyBp and loss of phase separation at this stage.
4. The observation that STAT3 and Ring1 mutually interact with Rybp and the STAT3 displaces Ring1b from RYBP bound active genes is interesting. From Fig. 2B it appears that there are no Rybp bound genes that are not either STAT3 or Ring1b bound. Is this surprising as STAT3 would appear among other transcription factors and one could naively presume that gene activation on RyBp bound genes would also occur via other factors without STAT3. It would further be interesting if there are STAT3 bound genes that are not RyBp targets and what the difference between these is to Rybp/STAT3 targets.
5. The observation that STAT3 interacts with RyBp in a mutual exclusive manner to a group of PRC1 complexes, is consistent with earlier work of the authors and reports of RyBp function in gene activation and interaction with TFs, and Padi4 (eg Morey

2013, Cell Rep 3:60, Gracia et al 1999, EMBO J 18:3404, and more recent including PMID: 36751888). This suggests RyBp as a structural factor of chromatin for activation and repression. However, it remains unclear if this would lead to a generalizable mechanism. From the data and text it appears that in pluripotent cells all non-repressed and PRC1 bound RyBp active RyBp target genes are targeted by STAT3. Conceptually, this would fundamentally link STAT3 with regulation of PcG mediated repression. Yet, this is surprising as other TFs and transcription signaling mediators latent or active are present in mouse ESCs. It would be important to state if indeed STAT3 overlaps wider with RyBp than any other of the main pluripotency factors to confirm its central position in regulating PcG repression.

6. The authors perform genetic ablation of RyBp and observe a measurable decrease of PRC1 and STAT3 leading on to effects on transcription. The fact that RyBp ablation does not fully abrogate STAT3 and Ring1 binding suggests alternative and parallel pathways. Could the authors speculate to what these mechanisms might be and position RyBp in the wider view.

Minor points

- a) Methods - BirA immunoprecipitation details on nuclear extract and when biotin was added, and the streptavidin pulldown should be briefly included along with protein g beads for flag pull down.
- b) Methods - recombinant proteins (RYBP, etc) production needs a reference for cDNA and purification used in vitro droplet experiments.
- c) Line 209 "The mRNA level of STAT3 was comparable to that of RYBP in mouse ESCs (Appendix Fig. S2A)" could be clarified in which regard similar is to be understood - abundance or regulation.

Point-by-point Responses

We appreciate all the in-depth comments of reviewers. Following their helpful advice, we have conducted experiments, data analysis and further revisions to our manuscript. In this letter, detailed point-by-point responses to the reviewers' comments were provided, along with the revision (highlighted with yellow background) in the revised manuscript.

Reviewer comments:

Referee #1:

The main finding of the work by Wei et al. is that RYBP associates with both TrxG and PRC components. RYBP mediates the binding of the TrxG component WDR5 and the PRC1 component RING1B at genomic loci. Through STAT3, RYBP prevents the binding of RING1B at loci that must remain active. RYBP, WDR5, and RING1B all contain intrinsically disordered regions (IDRs) and can form droplets in vitro. In cells, they form bodies or condensates; however, the presented data do not clearly demonstrate a functional role for these condensates in RYBP activity. Depletion of RYBP promotes ESC differentiation, facilitates the transition of ESCs to the 2CLC state, and downregulates factors involved in DNA repair. Accordingly, RYBP knockout ESCs exhibit elevated DNA damage, as indicated by increased γ H2AX levels. Additional data suggest that RYBP promotes the ESC-to-2CLC transition through increased DNA damage. Overall, the results are potentially interesting. However, the description and presentation of the data are extremely disorganized, making the study difficult to evaluate. There is a lack of precise methodological description, which complicates the assessment of the experiments and their conclusions. The rationale for including certain experiments in the main figures, others in the extended data (EV), and some in the appendix is unclear. In my opinion, there are too many datasets and figure panels, and some of the data representations are not informative, or in some cases, even misleading.

Response: We greatly appreciate the reviewer for the insightful comments, and have carefully addressed each of the concerns raised, which includes conducting additional experiments, clarifying methodology, and reorganizing the figures.

Below are some my major points.

1. The authors should provide some description of RYBP in the Introduction.

Response: We appreciate the reviewer's suggestion. A detailed description of RYBP was provided in the "Introduction" section, which includes its known functions and relevance in the context of our study as the follows:

"PRC1 can be classified into two distinct categories: canonical PRC1, which is defined by the presence of CBX7, and non-canonical PRC1, which is

characterized by the presence of RYBP (Morey et al, 2013; Tavares et al, 2012). RYBP plays a crucial role in modulating H2AK119ub1 levels on PcG target genes, facilitating communication between PRC1 and PRC2, and thereby inhibiting the expression of developmental genes (Morey et al., 2013; Rose et al., 2016). However, genes associated with RYBP exhibit reduced levels of RING1B and H2AK119ub modification, resulting in a higher transcriptional activity compared to those associated with CBX7. Functionally, RYBP-bound genes are predominantly implicated in the regulation of metabolic processes and cell cycle progression (Morey et al., 2013). Further studies have revealed that RYBP can activate the expression of pluripotency genes through a PRC1-independent pathway (Li et al, 2017). Our previous research also demonstrated that RYBP organizes long-range interactions between pluripotency-associated genes to facilitate co-activation via phase separation (Wei et al, 2022), and is required for super-enhancer activity in embryonic stem cells (Hong et al, 2024). In addition to RYBP, several PcG proteins, including PCGF6 and YY1 (Huang et al, 2019; Wang et al, 2018c), have also been reported to exhibit dual functions in transcriptional activation and repression. However, whether proteins with this characteristic are involved in the selective genomic binding of TrxG and PcG components remains to be determined.”

2. Additional details for the proteome analyses in the results section should be provided. Many of the experiment are not technical described in the Result section, this should be done. (Figs. 1C, EV1D and Appendix Fig. S1F, EV1A-C)

Response: We thank the reviewer’s suggestion to improve the manuscript, more detailed descriptions were provided as the follows:

(1) The details for Figs. 1C were provided in the “**Results**” section, and a more detailed description was provided in “**Method**” section.

In “Results” section: In details, the immunofluorescence staining results in mESCs showed that RYBP signals were enriched in the nucleus and exhibited punctate characteristics (Fig. 1C).

In “Method” section: After culturing mouse mESCs on gelatin-coated glass for 24 hours, the cells were fixed with 4% paraformaldehyde (PFA) for 15 minutes at room temperature. Subsequently, the cells were washed twice with PBS and permeabilized with 0.25% Triton X-100 for 5 minutes. Following this, the cells were blocked with 10% BSA (Sigma) for 30 minutes at 37°C and then incubated with primary antibodies overnight at 4°C. After three washes with PBS, the cells were incubated with secondary antibodies for 1 hour at room temperature. Following staining with DAPI for 15 minutes, the cells were imaged using either N-SIM or confocal microscopy.

(2) The details for Fig. EV1D and Appendix Fig. S1F were provided in the “**Results**” section, and a more detailed description was provided in “**Method**”

section.

In “Results” section: By stably expressing exogenous EGFP-tagged RYBP in mESCs, the results also showed that RYBP displayed punctate characteristics (Fig. EV1D). Additionally, Fluorescence Recovery After Photobleaching (FRAP) experiment was performed to detect the dynamics of RYBP puncta, the results indicated that the fluorescence of RYBP puncta quickly diminished after photobleaching, but could rapidly recover, suggesting the dynamics of RYBP puncta (Fig. EV1D and Appendix Fig. S1F).

In “Method” section: Embryonic stem cells (ESCs) stably expressing RYBP-EGFP were grown on gelatin-coated dishes for 24 hours. FRAP experiment were carried out using a Nikon Eclipse Ti microscope equipped with a 100x oil-immersion objective lens and a 488 nm laser. Fluorescence intensity was quantified in the background region, and normalization was done using the fluorescence intensity of an adjacent unbleached cell.

(3) The details for the proteome analyses, including EV1A-C, were provided in the “Results” section, and a more detailed description was provided in “Method” section in revised manuscript.

In “Results” section: In detail, the coding sequences (CDS) for FLAG and biotin-tagged RING1B, FLAG and biotin-tagged WDR5 were transfected into BirA mESCs (Fig. EV1A). Following drug selection, results from the western blot showed the successful establishment of cell lines with exogenously expressed RING1B or WDR5 (Fig. EV1B,C). Subsequently, nuclear extracts from control cells (BirA mESCs) and from those expressing FLAG and biotin-tagged RING1B were subjected to pull-down using streptavidin (SA) beads. The pulled-down proteins were then analyzed by mass spectrometry (Fig. EV1A). To identify the WDR5 interactome, the Stable Isotope Labeling using Amino Acids in Cell Culture (SILAC) method was employed. FLAG and biotin-tagged GFP mESCs, FLAG and biotin-tagged WDR5 mESCs were cultured in media containing different isotopes. Their nuclear extracts were then subjected to anti-FLAG bead pull-down followed by mass spectrometry. The signal intensities were compared with those from the control cells, and a cutoff of 1.5 times the control signal was established to identify significant interactions. A total of 52 proteins were identified as RING1B partners, and 269 proteins were identified as WDR5 partners (Fig. 1A).”

In “Method” section: BirA transgene was firstly introduced into mESCs followed by G418 (300 µg/mL) selection to obtain BirA ESCs. Then the coding sequences (CDS) for FLAG and biotin-tagged GFP, FLAG and biotin-tagged RYBP, FLAG and biotin-tagged RING1B, FLAG and biotin-tagged WDR5 were transfected into BirA mESCs. Following drug selection with 1µg/mL puromycin, stable cell lines of ^{FB}GFP, ^{FB}RYBP, ^{FB}RING1B and ^{FB}WDR5 were established, respectively. Based on the principles from published literature (Kim et al, 2009), BirA utilizes endogenous biotin within the cells to biotinylate the target protein.

Obtaining nuclear extracts: The cell pellet from the above cell lines was

resuspended in three times the pellet volume of Nuclear Extract Buffer A, which contained 10 mM HEPES, 1.5 mM MgCl₂, 10 mM KCl, 0.5 mM DTT, 0.2 mM PMSF, and a protease inhibitor cocktail. The mixture was incubated on ice for 10 minutes, followed by centrifugation at 4,300 x g for 5 minutes. The supernatant was discarded, and the pellet was washed twice with Nuclear Extract Buffer A. After centrifugation and removal of the supernatant, Nuclear Extract Buffer C (containing 20 mM HEPES, 25% glycerol, 1.5 mM MgCl₂, 0.42 M NaCl, 0.2 mM EDTA, 0.5 mM DTT, 0.2 mM PMSF, and a protease inhibitor cocktail) was added at a ratio of 2.5 mL per 1 x 10⁹ cells. The mixture was thoroughly mixed by pipetting and incubated at 4°C for 1 hour with rotation. Subsequently, it was centrifuged at 20,000 x g for 30 minutes at 4°C, and the supernatant was transferred to a new tube. The sample was then diluted with 100 volumes of Buffer D (containing 20 mM HEPES, 0.2 mM EDTA, 1.5 mM MgCl₂, 100 mM KCl, 20% glycerol, and 0.02% NP-40), and was dialyzed thoroughly. After transferring the sample to a new tube, it was centrifuged at 20,000 x g for 20 minutes at 4°C. Then the supernatant was transferred to a new tube as nuclear extracts.

RING1B and RYBP IP-MS: The protocol for immunoprecipitation followed by mass spectrometry (IP-MS) was adapted from our previous study (Costa et al, 2013). Nuclear extracts from BirA (control group), ^{FB}RYBP and ^{FB}RING1B ESCs were subjected to pre-clearing using 0.5 mL of Protein G agarose beads (Thermo Scientific) in IP DNP buffer containing Benzonase (Sigma) overnight at 4°C, incubated with 0.5 mL SA agarose beads (Invitrogen), and rotated for 6 hrs at 4°C. The beads underwent five washes with Buffer D. Elution of the bound proteins was performed by boiling the beads in Laemmli sample buffer for 5 minutes. For identification, samples were separated on a 10% NuPAGE 4%-12% Bis-Tris Gel (Thermo Scientific) and stained with GelCode Blue Safe Protein Stain buffer (Thermo Scientific). Protein bands were excised and sent for whole lane LC-MS/MS sequencing. The number of peptides corresponding to specific proteins in the ^{FB}RYBP or ^{FB}RING1B group exceeds two and is greater than 1.5 times that of BirA group, these proteins are identified as interactors

WDR5 IP-MS: The SILAC labeling strategy was employed to identify the WDR5 protein interactomes, following a previously published protocol with slight modifications (Ong et al, 2002). In brief, ^{FB}GFP ESCs were cultured in "Heavy" SILAC media, which contains ¹³C₆, ¹⁵N₂ L-Lysine and ³C₆, ¹⁵N₂ L-Arginine (CambridgeIsotope Laboratories). ^{FB}WDR5 ESCs were cultured in "Light" SILAC media, which contains ¹²C, ¹⁴N L-Lysine and L-Arginine (Thermo). Then the SILAC-labeled ESCs were trypsinized, washed with DPBS, and then processed for nuclear protein extraction. The affinity purification using anti-FLAG agarose beads (M2, Sigma) was conducted following the above protocol for SA agarose purification, with some modifications. Nuclear extracts were first pre-cleared with Protein G agarose (500 ul slurry per 10 mg protein) for overnight at 4°C with continuous mixing

(rotator wheel) in 15-ml tubes. Then the pre-cleared nuclear extracts were mixed with 500 μ L of anti-FLAG M2 agarose beads. The immuno-complexes were subsequently eluted four times for one hour each at 4°C, utilizing 0.3 mg/mL FLAG peptide in Buffer D, which contained 0.02% NP-40. Protein bands were excised and sent for LC-MS/MS sequencing. Proteins in the ^{FB}WDR5 group were identified as interactors if their signal intensities were greater than 1.5 times those of the ^{FB}GFP group.”

3. Lanes 130-131: "Thus, RYBP, a phase-separated protein, is a potential factor that regulates the genomic binding of TrxG and PcG proteins." This statement appears too strong in the context of the data presented. Up to this point, the results only show that RYBP associates and colocalizes with TrxG and PcG proteins. Since the authors have not introduced that RYBP binds DNA, readers have no basis to understand how RYBP could directly regulate the genomic binding of these complexes. Therefore, this conclusion should be toned down or better supported by additional information.

Response: We appreciate the reviewer's suggestions, and have toned down the description.

We describe our findings regarding the association and colocalization of RYBP with TrxG and PcG proteins as follows:

“Thus, RYBP, a phase-separated protein, co-localizes with PcG and TrxG components in condensates, respectively.”

4. Appendix Fig. S1L,M. Overexpression of RYBP led to a significant increase in the expression of RYBP-targeted active genes. What does it happen to repressed genes?

Response: We appreciate the reviewer's comments, RNA-seq was performed to analyze the expression of repressed genes upon RYBP overexpression.

Results from RNA-seq revealed numerous significantly changed genes upon RYBP overexpression (**Response R1 Figure 1A**). However, at a global level, **the expression of repressed genes does not show significant changes following RYBP overexpression (Response R1 Figure 1B).**

This result may be reasonable, as RYBP represses the expression of these genes, which are already expressed at low levels. Therefore, even if there is further repression of their expression upon RYBP overexpression, the alteration may be minor. **An explanation was provided in “Discussion” section** as the follows: “Our results indicate that RYBP participates in both transcriptional activation and repression. However, at a global level, the expression of repressed genes does not show significant changes following RYBP overexpression. This finding may be reasonable, as RYBP represses the expression of these genes, which are already expressed at low levels. Therefore, even if there is further repression of their expression upon RYBP overexpression, the alteration may be minor.”

The new figures were provided in Appendix Figure S1P-S1Q of revised manuscript.

Figure for reviewers removed

5. It is not clear to me how it was done the comparison between the mRNA levels of STAT3 and RYBP in mouse ESCs (Appendix Fig. S2A). Was it a qRT-PCR and normalized to what? For such comparisons it must be demonstrated that primers have the same efficiency. Moreover, I do not understand the reason of this information. I have a similar technical question for EV3A.

Response: We appreciate the reviewer's comments and apologize for the unclear description.

(1) The comparison of mRNA levels was based on the RPKM values from RNA-seq data, rather than qRT-PCR. The values were normalized to *Rybp* in Appendix Fig. S2A, as well as to *Atad2b* in EV3A.

(2) Figure S2A emphasizes that STAT3 is indeed expressed in ESCs. We observed that STAT3 motifs are enriched at RING1B-lacking sites (**Response R1 Figure 2A**); However, we are uncertain whether STAT3 is functionally expressed in ESCs. RYBP is expressed in ESCs, and the RPKM value of STAT3 is similar to that of RYBP (**Response R1 Figure 2B**), indicating that STAT3 is also expressed in ESCs. Furthermore, the western blot results presented in Figure EV3H support this finding (**Response R1 Figure 2C**).

(3) Figure EV3A provides support at the expression level for Figure 2C that *Atad2b* tends to activate, while *Fam110c* tends to repress. In Figure 2C of the original manuscript, *Atad2b* is inclined to activate regions lacking RING1B, while *Fam110c* is associated with regions enriched in RING1B that tend to exert a repressive effect (**Response R1 Figure 2D**). The higher RPKM value of *Atad2b* than that of *Fam110c* further supports their transcription state (**Response R1 Figure 2E**).

(4) The two figures were revised figures with clear y-axis (Response R1 Figures 2B and 2E).

(5) A clearer description for the data interpretation and reason has been provided in the revised manuscript as the follows:

“RNA-seq result in ESCs revealed that the rpkm value of Stat3 was comparable to that of Rybp (Appendix Fig. S2A), suggesting that Stat3 was expressed in ESCs.”

“The higher RPKM value of Atad2b compared to that of Fam110c further supports the transcriptionally active state of Atad2b and the transcriptionally repressive state of Fam110c (Fig. EV3A).”

The new figures were provided in Appendix Fig. S2A and Fig. EV3A of revised manuscript, other figures of Response R1 Figures 2A, 2C and 2L are the original figures of manuscript.

Figure for reviewers removed

6. Lane 214. Consider to modify the sentence "low levels of STAT3 were enriched at RING1B-enriched regions" in "RING1B-enriched regions were depleted of STAT3" Is the STAT3-fused RYBP construct also tagged with EGFP? The images in EV3B show EGFP expression in dox induced cells. This should be stated in the result section.

Response: We thank the reviewer's suggestions and apologize for the unclear description. A clearer description was provided in revised manuscript.

(1) The original sentence was replaced according to the suggestion in the revised manuscript as the follows: “In contrast, RING1B-enriched regions

were depleted of STAT3 (Fig. 2B)”

(2) The constructs are regulated by the same bidirectional promoter responsive to DOX and rtTA, allowing for independently driving their expression. A clearer description was provided in Fig.2I (**Response R1 Figure 3**), “Result” and “Method” section of revised manuscript as the follows:

In “Result” section: “A cell line expressing both STAT3-fused RYBP and EGFP construct induced by DOX was established (Figs. 2I and EV3B,C)”

In “Method” section: “To establish the fusion expression of RYBP and STAT3 cell line, the reverse tetracycline-controlled transactivator (rtTA) was first expressed randomly in ESCs. Then a plasmid containing a bidirectional promoter (TRE-BI) responsive to DOX and rtTA, which can independently drive the expression of EGFP and the STAT3-fused RYBP, was transfected into rtTA ESCs using Fugene (Promega).”

The new figures were provided in Fig.2I of revised manuscript.

Figure for reviewers removed

7. The ChIPseq data in ESCs expressing STAT3-fused RYBP construct should be represented also with heat maps (Figs. 2J,K and EV3F,) as done for the other ChIPseq data. The WB for RING1B and H2A119ub on chromatin fractions are very unclear, as the bands are almost cut away (EV3D,E). Moreover, they need to be quantified and additional controls should be included to show that these are chromatin fractions.

Response: We appreciate the reviewer's suggestions, and have provided the heatmaps, full information WB data, quantification and controls.

(1) Heatmaps of Figs. 2J,K and EV3F, as well as other figures were provided. The heatmaps of Figs. 2J,K and EV3F from original manuscript were provided (**Response R1 Figures 4A-4C**). Other figures, including the signals of STAT3 at RING1B-lacked and STAT3-deposited loci after RYBP knockout (**Response R1 Figure 4D**), as well as the heatmaps of WDR5 and RING1B enrichment signals following RYBP KO (**Response R1 Figures 4E-4F**), STAT3, WDR5 and RING1B enrichment signals following RYBP phase disruption (**Response R1 Figures 4G-4I**), have also been provided.

(2) WB figures with full information bands, quantification and controls, have been provided (Response R1 Figures 4J-4K), confirming that the conclusions remain unchanged. The extracted chromatin proteins contained

minimal cytoplasmic protein contamination (**Response R1 Figure 4L**).

The new figures were provided in Fig. 2K, 2R, Appendix Fig. S2E, S2F, S4P; Fig. EV2F, 2G, 3D, 3E, 3G, 4A, 4B of revised manuscript.

Figure for reviewers removed

8. I found the description of the RNAseq sparse among EV and Appendix Figures very confused and not all precise. For example, lanes 240-241 "inducing expression of STAT3-fused RYBP leading to an increase in 240 the expression of majority genes (Appendix Fig. S2E)". How many genes? This Figure (heatmap) is not at all informative as it does not show downregulated genes. I think the information that is here relevant is the proportion of genes bound by RYBP and RING1B that become upregulated upon the expression of the STAT3-fused RYBP construct. The representation of the data should be simplified and made clearer, using for example Volcano plots and Venn-diagrams.

Similarly, the appendix Figs. S2H,I showing RNAseq data with Gene set enrichment analysis is not the way to show this data. It is also unclear what set of genes were analysed: "upregulated genes upon RING1B deficiency that tended to remain upregulated after the induced expression of STAT3-fused RYBP" (Lanes 243-244) and "RING1B-targeted genes that exhibited upregulation after the induced expression of STAT3-fused RYBP" (lanes 245-246). If they want to show the results of this RNAseq, first they have to describe how the experiment was done (how many days after the washout of dox the analysis was performed?), show that the fusion proteins is not anymore expressed, and then provide the number and proportion of genes bound by RING1B that become upregulated upon STAT3-fused RYBP expression and that remain upregulated after the expression of the fusion protein was turned off.

Lanes 247-248 "As an example, TRIM67 functions in brain development." Consider to revise it since this work does not study brain development.

The quantification of Ring1B at RING1B-lacked foci is not very convincing (Fig. 2O). It looks like that these sites are bound by RinG1B and the increased signal upon STAT3-KD is minimal.

Response: We thank the reviewer's suggestions. Volcano plot and venn-diagram, as well as more detailed explanation was provided. Description for TRIM67 was revised according to the suggestion. A clearer presentation format for Figure 2O and the global statistics has been provided.

(1) A volcano plot and a Venn diagram showing the expression alterations of genes bound by RYBP and RING1B have been provided. The figures showed the number of both down-regulated and up-regulated genes (**Response R1 Figure 5A**). Correspondingly, the GO analysis in Appendix Figure S2E of the original manuscript has been revised to focus on genes that are targeted by both RYBP and RING1B (**Response R1 Figure 5B**).

(2) The objective for appendix Figs. S2G, H, I has been provided, as well as the number and proportion of changed genes. In our original

manuscript, we validated that the expression of STAT3-fused RYBP excluded the genomic binding of RING1B (**Response R1 Figure 5C**). To further support this conclusion, the gene expression trend of STAT3-fused RYBP might be similar to that seen upon RING1B-deficiency. To test this hypothesis, 3,220 up-regulated genes upon RING1B-deficiency were firstly identified (**Response R1 Figure 5D**). After the expression of STAT3 fused to RYBP, 849 of these genes (26.4%) were also significantly upregulated, while only 323 genes (10%) were significantly downregulated (**Response R1 Figure 5E**). This is what we meant by "upregulated genes upon RING1B deficiency that tended to remain upregulated after the induced expression of STAT3-fused RYBP" in the original manuscript. To avoid misunderstanding, we have replaced the original appendix Figures S2G with **Response R1 Figures 5D-5E**, and provided a detailed description for the objective of these analyses in the revised manuscript (Integrated with the subsequent points, this can be seen in the following third paragraph).

Similarly, to further support that STAT3-fused RYBP excludes the genomic binding of RING1B, the genes targeted by RING1B would also tend to derepress upon inducing the expression of STAT3 fused to RYBP. Therefore, the intention of the original manuscript's appendix Figures S2H and I was to verify this hypothesis. To present this result more clearly, and following the reviewers' previous suggestions, volcano plots and venn diagrams were used to illustrate the expression trends of RYBP and RING1B-targeted genes after the expression of STAT3-fused RYBP (**Response R1 Figures 5A-5B**).

We apologize for our unclear descriptions, which may have led the reviewer to believe that we were examining genes that continued to be upregulated after DOX washout; this was not the intention of our original manuscript. To avoid potential misleading, we have replaced the appendix Figures S2H and I in the original manuscript with **Response R1 Figures 5A and 5B**, and providing a clear description of the purpose for these analyses in the revised manuscript as the follows:

"To further support the exclusion effect of STAT3-fused RYBP on RING1B from a gene expression perspective, we first examined whether the genes co-targeted by RYBP and RING1B showed a tendency to be upregulated following the induction of STAT3-fused RYBP expression. The results indicated that 1,333 genes were upregulated, while only 547 genes were downregulated (Appendix Fig. S2E), with the upregulated genes being significantly enriched in developmental pathways (Appendix Fig. S2F). Next, whether the gene expression trends following the induction of STAT3-fused RYBP expression resembled those observed after RING1B-deficiency were assessed. A total of 3,220 genes were found to be significantly upregulated following RING1B-deficiency (Appendix Fig. S2G). After inducing the expression of STAT3-fused RYBP, 26.4% of these genes were also significantly upregulated, while only 10% of the genes were significantly downregulated (Appendix Fig. S2H)."

(3) According to the suggestion, the description for TRIM67 function has been removed, and it is now described merely as a representative example of a gene with suppressed expression.

“As an example, near the TSS of Trim67 gene, RYBP, WDR5 and RING1B are enriched at this locus, while minimal STAT3 signal is observed”

(4) We apologize for the unclear labeling for Fig. 2O. In fact, the increase of RING1B at the STAT3-KD site in Figure 2O is obvious (as indicated by yellow background region) (**Response R1 Figure 5F**). In the revised manuscript, we employed different colors to distinguish between the two comparison groups, using blue for the control group and red for the STAT3-KD group (**Response R1 Figure 5F**), and the statistics also show that the increase of RING1B after STAT3 KD is significant (**Response R1 Figure 5G**).

The new figures were provided in Fig.2O, 2P, Appendix Fig.S2G-2K of revised manuscript.

Figure for reviewers removed

9. It is not clear how the depletion of RYBP was performed. Figs. 2R-T and

EV3H,I were labelled with RYBP^{-/-}, which is usually used for KO and not sh/siRNA-KD. This is a relevant information as it was indicated that these measurements were done after "depletion of RYBP for 2 days" (lane 262).

Lanes 264-265. "Additionally, the depletion of RYBP led to decreased STAT3 signals at RING1B-lacked loci (Fig. 2S,T)." I think that here the authors must be more precise as RING1B-lacked loci is too generic. They should have measured RYBP & STAT3-bound regions (which they show to lack RING1B association).

Similar to the above point. Lanes 268: "RYBP is required for the genomic binding of STAT3." The statement implies that RYBP is required for the binding of STAT3 everywhere in the genome, which I do not think it is the case. The heatmap of Fig. 2R should be stratified according to RYBP-bound and not bound sites. This will determine whether the reduction of STAT3 binding occurs only or not at RYBP-bound sites.

Response: We appreciate the reviewer's suggestions. The detailed description for "RYBP^{-/-}" was provided, as well as the analysis for RYBP & STAT3-bound regions.

(1) *Rybp*^{-/-} indeed indicates the knockout of RYBP. RYBP knockout was performed in *Rybp*^{fl/fl} *Rosa26::CreERT2* (*Rybp*^{+/+}) ESCs, which was reported in our previous reported work (PMID: 35768498). RYBP-depleted ESCs (*Rybp*^{-/-}) were generated by supplementing the medium with 5 μM 4-hydroxytamoxifen (4-OHT) to delete the endogenous *Rybp* gene in the genome. Our previous work has also validated that treatment with 4-OHT for 2 days is sufficient to achieve RYBP depletion in *Rybp*^{+/+} ESCs (PMID: 35768498). The revised manuscript includes this additional information in both "Result" and "Method" sections as the follows:

In "Result" section: "RYBP depletion (*Rybp*^{-/-}) was performed in *Rybp*^{fl/fl} *Rosa26::CreERT2* (*Rybp*^{+/+}) ESCs for treatment with 4-OHT treatment. A two-day treatment of 4-OHT is sufficient to deplete RYBP in *Rybp*^{+/+} ESCs (Wei et al., 2022) , and this depletion does not result in a reduction in STAT3 expression (Fig. EV3I)"

In "Method" section: "Mouse *Rybp*^{fl/fl} *Rosa26::CreERT2* (Wei et al., 2022) (*Rybp*^{+/+}) ESCs were cultured in ESC medium. RYBP-depleted ESCs (*Rybp*^{-/-}) were generated by supplementing the medium with 5 μM 4-hydroxytamoxifen (4-OHT, Sigma)."

(2) The analysis for STAT3 binding at RYBP & STAT3-bound regions (which they show to lack RING1B association) is provided. We apologize for the unclear description in our original manuscript. The definition of RING1B-lacked loci refers to the regions with RYBP and WDR5 deposition, while lacking RING1B (**Response R1 Figure 2A**). Therefore, the RING1B-lacked and STAT3-deposited loci actually refer to the RYBP and STAT3-bound regions, where lacks the deposition of RING1B (**Response R1 Figures 6A-**

6B). This result can be validated in the example figures of the original manuscript (**Response R1 Figure 6C**). To avoid potential misleading, a clear description of the focused loci has been provided in the figure legend as the follows:

“(R, S) The heatmap and curve graph show the ChIP signal of STAT3 at STAT3-deposited and RING1B-lacked loci following RYBP depletion. The RING1B-lacked loci refer to regions where RYBP and WDR5 are deposited, but RING1B is absent, as defined in Figure 2A.”

(3) After RYBP KO, the enrichment of STAT3 signals at STAT3-deposited loci downregulated, regardless of RYBP binding (Response R1 Figures 6D-6F). A possible explanation for this observation is that RYBP may also regulate the genomic enrichment of STAT3 through indirect pathways. We have included a discussion on this point in the “Discussion” section of the manuscript as the follows:

“In addition, the absence of RYBP leads to a reduction in the genomic binding of STAT3. However, the effect of RYBP on the genomic binding of STAT3 may involve both direct and indirect actions, as the loss of RYBP results in decreased enrichment of STAT3 at both RYBP-bound and non-bound sites.”

The new figures were provided in Fig.2R-2T, Appendix Fig.S2R-2T of revised manuscript.

Figure for reviewers removed

10. Figure 3 is for me very problematic. Why did they introduced TSA-seq when they used the APEX technology to map genomic domains at the RYBP bodies? Why did they do this experiment having already performed a RYBP ChIPseq. Why RYB9 ChIPseq tracks of Figs. 3H,I are not shown? In other words, the DNA-FISH could be performed without the APEX-based data since RYBP-associated domains were already provided by the ChIPseq.

Response: We appreciate the reviewer's comments. The explanation for the role of RYBP APEX-DNA experiment was provided in the revised manuscript, as well as the RYBP track of of Figs. 3H,I.

(1) Explanation for the role of RYBP APEX-DNA experiment: Figure 1 mentioned that RYBP can selectively interact with WDR5 and RING1B to form distinct condensates. Therefore, our next focus is to investigate whether RYBP regulates gene expression within these condensates. Therefore, it is essential to identify the DNA present within the RYBP bodies. ChIP-seq can only reveal the DNA regions bound by RYBP protein, but these regions may not be located within RYBP condensates. Therefore, APEX-DNA-seq was used to identify the DNA within the RYBP bodies. An explanation for the aim of this figure was provided at the very beginning of the corresponding paragraph in the revised manuscript as the follow:

“Previous results indicate that RYBP selectively interacts with RING1B at specific genomic regions, and it also selectively forms condensates with RING1B. Next, we aim to determine whether RYBP regulates gene expression within these condensates. Thus, it is essential to identify the genes located within RYBP condensates.”

(2) RYBP track of of Figs. 3H,I was provided (Response R1 Figures 7A-7B).

(3) The explanation for the DNA-FISH experiment is based on APEX-based data rather than ChIP data. APEX-DNA-seq was used to quantify the DNA present within RYBP condensates, while DNA-FISH serves to further validate whether this DNA is indeed located within the RYBP condensates. ChIP-seq reveals regions of DNA bound by RYBP, but if the amount of RYBP binding at these sites is low and not in the form of condensates, the DNA-FISH results may not show its presence within the RYBP puncta.

The new figures were provided in Fig.3H, 3I of revised manuscript.

Figure for reviewers removed

There is no data showing a role of condensates in RYBP function. As such, the data are just correlative. To study this, they should have generated a RYBP mutant unable to form condensates

Response: We appreciate the reviewer's great comment, which is important to improve the present work. The genomic binding of STAT3, WDR5 and RING1B, as well as the alteration of gene expression after the phase disruption of RYBP was provided.

(1) Our previously published article (PMID: 35768498) validated that a mutant of RYBP can disrupt its aggregation. In endogenous RYBP-inducible knockout ESCs, we ectopically expressed full-length RYBP (WT RYBP) and the RYBP with the 21 amino acid deletion (RYBP- Δ IDR21) (**Response R1 Figure 8A**), and flow cytometry was used to obtain cell lines with consistent levels of exogenous protein expression (**Response R1 Figure 8B**). Both image data and statistical results indicated that the aggregation of RYBP- Δ IDR21 significantly reduced (**Response R1 Figures 8C-8D**). This published article (PMID: 35768498) also provides substantial additional evidence supporting the disruptive effect of this mutation on the phase separation of RYBP.

(2) Phase disruption of RYBP reduced the genomic binding of STAT3, WDR5 and RING1B. Compared to WT RYBP ESCs, results from ChIP-seq revealed that phase disruption of RYBP (RYBP- Δ IDR21) impaired the genomic binding of STAT3 and WDR5 in RYBP-WDR5 loci (**Response R1 Figures 8E-8F**), as well as impairing RING1B deposition in RYBP-WDR5-RING1B loci (**Response R1 Figure 8G**).

(3) Phase disruption of RYBP increased the expression of developmental genes, and reduced the expression of DNA repair genes. In original manuscript, loss of RYBP leads to the upregulation of numerous developmental genes, while downregulating the expression of DNA repair genes (**Figure 5 of manuscript**). RNA-seq in WT RYBP and RYBP- Δ IDR21 ESCs were additionally performed, which obtained the similarly results (**Response R1 Figures 8H-8I**).

All the newly generated figures could be observed in Appendix Fig.S4P, S5A, S5M, Fig.EV4A-4B of revised manuscript, Response R1 Figures 8A-8D could be observed in our published article (PMID: 35768498).

Figure for reviewers removed

Lanes 279-280. Neither Fig. 3A nor Appendix Fig. S3A show that RYBP-proximal DNA was specifically labelled. The schematic of Fig. 3A was not all explained, in particular for the DNA-FISH used for the cut off. Which probes and how many probes were used?

Response: We appreciate the reviewer's suggestions, all the information was provided in the revised manuscript.

(1) RYBP-proximal DNA, specifically labeled in Fig. 3A or Appendix Fig. S3A, was provided. According to the suggestions, the RYBP-proximal DNA was labelled (Red fragment in **Response R1 Figures 9A**) in Fig. 3A of revised manuscript. DNA, which is absence of RYBP adjacent, was not labeled with biotin (Blue fragment in **Response R1 Figures 9A**). To determine the specificity of the enriched signals in Appendix Fig. S3A, qPCR results were provided to verify that the DNA identified by RYBP-APEX-DNA-seq as high (above the cutoff) was indeed significantly enriched compared to the Input (**Response R1 Figures 9B**). In contrast, the enrichment of DNA below the cutoff did not show a signal stronger than that of the Input (**Response R1 Figures 9C**).

(2) The criteria for setting the cutoff were also provided in the revised manuscript. The cutoff value of 0.2924 was determined based on the inflection point of the fitted cubic polynomial curve ($y=2.42-17.1x+45x^2-38.8x^3$). Specifically, the inflection point corresponds to the location where the second derivative equals zero, indicating a transition in curvature and therefore a change in the rate of increase of the response variable. The biological phenomenon revealed is that DNA above the cutoff is minimally affected by the enrichment score, and it is spatially very close to the puncta. This suggests that the DNA classified as above the cutoff is in close proximity to the puncta, indicating a strong association with the RYBP condensates. The criteria for setting the cutoff and biological explanation were provided in “Method” and “Result” section of revised manuscript as the follows:

In “Method” section:

“Criteria for setting the cutoff of APEX-DNA-seq

The cutoff value of 0.2924 was determined based on the inflection point of the fitted cubic polynomial curve ($y=2.42-17.1x+45x^2-38.8x^3$). Specifically, the inflection point corresponds to the location where the second derivative equals zero, indicating a transition in curvature and therefore a change in the rate of increase of the response variable.”

In “Result” section:

“According to the formula, the inflection point value of the curve is 0.2924. DNA fragments above this value are minimally affected by the enrichment score and are spatially very close to the puncta. Thus, DNA with an enrich score above 0.2924 was identified as residing within RYBP condensates (Fig. 3D).”

(3) The information of probes, including the number and location, was provided as a supplemental table in the revised manuscript. The table was also highlighted in the “Result” section of revised manuscript as the follows:

“Utilizing the distance data from multiple FISH probes (**Appendix Table S1**), we derived a formula describing the distance of DNA to RYBP puncta based

on the enrich score (Fig. 3D)”

The new figures were provided in Fig.3A, Appendix Fig.S3B, S3C of revised manuscript

Figure for reviewers removed

The heatmap of the ChIPseqs (Fig. 3G) was already more or less described in the previous sections (RYBP/WDR5 loci lacking RING2B are active, the ones with RING2B are repressed). Why the GO terms have been placed on the right of the heatmap?

Response: We appreciate the reviewer's comments, we provided annotation information indicating that the loci in Fig. 3G are different from those in the previous figure, and GO terms were removed.

(1) The annotation information was provided to indicate that the loci in Fig. 3G are different from those in the previous figure. The sites in Figure 1 correspond to regions of RYBP ChIP peaks, whereas Fig. 3G shows sites with an enrichment score greater than 0.2924 from RYBP-APEX-DNA-seq. We highlighted this information at the top of revised Figure (**Response R1 Figure 10A**), as well as figure legends as the follows:

“(E) The heatmap showing the different ChIP signals at the WDR5-binding loci with or without RING1B deposition. The enrich scores of these regions are above 0.2924.”

(2) The GO terms on the right side of the heatmap were removed. Originally, the inclusion of the GO terms was intended to be reflected in our final model depicted in Fig. 3J. However, this information is also presented in our original Appendix Fig. S3B-3C (**Response R1 Figures 10B-10C**), so this redundant information has been removed.

The new figures were provided in Fig.3E-3G of revised manuscript

Figure for reviewers removed

The DNA probe for Figure 3C showed a very large signal and it does not overlap with the RYBP bodies, it is adjacent to them. The signal of the histone locus and Pdch are not visible in the merges of Figs. 3H,I.

Response: We appreciate the reviewer's comments to improve our work.

(1) Based on the principles of TSA-seq, the size of the enrichment score reflects the proximity of DNA to the puncta: a higher enrichment score indicates that the DNA is closer to the RYBP puncta. Even with a high enrichment score, the FISH signals do not completely overlap with the puncta in all cells; there are instances where the signals are merely in close proximity.

(2) A more representative image showing the overlap between RYBP puncta and DNA has also been provided for Figure 3C (**Response R1 Figure 11A**).

(3) The FISH signals were obscured within the puncta signals, and inappropriate color combinations rendered the FISH signals in Figs. 3H and I invisible. To avoid misunderstandings, we have changed the colors of the different channels and replaced them with more typical images (**Response R1 Figure 11B**). In this figure, since the FISH signal is still totally embedded within the puncta signal, the position of the overlapping white signal represents the FISH signal (highlighted by a blue box). Fig. 3I displays the green FISH signal along with the white signal indicating the overlapping regions (**Response R1 Figure 11C**).

The new figures were provided in Fig.3B, 3H and 3I of revised manuscript

Figure for reviewers removed

Why in the model of Fig. 3J there is not RYBP?

Response: We appreciate the reviewer's comment, and provided the RYBP in the model of Fig. 3J (**Response R1 Figure 12A**), which was showed with the gray circles, as well as other models across Fig.4~Fig.6 (**Response R1 Figures 12B-12D**).

The new figures were provided in Fig.3J, 4A, 5A and 6O of revised manuscript

Figure for reviewers removed

11. Figure 4 shows that RYBP depletion decreases the binding of WDR5 and RING1B. This result should have been placed much earlier in the manuscript, even before the STAT3 experiments of Figure 2. Also in this case, the role of the condensates remains marginal as no data were provided to explain whether the condensates play a role. has not be clarified. Note that for this part I have not raises specific points in the analyses as done with the previous results, but only general remarks.

Response: We appreciate the reviewer's suggestions to improve the logic of manuscript, and provided the results in Figure1. The genomic binding of STAT3, WDR5 and RING1B, as well as the alteration of gene expression after the phase disruption of RYBP was provided.

(1) The changes in the enrichment of WDR5 and RING1B at RYBP-targeted active loci and RYBP-targeted repressive loci upon RYBP-deficiency are provided in Figure 1 (Response R1 Figures 13A-13B). This is a great suggestion; by explaining the binding patterns of RYBP with WDR5 and RING1B on the genome in Figure 1 and immediately following it with an explanation of how the loss of RYBP affects their genomic binding, we enhance the logical flow of the manuscript.

(2) Since Figure 3 initiates the analysis of RYBP's role in regulating gene expression within its condensates, Figure 4 focuses on the impact of RYBP on the genomic binding of RING1B and WDR5 within the same condensates. The current Figure 4 observes different loci compared to **Response R1 Figures 13A-13B**, with the latter focusing on RYBP-targeted active or repressive loci, while the former concentrates on the loci identified in the phase separation of Figure 3. We recommend retaining Figure 4 in its current position to continue exploring the questions raised in Figure 3.

(3) The data to analyze the relationship between RYBP phase separation and phenotypes has been provided. The genomic binding of STAT3, WDR5, and RING1B, as well as the changes in gene expression following the phase disruption of RYBP, are included. Detailed results can be found in **Response R1 Figure 8.**

The new figures were provided in Fig.EV2F, 2G of revised manuscript.

Figure for reviewers removed

12. Lanes 371-373. The sentence is not complete.

Figure 5. Out of the blue, the authors show data on DNA damage and ESC differentiation, suggesting that RYBP depletion induced DNA damage and

promote differentiation. As for the other results, I found very annoying to follow the data by looking at the many data distribute between main Figures, EV data, and Appendix Figures. The experiments shown for ESC differentiation are very unclear. A good way should have been the analysis of pluripotency with AP staining in RYBP depleted cells and the gene expression measurements of mesoderm, ectoderm, and endoderm markers upon LIF withdrawal. Similar analyses should have been done of the overexpression. The representation of the data in EV5A is not at all informative.

Response: We appreciate the reviewer's suggestions. The incomplete sentence is revised, data in the Figure 5 has been reorganized, AP staining of ESCs and embryoid body differentiation upon RYBP-deficiency and overexpression were assessed. Fig.EV5A with full information is provided.

(1) The incomplete sentence is revised as the follows: “Given that DNA repair genes and lineage genes were enriched at RYBP-WDR5 loci and RYBP-WDR5-RING1B loci, respectively (Fig. 3), the impact of RYBP depletion on expression of DNA repair and lineage genes, as well as its role in DNA damage and ESC differentiation was investigated (Fig. 5A).”

(2) The data in the Figure 5 has been reorganized, and redundant information has been removed for simplification. In revised manuscript, we transferred all the data from the Fig. EV5 that included Figure 5 to Appendix Figure S5, while also removing the detection of pluripotency-related genes during the differentiation process.

(3) After RYBP deficiency, AP staining of ESCs and embryoid body (EB) differentiation was assessed. The results indicated that RYBP deficiency promotes the differentiation of ESCs into the three germ layers. IAA-induced RYBP degradation cells (RYBP-AID) were used for alkaline phosphatase (AP) staining, the results showed that the absence of RYBP significantly increases the proportion of partially and fully differentiated ESC colonies (**Response R1 Figure 14A**). Following RYBP knockout and the removal of LIF for EB differentiation for 12 days, compared to the control group, the expression of several germ layer-related genes was significantly increased, including ectoderm-related genes such as *Gfap* and *Neurod1* (**Response R1 Figure 14B**); mesoderm-related genes such as *Mesp1* and *Twist1* (**Response R1 Figure 14C**); as well as endoderm-related genes *Gata4*, *Sox17* and *Eomes* (**Response R1 Figure 14D**).

(4) After RYBP overexpression, AP staining of ESCs and EB differentiation was assessed. RYBP overexpression did not significantly increases the proportion of partially and fully differentiated ESC colonies (**Response R1 Figure 14E**). This result is consistent with the finding that there is no significant change in repressed genes following the overexpression of RYBP in ESC (**Response R1 Figure 1B**). Following RYBP overexpression during EB differentiation for 12 days, compared to the control

group, the expression of several germ layer-related genes significantly decreased, including ectoderm, mesoderm and endoderm-related genes (**Response R1 Figures 14F-14H**).

(5) The original Figure EV5A merely displayed the up-regulated genes. In revised manuscript, **both up-regulated and down-regulated genes associated with mesoderm and endoderm tissues was provided (Response R1 Figure 14I)**.

The new figures were provided in Appendix Fig.S5D-S5L of revised manuscript.

Figure for reviewers removed

13. The transition of ESCs to 2CLC an interesting results. However, also in this case the data are not well represented. Fig. 6E shows an RYBP+/+ ESC colony with one cell expressing GFP-MERVL. Why to not to show ESCs upon RYBP-KO?

What does it mean "fold changes of percentage" in the y-axis of Figs. 6F,H? As indicated by the authors, a previous study showed that DNA damage induces the ESC-to-2CLC transition upon DNA damage. The data showing that RYBP deletion induced this transition via DNA damage is extremely correlative and does not add anything. If this was the case they should find a system where RYBP deletion does not induce DNA damage. The data of Fig. 6H upon treatment with AV-153 or Y27632 is very interesting, but it needs more data to support the results by showing images and quantifications of gH2AX and FACS plots of GFP-MERVL. The results obtained with BRAC1 deficiency are unnecessary and do not add anything related to RYBP as the decrease of the expression of DNA damage factors was already shown in Fig. 5.

Response: We appreciate the reviewer's suggestions, all the mentioned information and experiments above have been provided in revised manuscript.

(1) Representative images showing the MERVL-GFP signal upon RYBP KO was provided (Response R1 Figure 15A), which shows an increase in GFP signals following RYBP knockout.

(2) The "fold changes of percentage" indicates the fold change in the proportion of MERVL-positive cells in the treatment group compared to the control group. To present this information more clearly, we have revised the labeling of the y-axis and the annotations at the top of the figure (**Response R1 Figures 15B-15C**), while also providing an explanation in the figure legend as the follows:

"(F) Flow cytometry and histogram showed that the percentage of MERVL-GFP positive cells after RYBP depletion. Two-tailed Welch's t-test, $n = 3$ for the two groups. The "Fold change" indicates the fold change in the proportion of MERVL-positive cells in the $Rybp^{+/+}$ group compared to the $Rybp^{-/-}$ group."

"(H) Histogram showed that the percentage of relative MERVL-GFP positive cells after AV-153 and Y27632 treatment in MERVL-GFP $Rybp^{-/-}$ cells. Two-tailed Welch's t-test, $n = 4$ for each group. The "Fold change" indicates the fold change in the proportion of MERVL-positive cells in the AV-153 or Y27632 group compared to the DMSO group."

(3) Images, quantifications of gH2AX and FACS plots of MERVL-GFP were provided upon anti-DNA damage reagent treatment. The results show that both AV-153 and Y27632 treatments significantly reduced the number of gH2AX puncta (**Response R1 Figures 15D-15E**). Additionally, FACS analysis revealed that the proportion of MERVL-positive cells also decreased (**Response R1 Figure 15F**).

(4) We apologize for the unclear description regarding the purpose of the experiments related to BRCA1 deficiency. **The intention was to demonstrate that the reduction in BRCA1 expression indeed leads to subsequent DNA damage and changes in pluripotent phenotypes. This finding may be important in supporting our final model**; therefore, we recommend retaining this data. To avoid potential misunderstandings, we have added a description of the objectives of this portion of the study in the results section:

“Given that RYBP collaborates with WDR5 to activate DNA repair gene expression (Fig. 5B,C), whether RYBP depletion decreased the transcription of DNA repair genes to induce DNA damage, and further causing 2CLC generation, was further investigated. BRCA1, which plays a vital role in DNA break repair (Salunkhe et al, 2024). Brca1 gene localized at RYBP-WDR5 loci in ESCs (Fig. 6I). RYBP depletion reduced the level of BRCA1 protein, along with the increased accumulation of γ H2AX (Fig. 6J). To further investigate whether the reduction in Brca1 expression leads to an accumulation of DNA damage and an increase in the proportion of 2CLC cells, we knocked down Brca1 expression in ESCs (Appendix Fig. S6J).”

The new figures were provided in Fig.6E, 6F, 6H, Appendix Fig.S6G-S6I of revised manuscript

Figure for reviewers removed

Referee #2:

The manuscript entitled "RYBP regulates the selective genomic binding of TrxG and PcG components in phase separation for cell fate control" by Wei et al. investigates the mechanism of RyBp in regulating gene expression in mouse pluripotent stem cells. Following their earlier work the authors perform proteomics of RING1B and WDR5 interactors and identify 15 shared proteins including Rybp. Prompted by a large number of IDR containing proteins the authors investigate properties of phase separation. In vitro recombinant RYBP forms aggregates, which is further shown to recruit TrxG proteins. Interaction of RyBp and TrxG proteins including MLL2/3, SET1A/B methyltransferases, Ash2L and DPY30 is confirmed by APEX or biotin tagged RYBP proteomics in cells.

The authors show RyBp binding at active genes independent of the presence of PcG components but in association with TrxG proteins H3K4me3 and H3K27ac, but not H3K27me3, which is consistent with earlier work (eg Morey et al, 2013 and the authors own work). Active RyBp bound gene promoters are associated with binding sites for STAT3. Conversely, the authors find that Ring1b bound regions are generally low in STAT3 binding, further STAT3-RING1B fusion protein expression tethering STAT3 to RyBp bound regions decreases Ring1b occupancy. Albeit some difficulty in interpreting this experiment, it supports the idea for an antagonism of STAT3 on Ring1b binding. In addition, depleting STAT3 by RNAi, leads to increased Ring1 binding. Genetic deletion of RyBp is used to show that both gene activation and repression are affected, albeit, to various and potentially lineage specific extent. Similarly, STAT3 bindings and Ring1 binding are affected after loss of RyBp. Notably, in pluripotent stem cells loss of RyBp associates with either induction of DNA repair signals or DNA damage leading on to upregulation of ZSCAN4 and Dux genes that have been associated with a transition to a 2 cell like state. The authors infer that this is likely caused by reduced Braca1 binding after RyBp depletion. They find that increase of 2C like cell MERVL reporter upregulation in RyBp or Braca1 depleted ESCs. Lack of 2C reporter induction by RyBp depletion in the presence of inhibitors of DNA damage shows a contribution to induction of 2C like cell states. RyBp depletion also affected lineage gene repression by Ring1 and neural differentiation.

In conclusion, this is a comprehensive study that makes a substantial mechanistic advance on PcG/TrxG regulation. The conclusions are based on high quality evidence and of interest to a wide readership in gene regulation and stem cells. However, the present version falls short on the clear positioning over earlier work and details of the mechanistic model/generalization. A number of points should be considered by the authors before publication can be considered.

Response: We thank the reviewer for your thorough review and valuable suggestions on our manuscript. We carefully considered all the comments, and revised the manuscript according to the suggestions.

Specific points:

1. For Figure 1A it would be important to show some data that the antibodies are indeed of appropriate specificity, in particular TxG antisera have been associated with often cross react with other cellular proteins. The authors have mutant cell lines that could be useful for showing control IPs where the targets are absent to confirm that no interactors are detected. The text describes a large number of interactors are IDR domain proteins, which the reader might suspect to be prone to unspecific interactions in the proteome.

Response: We greatly appreciate the reviewer's suggestions and apologize for the unclear description of our method regarding IP-MS.

In fact, our IP-MS was conducted using Streptavidin beads IP or anti-FLAG beads IP, while also incorporating control groups to remove background signals. **It was not performed using antibodies against WDR5 or RING1B;** therefore, the specificity of WDR5 or RING1B does not affect this result. We will provide a more explicit and clear description of our methods in both the Results and Methods sections of revised manuscript.

(1) RING1B IP-MS was performed in FLAG and biotin tagged RING1B cell lines. Streptavidin beads, rather than anti-RING1B antibody, were used for immunoprecipitation of RING1B interactors. The same immunoprecipitation was performed in BirA ESCs to remove background signals. The detailed description was provided in "method" section as the follows:

"BirA transgene was firstly introduced into mESCs followed by G418 (300 µg/mL) selection to obtain BirA ESCs. Then the coding sequences (CDS) for FLAG and biotin-tagged GFP, FLAG and biotin-tagged RYBP, FLAG and biotin-tagged RING1B, FLAG and biotin-tagged WDR5 were transfected into BirA mESCs. Following drug selection with 1µg/mL puromycin, stable cell lines of ^{FB}GFP, ^{FB}RYBP, ^{FB}RING1B and ^{FB}WDR5 were established, respectively. Based on the principles from published literature (Kim et al, 2009), BirA utilizes endogenous biotin within the cells to biotinylate the target protein.

RING1B and RYBP IP-MS: The protocol for immunoprecipitation followed by mass spectrometry (IP-MS) was adapted from our previous study (Costa et al, 2013). Nuclear extracts from BirA (control group), ^{FB}RYBP and ^{FB}RING1B ESCs were subjected to pre-clearing using 0.5 mL of Protein G agarose beads (Thermo Scientific) in IP DNP buffer containing Benzonase (Sigma) overnight at 4°C, incubated with 0.5 mL SA agarose beads (Invitrogen), and rotated for 6 hrs at 4°C. The beads underwent five washes with Buffer D. Elution of the bound proteins was performed by boiling the beads in Laemmli sample buffer for 5 minutes. For identification, samples were separated on a 10% NuPAGE 4%-12% Bis-Tris Gel (Thermo Scientific) and stained with GelCode Blue Safe Protein Stain buffer (Thermo Scientific). Protein bands

were excised and sent for whole lane LC-MS/MS sequencing. The number of peptides corresponding to specific proteins in the ^{FB}RYBP or ^{FB}RING1B group exceeds two and is greater than 1.5 times that of BirA group, these proteins are identified as interactors”

(2) WDR5 IP-MS was performed in FLAG and biotin tagged WDR5 cell lines. Anti-FLAG beads, rather than anti-WDR5 antibody, were used for immunoprecipitation of WDR5 interactors. The same immunoprecipitation was performed in FLAG and biotin tagged GFP cell lines to remove background signals. The detailed description was provided in “method section” as the follows:

“WDR5 IP-MS: The SILAC labeling strategy was employed to identify the WDR5 protein interactomes, following a previously published protocol with slight modifications (Ong et al, 2002). In brief, ^{FB}GFP ESCs were cultured in “Heavy” SILAC media, which contains ¹³C₆, ¹⁵N₂ L-Lysine and ³C₆, ¹⁵N₂ L-Arginine (CambridgeIsotope Laboratories). ^{FB}WDR5 ESCs were cultured in “Light” SILAC media, which contains ¹²C, ¹⁴N L-Lysine and L-Arginine (Thermo). Then the SILAC-labeled ESCs were trypsinized, washed with DPBS, and then processed for nuclear protein extraction. The affinity purification using anti-FLAG agarose beads (M2, Sigma) was conducted following the above protocol for SA agarose purification, with some modifications. Nuclear extracts were first pre-cleared with Protein G agarose (500 ul slurry per 10 mg protein) for overnight at 4°C with continuous mixing (rotator wheel) in 15-ml tubes. Then the pre-cleared nuclear extracts were mixed with 500 µL of anti-FLAG M2 agarose beads. The immuno-complexes were subsequently eluted four times for one hour each at 4°C, utilizing 0.3 mg/mL FLAG peptide in Buffer D, which contained 0.02% NP-40. Protein bands were excised and sent for LC-MS/MS sequencing. Proteins in the ^{FB}WDR5 group were identified as interactors if their signal intensities were greater than 1.5 times those of the ^{FB}GFP group.”

(3) A general description of the IP-MS for RING1B and WDR5 was also included in the "Results" section as the follows:

“The proteomes of the core factors for PcG (RING1B) and TrxG (WDR5) (Ali & Tyagi, 2017; Gao et al., 2012) were detected by immunoprecipitation mass spectrometry (IP-MS). In detail, the coding sequences (CDS) for FLAG and biotin-tagged RING1B, FLAG and biotin-tagged WDR5 were transfected into BirA mESCs (Fig. EV1A). Following drug selection, results from the western blot showed the successful establishment of cell lines with exogenously expressed RING1B or WDR5 (Fig. EV1B,C). Subsequently, nuclear extracts from control cells (BirA mESCs) and from those expressing FLAG and biotin-tagged RING1B were subjected to pull-down using streptavidin (SA) beads. The pulled-down proteins were then analyzed by mass spectrometry (Fig. EV1A). To identify the WDR5 interactome, the Stable Isotope Labeling using Amino Acids in Cell Culture (SILAC) method was

employed. FLAG and biotin-tagged GFP mESCs, FLAG and biotin-tagged WDR5 mESCs were cultured in media containing different isotopes. Their nuclear extracts were then subjected to anti-FLAG bead pull-down followed by mass spectrometry. The signal intensities were compared with those from the control cells, and a cutoff of 1.5 times the control signal was established to identify significant interactions. A total of 52 proteins were identified as RING1B partners, and 269 proteins were identified as WDR5 partners (Fig. 1A)."

2. WDR5 appears to have a larger number of interactors and the number would suggest that these are in excess of what would be included in TrxG proteins. Is WDR5 a good choice for representing Trithorax complexes?

Response: We have added an analysis of the correlation of WDR5 and other TrxG proteins at H3K4me3-enriched sites, while also toning down the descriptions.

(1) Compared to other components of the TrxG complex, WDR5 exhibits a high correlation at H3K4me3 sites (Response R2 Figure 16A). The primary function of TrxG is to catalyze H3K4me3, and WDR5 shows the highest correlation with H3K4me3 compared to other TrxG components such as RBBP5, MLL2, and SET1A. In RYBP-targeted active loci (RYBP positive, H3K4me3 positive, H3K27me3 negative), WDR5 also demonstrates the highest correlation (**Response R2 Figure 16B**). At repressive sites, PcG often forms bivalent domains characterized by H3K4me3 and H3K27me3. Consequently, in the observed RYBP-targeted bivalent domains (RYBP positive, H3K4me3 positive, H3K27me3 positive), WDR5 again displays the highest correlation (**Response R2 Figure 16C**).

(2) Despite the above analyses, WDR5 cannot fully substitute for TrxG. Therefore, in our manuscript's analysis of RYBP-targeted active loci, we specifically examined sites co-targeted by RYBP and H3K4me3, where significant H3K4me3 presence further reinforces WDR5's representativeness within TrxG at these loci.

(3) Additionally, **we have toned down the description throughout the manuscript regarding the results related to WDR5**, emphasizing that it is a component of TrxG rather than the entirety of TrxG. In the model figures from Fig. 2 to Fig. 6, we have focused solely on WDR5, rather than TrxG (**Response R1 Figure 12**).

The new figures were provided in Appendix Fig.S1S of revised manuscript

Figure for reviewers removed

3. Figure 1 and the first section in the results convincingly demonstrates phase separation properties of RYBP. However, I am wondering if a requirement of phase separation for regulation of TrxG and PcG binding to chromatin can be inferred from the presented data in the study as its title suggests. The authors show clearly that loss of RyBp protein affects STAT3 binding and gene activation as well as repression via Ring1. Could a mutant fro RyBp that does no longer form aggregates but nuclear localizes to show that phase separation properties are functionally relevant. Conversely, it could potentially be interesting if the Zn-finger mutation of RyBp which Ub interaction would result in a phenotype despite the phase separation properties not affected. The authors analysis does not differentiate between absence of RyBp and loss of phase separation at this stage.

Response: We appreciate the reviewer's a lot for these great suggestions, which are important to improve our work. The genomic binding of STAT3, WDR5 and RING1B, as well as the alteration of gene expression after the phase disruption and Zn-finger mutation of RYBP was provided.

(1) Our previously published article (PMID: 35768498) validated that a mutant of RYBP can disrupt its aggregation. In endogenous RYBP-inducible knockout ESCs, we ectopically expressed full-length RYBP (WT RYBP) and the RYBP with the 21 amino acid deletion (RYBP- Δ IDR21) (**Response R1 Figure 17A**), and flow cytometry was used to obtain cell lines with consistent levels of exogenous protein expression (**Response R1 Figure 17B**). Both image data and statistical results indicated that the aggregation of RYBP- Δ IDR21 was significantly reduced (**Response R1 Figures 17C-17D**). We confirmed that the localization of RYBP- Δ IDR21 in the nucleus (**Response R1 Figure 17E**). This published article (PMID: 35768498) also provides substantial additional evidence supporting the disruptive effect of this mutation on the phase separation of RYBP.

(2) Phase disruption of RYBP reduced the genomic binding of STAT3, WDR5 and RING1B. The previously published article confirmed that the TF-

AA mutation within the zinc finger domain of RYBP impaired its physical interaction with ubiquitinated proteins (PMID: 17070805). We verified that the TF-AA mutation does not significantly reduce the number of RYBP puncta (**Response R1 Figure 17F**). Results from ChIP-seq revealed that the disruption of RYBP phase separation (RYBP- Δ IDR21) impaired the deposition of STAT3, WDR5 and H3K4me3 in numerous RYBP-WDR5 loci, as well as impairing the deposition of RING1B and H2AK119ub1 in numerous RYBP-WDR5-RING1B loci (**Response R1 Figures 17G-17K**). Regarding these signal-impaired loci, the TF-AA mutation reduced STAT3 binding at 49.8% of the loci, while 50.2% of the loci did not show a reduction (**Response R1 Figure 17L**). WDR5 did not decrease at 69.1% of the loci (**Response R1 Figure 17M**). This indicates that RYBP's phase separation contributes to the signal enrichment at these sites by approximately 50% or even higher.

Regarding the enrichment of RING1B, 77% of the loci showed a decrease in RING1B signal after the TF-AA mutation (**Response R1 Figure 17N**). This suggests that RYBP's phase separation contributes to at least 23% of the reduction in RING1B signal at these loci. The significant decrease in RING1B signal due to the TF-AA mutation is reasonable, as previously published data indicate that the zinc finger domain of RYBP is necessary for the integration of RYBP and RING1B into the PcG body (PMID: 17070805). Although there was no significant reduction in the number of RYBP puncta, the content of RING1B within the phase separation of RYBP was reduced. This could be an important reason for the substantial decrease in RING1B enrichment across the genome. The alteration of components within RYBP phase separation, leading to changes in their genomic binding, also supports the importance of RYBP phase separation for RING1B genomic binding. We also provided an explanation for these results in the "Discussion" section as the follows:

"By comparing with a mutation that does not disrupt the phase separation of RYBP, disruption of RYBP phase separation via 21aa deletion revealed a significant reduction in the enrichment signals of STAT3 and WDR5 at a large proportion of loci. This underscores the importance of RYBP's phase separation for their genomic binding. However, among the RING1B signal-decreased loci upon RYBP phase disruption, the Zn-finger mutation also detected a significant reduction (77%). This result might be reasonable, as previously published data indicate that the zinc finger domain of RYBP is necessary for the integration of RYBP and RING1B into the PcG body (Arrigoni et al., 2006). Although there was no significant reduction in the number of RYBP puncta, the content of RING1B within RYBP phase separation might reduce. This could be an important reason for the substantial decrease in RING1B enrichment across the genome. The alteration of components within RYBP phase separation, leading to changes in their genomic binding, also supports the importance of RYBP phase separation for RING1B genomic binding."

(3) Phase disruption of RYBP increased the expression of

developmental genes, and reduced the expression of DNA repair genes. In our original manuscript, we revealed that the loss of RYBP leads to the upregulation of numerous developmental genes, while downregulating the expression of DNA repair genes (**Figure 5 of manuscript**). Among the up-regulated genes upon RYBP phase disruption, 70.4% of them did not up-regulated upon TF-AA mutation (**Response R1 Figure 17O**), these genes are also enriched in developmental terms (**Response R1 Figure 17P**). Among the down-regulated genes upon RYBP phase disruption, 67.9% of them did not down-regulated upon TF-AA mutation (**Response R1 Figure 17Q**), these genes are also enriched in DNA repair terms (**Response R1 Figure 17R**).

All the newly generated figures could be observed in Appendix Fig.S4N-S4S, S5B-S5C, S5N-S5O Fig.EV4A-4D of revised manuscript, Response R2 Figures 17A-17D could be observed in our published article (PMID: 35768498).

Figure for reviewers removed

4. The observation that STAT3 and Ring1 mutually interact with Rybp and the STAT3 displaces Ring1b from RYBP bound active genes is interesting. From Fig. 2B it appears that there are no Rybp bound genes that are not either STAT3 or Ring1b bound. Is this surprising as STAT3 would appear among other transcription factors and one could naively presume that gene activation on RyBp bound genes would also occur via other factors without STAT3. It would further be interesting if there are STAT3 bound genes that are not RyBp targets and what the difference between these is to Rybp/STAT3 targets.

Response: We appreciate the reviewer's comments. Other potential factors that may be involved in the activation role of RYBP were analyzed, as well as the differences between STAT3-bound genes that are not RYBP targets and those that are co-targeted by RYBP and STAT3 were also examined.

(1) Other factors, such as OCT4 and NMYC, may also participate in the transcriptionally active function of RYBP independently of STAT3. In the RING1B-lacked region, the majority of the sites are indeed covered by STAT3 signals (**Response R2 Figure 18A**). However, there are still a subset of sites in the RING1B-lacked region that lack STAT3 signals (**Response R2 Figure 18A, red fragment**). These locations are enriched with signals from factors such as OCT4, NMYC, CTCF and E2F1 (**Response R2 Figure 18B**), suggesting that they may participate in the transcriptionally active function of RYBP independently of STAT3.

(2) The distinction between STAT3-targeted genes without RYBP and RYBP/STAT3 co-targeted genes was analyzed. 37.8% of STAT3 targets do not have RYBP targeting (**Response R2 Figure 19A**). The RYBP/STAT3 co-targeted genes are mainly enriched in pathways related to transcription regulation, cell population proliferation, and double-strand break repair (**Response R2 Figure 18C**). The STAT3-targeted genes without RYBP are enriched in pathways associated with cholesterol homeostasis, oxidative phosphorylation, and chromatin remodeling (**Response R2 Figure 18D**).

(3) An explanation was provided in “Discussion” section of revised manuscript. The detailed description was integrated in discussion of **Point 5**.

All the newly generated figures could be observed in Appendix Fig.S2L-S2M, S2P-S2Q of revised manuscript.

Figure for reviewers removed

5. The observation that STAT3 interacts with RyBp in a mutual exclusive manner to a group of PRC1 complexes, is consistent with earlier work of the authors and reports of RyBp function in gene activation and interaction with TFs, and Padi4 (eg Morey 2013,Cell Rep 3:60, Gracia et al 1999, EMBO J 18:3404, and more recent including PMID: 36751888). This suggests RyBp as a structural factor of chromatin for activation and repression. However, it remains unclear if this would lead to a generalizable mechanism. From the data and text it appears that in pluripotent cells all non-repressed and PRC1 bound RyBp active RyBp target genes are targeted by STAT3. Conceptually, this would fundamentally link STAT3 with regulation of PcG mediated repression. Yet, this is surprising as other TFs and transcription signaling mediators latent or active are present in mouse ESCs. It would be important

to state if indeed STAT3 overlaps wider with RyBp than any other of the main pluripotency factors to confirm its central position in regulating PcG repression. Response: We appreciate the reviewer's suggestions to improve our work. The overlap between RYBP and other factors was analyzed, and an explanation was provided for the potential mechanism.

(1) Compared to several transcription factors, such as ESRRB, KLF4, NANOG, SOX2, and OCT4, STAT3 and RYBP exhibit a higher overlap ratio in their genomic enrichment (Response R2 Figure 19A).

(2) Multiple factors might be involved in the activation and suppression function of RYBP. The enrichment signals of various factors in RING1B-lacked and RING1B-enriched regions were examined. Some factors, such as ESRRB, KLF4, and NMYC, are enriched in both regions (**Response R2 Figure 19B**), suggesting that they may participate in the transcriptional activation and repression regulation of RYBP simultaneously. In contrast, factors like NANOG and cMYC, similar to STAT3, predominantly enrich in RING1B-lacked regions (**Response R2 Figure 19B**), indicating their potential role in the activation regulation of RYBP. We explained this possibility in “Discussion” section as the follows:

“STAT3 assists RYBP-regulated gene activation. However, it does not exclude the involvement of other factors in the transcriptional regulation process of RYBP. Some factors, such as ESRRB, KLF4, and NMYC, may participate in both the activation and suppression of RYBP, while others, similar to STAT3, primarily contribute to the activation of RYBP, such as NANOG and SOX2. These factors may also play a role in the exclusion of RING1B at activation sites. Nonetheless, the activation regulation of RYBP may not be entirely dependent on STAT3; other factors like OCT4 and NMYC may assist in the transcriptional regulatory role of RYBP at sites where STAT3 is not enriched. However, OCT4 and NMYC were also detected in RING1B-enriched regions, suggesting that they may not be the primary factors responsible for excluding RING1B from the STAT3-deficient within RING1B-lacked regions. It is possible that other factors are involved in this exclusion of RING1B in those regions.”

All the newly generated figures could be observed in Appendix Fig.S2N-S2O of revised manuscript

Figure for reviewers removed

6. The authors perform genetic ablation of RyBp and observe a measurable decrease of PRC1 and STAT3 leading on to effects on transcription. The fact that RyBp ablation does not fully abrogate STAT3 and Ring1 binding suggests alternative and parallel pathways. Could the authors speculate to what these mechanisms might be and position RyBp in the wider view.

Response: We appreciate the reviewer's suggestions to improve our manuscript. The discussion was provided to position RyBp in the wider view as the follows:

“RYBP deletion led to a significant reduction in the genomic signals of RING1B and STAT3, these signals did not completely disappear. The incomplete loss of RING1B may be attributed to the distinction between canonical PRC1 and non-canonical PRC1, with the former lacking RYBP (Morey et al., 2013; Tavares et al., 2012). Although canonical and non-canonical PRC1 often co-bind at genomic loci (Morey et al., 2013), RYBP may not directly interacts with RING1B in canonical PRC1. Therefore, the impact of RYBP deletion on RING1B enrichment in canonical PRC1 may be relatively minor. Regarding STAT3, it possesses specific amino acid sequences that enable it to directly bind to specific DNA sequences such as AT-rich DNA sequence sites (Timofeeva et al, 2012). The acetylation of STAT3 can facilitate its sequence-specific DNA binding ability (Wang et al, 2005). Additionally, various factors might regulate the genomic binding of STAT3. In embryonic stem cells, STAT3 can co-localize with multiple transcription factors, including NANOG, OCT4 and SOX2. The loss of OCT4 also impaired the genomic binding of STAT3 (Chen et al, 2008). Therefore, STAT3 binding may depend partly on direct interactions with DNA sequences or the recruitment of other factors, while RYBP might further enhance the genomic enrichment of STAT3, thereby collaboratively regulating gene expression.”

Minor points

a) Methods - BirA immunoprecipitation details on nuclear extract and when

biotin was added, and the streptavidin pulldown should be briefly included along with protein g beads for flag pull down.

Response: We appreciate the reviewer's suggestions to improve our manuscript, and provided the details for the method.

(1) The details for the obtain of nuclear extracts were provided as the follows:

“Obtaining nuclear extracts: The cell pellet from the above cell lines was resuspended in three times the pellet volume of Nuclear Extract Buffer A, which contained 10 mM HEPES, 1.5 mM MgCl₂, 10 mM KCl, 0.5 mM DTT, 0.2 mM PMSF, and a protease inhibitor cocktail. The mixture was incubated on ice for 10 minutes, followed by centrifugation at 4,300 x g for 5 minutes. The supernatant was discarded, and the pellet was washed twice with Nuclear Extract Buffer A. After centrifugation and removal of the supernatant, Nuclear Extract Buffer C (containing 20 mM HEPES, 25% glycerol, 1.5 mM MgCl₂, 0.42 M NaCl, 0.2 mM EDTA, 0.5 mM DTT, 0.2 mM PMSF, and a protease inhibitor cocktail) was added at a ratio of 2.5 mL per 1 x 10⁹ cells. The mixture was thoroughly mixed by pipetting and incubated at 4°C for 1 hour with rotation. Subsequently, it was centrifuged at 20,000 x g for 30 minutes at 4°C, and the supernatant was transferred to a new tube. The sample was then diluted with 100 volumes of Buffer D (containing 20 mM HEPES, 0.2 mM EDTA, 1.5 mM MgCl₂, 100 mM KCl, 20% glycerol, and 0.02% NP-40), and was dialyzed thoroughly. After transferring the sample to a new tube, it was centrifuged at 20,000 x g for 20 minutes at 4°C. Then the supernatant was transferred to a new tube as nuclear extracts.”

(2) We apologize for the unclear description regarding the biotinylation of the target protein. In fact, endogenous biotin within the cells was used for the biotinylation of the target protein. We have added this explanation about the underlying principle to the methods section as the follows:

“BirA transgene was firstly introduced into mESCs followed by G418 (300 µg/mL) selection to obtain BirA ESCs. Then the coding sequences (CDS) for FLAG and biotin-tagged GFP, FLAG and biotin-tagged RYBP, FLAG and biotin-tagged RING1B, FLAG and biotin-tagged WDR5 were transfected into BirA mESCs. Following drug selection with 1µg/mL puromycin, stable cell lines of ^{FB}GFP, ^{FB}RYBP, ^{FB}RING1B and ^{FB}WDR5 were established, respectively. Based on the principles from published literature (Kim et al, 2009), BirA utilizes endogenous biotin within the cells to biotinylate the target protein.”

(3) We also apologize for the unclear description regarding the FLAG IP-MS. Specifically, it relies on the SALIC method combined with FLAG IP, and does not involve SA IP. Protein G beads were indeed used for the pre-clearing of the nuclear extracts. A detailed description of the method is provided in the revised manuscript and can also be found in our **response to Point 1**.

b) Methods - recombinant proteins (RYBP, etc) production needs a reference

for cDNA and purification used in vitro droplet experiments.

Response: We appreciate the reviewer's suggestions to improve our manuscript, the NCBI accession number of cDNA, as well as the reference for protein purification was provided in the "Method" section as the follows:

"The cDNA corresponding to the target proteins, including RYBP (NM 019743.3), RING1A (NM 009066.3), RING1B (NM 001360844.2), WDR5 (NM 080848.2), RBBP5 (NM 172517.2) and DPY30 (NM 001146222.1), was incorporated into pET28a vectors that contain His-tags and EGFP/mCherry, respectively. Protein expression and purification were performed following reported work (Wei et al., 2022)".

c) Line 209 "The mRNA level of STAT3 was comparable to that of RYBP in mouse ESCs (Appendix Fig. S2A)" could be clarified in which regard similar is to be understood - abundance or regulation.

Response: We appreciate the reviewer's suggestions to improve our manuscript, the sentence was revised.

The original statement intends to express that in the RNA-seq data, the RPKM values of these two genes are similar. Since RYBP is clearly expressed, this supports the notion that STAT3 is also expressed in the cells. A clearer description for Appendix Figure S2A has been provided in the revised manuscript as the follows:

"RNA-seq result in ESCs revealed that the rpk value of Stat3 was comparable to that of Rybp (Appendix Fig. S2A), suggesting that Stat3 was expressed in ESCs."

Dear Dr Ding,

Thank you again for submitting your amended manuscript (EMBOJ-2025-122466R) to The EMBO Journal. Please accept my apologies for the unusual protraction with the reassessment due to delayed expert input at this time of the year and detailed discussion in the editorial team. Your revised study was sent back to the referees for their scientific reassessment, and we have received detailed re-reports from both, which I enclose below. As you will see, the referees state that the work has been substantially enhanced by the revisions and they are now in favour of publication, pending minor revision.

Thus, we are pleased to inform you that your manuscript has been accepted in principle for publication in The EMBO Journal.

Please carefully consider the remaining minor issues still raised by the referees by adjusting data presentation and discussion of the findings in the manuscript text.

Also, we now need you to take care of a number of minor issues related to formatting and data annotation, which I will share shortly in a separate message, together with additional changes and requests by our production team for Source Data provision.

As you might have seen on our web page, every paper at the EMBO Journal now includes a 'Synopsis', displayed on the html and freely accessible to all readers. The synopsis includes a 'model' figure as well as 2-5 one-short-sentence bullet points that summarize the article. I would appreciate if you could provide this figure and the bullet points.

Please submit a revised version of the manuscript using the link enclosed below, addressing the advisor's comments.

Thank you again for giving us the chance to consider your manuscript for The EMBO Journal, I look forward to hearing from you and receiving your final revised version of the manuscript.

Kind regards,

Daniel Klimmeck

>> Author Contributions: Remove the author contributions information from the manuscript text. Note that CRediT has replaced the traditional author contributions section as of now because it offers a systematic machine-readable author contributions format that allows for more effective research assessment. and use the free text boxes beneath each contributing author's name to add specific details on the author's contribution.

More information is available in our guide to authors.
<https://link.springer.com/journal/44318/submission-guidelines>

>> Section order should be as follows: title page with complete author information, abstract, keywords, introduction, results, discussion, methods, data availability section, acknowledgements, disclosure and competing interests statement, references, main figure legends, tables, expanded figure legends.'

>> Funding: Please add the complete list of funders indicated in our system the funding in Acknowledgments in the manuscript: currently the following information is inconsistent: 'Applied Basic Research Foundation (2025B1515020011), the Postdoctoral Fellowship Program of CPSF (GZC20241143) and Natural Science Foundation of Guangdong Province, China (grant no. 2023A1515010197)' the list will be linked to the central database in PubMed in all published articles, it is therefore essential that the list in our system is complete and accurate.

>> Remove the Reagents and Tools table from the manuscript and provide it just as a separate file.

>> Appendix file: author list and affiliations need to be removed from the Appendix.

>> References: add the Dataset citations as [DATASET] entries to the Reference list. Add "Data ref:" as a prefix.

Further information is available in our Guide For Authors: <https://link.springer.com/journal/44318/submission-guidelines>

-----]

Referee #1:

The revised manuscript is greatly improved, particularly in terms of clarity, and the authors have addressed my previous comments well.

I have only one minor comment regarding the "Explanation for the role of the RYBP APEX-DNA experiment." The authors state that APEX-DNA-seq was used to identify the DNA within the RYBP bodies. However, APEX-DNA-seq should in principle capture all RYBP-associated DNA regions and not exclusively those located within condensates. This is a relatively minor point, and I will not insist on further clarification.

Referee #2:

The revised version of the manuscript "RYBP regulates the selective genomic binding of TrxG and PcG components in phase separation for cell fate control" by Wei et al. has additions to the text that have clarified critical experimental detail and new experimental data that address my earlier concerns in a satisfactory way. The authors now include details on the mass spectrometric methods and have investigated the relative overlap of STAT3 and other pluripotency factors with WDR5 and RyBp. The new results further strengthen their earlier conclusions and present a wider view. The observation that RYBP is required for the genomic binding of STAT3 and that STAT3 is the major factor for displacing Ring1B in ESCs is a notable finding. The study advances the idea of WDR5 and Rybp as a binding platform on chromatin consistent with Polycomb bodies and inferred phase separation properties. New data presented on lines 428ff and discussed line 487ff demonstrate that a phase-separation mutant RyBp not sufficient for recruitment. This is in my opinion a critical result allowing to link aggregation properties to regulation in the cells, which and will be of high interest to current research into chromatin regulation.

Minor points

a) line 244, consider "regions lacking RING1B and RING1B enriched regions"

b) A brief statement on the interpretation of the phenotype of RyBp mutation in mice, as reported in Purity et al. (2005) MCB 25:7193, could be added in the light of the new results of the authors. Would the effects on STAT3 recruitment explain the lack of proliferation after implantation or is this likely entirely a consequence of the extraembryonic lineages?

Point-by-point Responses

We appreciate all the comments from reviewers. Following their helpful advice, we have conducted further revisions to our manuscript. In this letter, detailed point-by-point responses were provided, along with the revision in the revised manuscript.

Reviewer comments:

Referee #1:

The revised manuscript is greatly improved, particularly in terms of clarity, and the authors have addressed my previous comments well. I have only one minor comment regarding the "Explanation for the role of the RYBP APEX-DNA experiment." The authors state that APEX-DNA-seq was used to identify the DNA within the RYBP bodies. However, APEX-DNA-seq should in principle capture all RYBP-associated DNA regions and not exclusively those located within condensates. This is a relatively minor point, and I will not insist on further clarification.

Response: We sincerely appreciate the reviewers for providing many important suggestions for the previous version of the manuscript, as well as for their recognition of our previous round of revisions.

We also thank the reviewers for their new comments on the manuscript, an explanation was provided to reveal how to use APEX-DNA-seq to identify the DNA within condensates.

Using spatial distance information obtained from multiple FISH probes in combination with the signal intensity from APEX-DNA-seq, we screened for DNA within the condensates. APEX-DNA-seq indeed captured all RYBP-associated DNA regions, but we actually identified a subset of these within the condensates. According to the principles of TSA-seq described in published articles (PMID: 30154186), the stronger the DNA signal based on proximity labeling, the closer the specific condensates are. Therefore, we used FISH technology to obtain distance information of multiple DNA regions from RYBP condensates. By combining this DNA enrichment signal information from RYBP APEX-DNA-seq, we derived a mathematical formula relating DNA enrichment signals to the distances from RYBP condensates (**Figure 3D of manuscript**), thereby identifying the subset of DNA within RYBP condensates. The detailed description was provided in the manuscript as the follows:

"Protein proximal DNA could be identified by TSA-seq, and their distance to nuclear body was evaluated by TSA-seq score (Chen et al, 2018). Based on the theory, an APEX-based DNA sequencing (APEX-DNA-seq) methodology to evaluate the distance of DNA to RYBP puncta was established (Fig. 3A). Firstly,

a cell line expressing RYBP-APEX fusion protein was established (Appendix Fig. S1I). Upon treatment with biotin-phenol and H₂O₂, RYBP-proximal DNA was specifically labeled (Fig. 3A and Appendix Fig. S3A-C). By comparing with the input DNA, the intensity of RYBP at DNA was quantified using an enrich score (ES) (Fig. 3A). Subsequently, FISH probes targeting DNA with different enrich scores were designed. Single-cell DNA FISH coupled with RYBP immunofluorescence was performed using these probes, and the distance to the nearest RYBP puncta was calculated (Fig. 3A). Consistent with the findings from TSA-seq (Chen et al., 2018), DNA with high enrich scores displayed a closer distance to RYBP puncta (Fig. 3B,C). Conversely, DNA with low enrich scores exhibited a farther distance from RYBP puncta (Fig. 3B,C). Utilizing the distance data from multiple FISH probes (Appendix Table S1), we derived a formula describing the distance of DNA to RYBP puncta based on the enrich score (Fig. 3D). According to the formula, the inflection point value of the curve is 0.2924. DNA fragments above this value are minimally affected by the enrichment score and are spatially very close to the puncta. Thus, DNA with an enrich score above 0.2924 was identified as residing within RYBP condensates (Fig. 3D)."

Referee #2:

The revised version of the manuscript "RYBP regulates the selective genomic binding of TrxG and PcG components in phase separation for cell fate control" by Wei et al. has additions to the text that have clarified critical experimental detail and new experimental data that address my earlier concerns in a satisfactory way. The authors now include details on the mass spectrometric methods and have investigated the relative overlap of STAT3 and other pluripotency factors with WDR5 and RyBp. The new results further strengthen their earlier conclusions and present a wider view. The observation that RYBP is required for the genomic binding of STAT3 and that STAT3 is the major factor for displacing Ring1B in ESCs is a notable finding. The study advances the idea of WDR5 and Rybp as a binding platform on chromatin consistent with Polycomb bodies and inferred phase separation properties. New data presented on lines 428ff and discussed line 487ff demonstrate that a phase-separation mutant RyBp not sufficient for recruitment. This is in my opinion a critical result allowing to link aggregation properties to regulation in the cells, which and will be of high interest to current research into chromatin regulation.

Response: We sincerely thank the reviewers for their important suggestions to improve our work and for their recognition of this work. We have also provided point-by-point modifications and responses to the comments raised.

Minor points

a) line 244, consider "regions lacking RING1B and RING1B enriched regions"
Response: We thank the reviewers for the suggestion, and revised the sentence according to the suggestion as the follow:

"The DNA regions co-localized by RYBP and WDR5 were partitioned into regions lacking RING1B and RING1B-enriched regions"

b) A brief statement on the interpretation of the phenotype of RyBp mutation in mice, as reported in Pirity et al. (2005) MCB 25:7193, could be added in the light of the new results of the authors. Would the effects on STAT3 recruitment explain the lack of proliferation after implantation or is this likely entirely a consequence of the extraembryonic lineages?

Response: We thank the reviewers for the suggestion, a statement with the suggested citation on the interpretation of the phenotype of RyBp mutation, as well as a discussion for the effects of STAT3 recruitment on extraembryonic lineages was provided as the follow:

"Previous studies have shown that STAT3 is necessary for maintenance of embryonic stem cells (Wang et al, 2017), and also regulates the fate determination of trophectoderm cells (Tai et al, 2014). The deficiency of RYBP impairs the genomic binding of STAT3, which may be one of the key reasons why the absence of RYBP leads to the failure of blastocyst survival or the inability to yield extraembryonic cells during early embryonic development (Pirity et al, 2005)."

Dear Dr Ding,

Thank you for submitting the revised version of your manuscript. I have now evaluated your amended manuscript and concluded that the remaining minor concerns have been sufficiently addressed.

I am thus pleased to inform you that your manuscript has been accepted for publication in the EMBO Journal.

Best regards,

Daniel Klimmeck

Daniel Klimmeck, PhD
Senior Editor
The EMBO Journal
EMBO
Postfach 1022-40
Meyerhofstrasse 1
D-69117 Heidelberg
contact@embojournal.org

Please note that it is The EMBO Journal policy for the transcript of the editorial process (containing referee reports and your response letters) to be published as an online supplement to each paper. If you should prefer removal of any referee-only figures included in the point-by-point response(s), e.g. because they may still be used for future publication or because they have been reproduced from published work by others, please do let us know immediately via response email. More information is available here: <https://link.springer.com/partners/embo-press/editorial-policies#Peer%20review>